# Single-neuron representations of odours in the human brain

Marcel S. Kehl[1,2], Sina Mackay[1], Kathrin Ohla[3], Matthias Schneider[4], Valeri Borger[4], Rainer Surges[1], Marc Spehr[5✉] & Florian Mormann[1✉]

Olfaction is a fundamental sensory modality that guides animal and human behaviour[1,2]. However, the underlying neural processes of human olfaction are still poorly understood at the fundamental—that is, the single-neuron—level. Here we report recordings of single-neuron activity in the piriform cortex and medial temporal lobe in awake humans performing an odour rating and identification task. We identified odour-modulated neurons within the piriform cortex, amygdala, entorhinal cortex and hippocampus. In each of these regions, neuronal firing accurately encodes odour identity. Notably, repeated odour presentations reduce response firing rates, demonstrating central repetition suppression and habituation. Different medial temporal lobe regions have distinct roles in odour processing, with amygdala neurons encoding subjective odour valence, and hippocampal neurons predicting behavioural odour identification performance. Whereas piriform neurons preferably encode chemical odour identity, hippocampal activity reflects subjective odour perception. Critically, we identify that piriform cortex neurons reliably encode odour-related images, supporting a multimodal role of the human piriform cortex. We also observe marked cross-modal coding of both odours and images, especially in the amygdala and piriform cortex. Moreover, we identify neurons that respond to semantically coherent odour and image information, demonstrating conceptual coding schemes in olfaction. Our results bridge the long-standing gap between animal models and non-invasive human studies and advance our understanding of odour processing in the human brain by identifying neuronal odour-coding principles, regional functional differences and cross-modal integration.

Olfaction, the sense of smell, is vital for humans[2]. Enhancing our understanding of the underlying neuronal mechanisms is essential, considering the importance of olfaction in health and disease. Olfactory processing commences when airborne odour molecules activate olfactory sensory neurons in the olfactory epithelium (Fig. 1a). Axons of neurons expressing the same olfactory receptor[3] converge onto specific glomeruli in the olfactory bulb, representing odour information as a topographic map of receptor activation[4]. After olfactory bulb processing[4], mitral and tufted cells relay information to several cortical areas that constitute the primary olfactory cortex, including the piriform cortex (PC), amygdala and entorhinal cortex (EC)[5]. Direct projections to the EC are established in rodents[6,7] but have not yet been confirmed in humans[8]. The PC is key for odour processing[9]. In contrast to the olfactory bulb, there is no apparent topography representing odour quality or identity in the PC[1,9–11], raising the question of how odour-specific information is organized within the human PC. While human imaging[12,13] and intracranial electroencephalography[14] studies showed odour-related PC activation at the macroscopic level, recordings in rodents demonstrated odour-related responses of individual PC

neurons[10,15–18], and provided a deeper understanding of odour identity and intensity coding in the PC[19–21]. Besides the PC, multiple medial temporal lobe (MTL) regions contribute to central olfactory processing. In animal models, neurons responsive to odours have been identified in the amygdala, EC and hippocampus[22,23]. Human imaging studies have complemented these findings by demonstrating odour-related activation in these regions (amygdala[13,24], EC[13,24] and hippocampus[25]).

Human single-unit recordings have substantially advanced our conceptual understanding in various areas of neuroscience such as auditory processing[26], object representation[27] and memory formation[28]. However, such studies are lacking in olfaction. In humans, it remains unclear whether and how individual neurons respond to olfactory cues and encode odour identity. We therefore investigated the individual contributions of central olfactory areas to odour processing and their link to human behaviour at the neuronal level. We took advantage of the rare opportunity to record individual neuron activity in the human PC and MTL during an odour rating and identification task. Such single-unit recordings offer unique insights that bridge the long-standing gap between animal electrophysiology and human

[1]Department of Epileptology, University Hospital Bonn, Bonn, Germany. [2]Department of Experimental Psychology, University of Oxford, Oxford, UK. [3]Science & Research, dsm-firmenich, Satigny, Switzerland. [4]Department of Neurosurgery, University Hospital Bonn, Bonn, Germany. [5]Department of Chemosensation, Institute for Biology II, RWTH Aachen University, Aachen, Germany. ✉e-mail: m.spehr@sensorik.rwth-aachen.de; florian.mormann@ukbonn.de

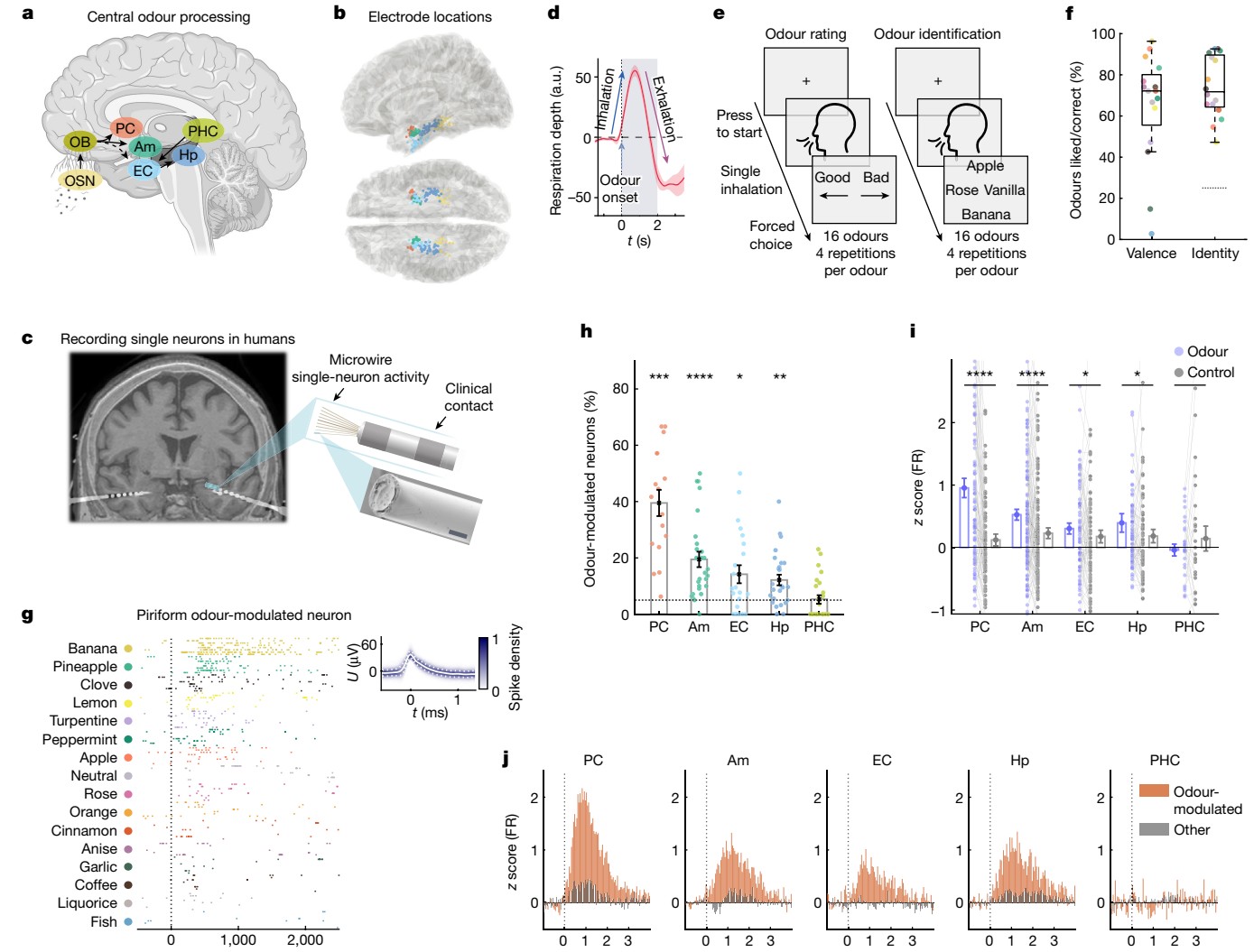

**Fig. 1 | Odours modulate human PC and MTL firing. a**, Odours activate olfactory sensory neurons (OSNs), which project to the olfactory bulb (OB). OB neurons innervate the PC, amygdala (Am) and putatively EC, which is connected to hippocampus (Hp) and PHC. **b**, Innermost clinical electrodes projected to the MNI-ICBM152 template. Sites are coloured as in **a**. **c**, The post-implantation computed tomography (CT) scan, co-registered onto the pre-implantation MRI scan, visualizes Behnke–Fried electrodes (left). Right, schematic (top right) and scalpel-trimmed microwire (bottom right; scanning electron microscopy (SEM)). Scale bar, 20 μm. **d**, Respiratory depth (mean ± s.e.m.) aligned to odour delivery. $n = 13$ sessions. a.u., arbitrary units. **e**, The odour rating and identification task: 15 odours (+1 odourless control) were presented 8 times in a pseudorandom order. Rating: during four presentation cycles, the participants rated (like or dislike) each odour. Identification: next, the participants identified the correct odour (four options; four times per odour). **f**, The behavioural performance per odour, showing ratings (left) and correct identification (right). $n = 27$ sessions. The box plots show the median values (centre lines), 25th–75th percentiles (box limits), and the whiskers span data within 1.5× the interquartile range. Statistical analysis of odour identification was performed using two-sided Wilcoxon signed-rank tests versus chance (25%; dashed line); for all 15 odours, $P < 0.01$. Colours are as in **g**. **g**, Example odour-modulated PC neuron. The firing rate varied significantly with odour identity (left; one-way analysis of variance (ANOVA), $F_{15,112} = 13.8$, $P < 10^{-10}$, $n = 128$ trials). Right, spike-shape density

(mean ± s.d.; white, polarity inverted for visualization). **h**, Odour-modulated neurons per session and region (mean ± s.e.m.). The PC, amygdala, EC and hippocampus host significant populations of odour-modulated neurons (PC, 39.5 ± 4.7%, $n = 17$ sessions, $Z = 3.6$, $P = 0.00029$; amygdala, 19.5 ± 2.7%, $n = 27$, $Z = 4.3$, $P = 1.9 \times 10^{-5}$; EC, 14.2 ± 3.2%, $n = 22$, $Z = 2$, $P = 0.049$; hippocampus, 12.1 ± 1.9%, $n = 27$, $Z = 3.1$, $P = 0.0019$; PHC, 5.31 ± 1.4%, $n = 26$, $Z = -0.27$, $P = 0.78$; two-sided Wilcoxon signed-rank tests versus chance; the dashed line indicates 5%). **i**, Odour-modulated neurons in the PC, amygdala, EC and hippocampus increase their firing rate (FR) after odour stimulation versus the odourless controls (PC, $n = 99$ neurons, $Z = 5.7$, $P = 9.9 \times 10^{-9}$; amygdala, $n = 129$, $Z = 4.1$, $P = 3.4 \times 10^{-5}$; EC, $n = 74$, $Z = 2$, $P = 0.043$; hippocampus, $n = 73$, $Z = 2.3$, $P = 0.019$; PHC, $n = 29$, $Z = -0.49$, $P = 0.63$; all compared with control: $n = 404$, $Z = 7.0$, $P < 10^{-10}$; two-sided Wilcoxon signed-rank tests). The $y$ axis displays 95% of data. **j**, PSTHs (odour-modulated (red) versus other (grey) neurons; 50 ms bins). Odour-modulated neurons increase firing in all regions except in the PHC (two-sided Wilcoxon signed-rank tests comparing $z$-scored firing rates (0–2 s after odour onset) against zero; PC, $n = 99$ neurons, $Z = 5.8$, $P = 7.2 \times 10^{-9}$; amygdala, $n = 130$, $Z = 5.3$, $P = 1.5 \times 10^{-7}$; EC, $n = 74$, $Z = 3$, $P = 0.0028$; hippocampus, $n = 74$, $Z = 2.8$, $P = 0.005$; PHC, $n = 29$, $Z = -0.46$, $P = 0.64$). ****$P < 0.0001$, ***$P < 0.001$, **$P < 0.01$, *$P < 0.05$. Diagrams were created using BioRender (**a**) and Noun Project (**e**).

imaging studies in olfactory research. We identified odour-modulated neurons that effectively encode odour identity. We further demonstrate a distinct role of the amygdala in emotional processing of odours and highlight hippocampal involvement in odour identification.

Notably, not only do our recordings reveal that PC neurons are able to encode the identity of odour-related images, but they also demonstrate cross-modal integration of visual and olfactory information in both the PC and amygdala.

## Odours modulate human PC and MTL firing

Although single-neuron recordings in animal models have greatly advanced our understanding of olfaction, concepts of single-neuron and circuit function in human olfactory processing are largely unexplored. To bridge this knowledge gap, we recorded the activity of single neurons in the human PC and MTL while patients smelled different odours. Overall, we recorded human single-neuron activity (2,416 neurons across 27 sessions) during odour rating and identification tasks in 17 patients undergoing presurgical epilepsy monitoring (Fig. 1b,c and Extended Data Fig. 1). Respiratory measurements confirmed alignment of inhalation with odour presentation (Fig. 1d). Patients reported to have liked the odours in 64.8 ± 2.0% of cases (Fig. 1e,f (left)) and they correctly identified them in 74.1 ± 1.5% of trials (Fig. 1e,f (right); the performance per participant is shown in Extended Data Fig. 2).

First, we investigated whether neuronal firing in the human PC encodes chemical odour identity. Figure 1g shows an example neuron in the left PC that increased firing in response to specific odours. We refer to these neurons, which significantly change their firing based on odour identities, as odour-modulated neurons (further examples are shown in Extended Data Fig. 3). Overall, approximately 40% of PC neurons showed odour-modulated response patterns, emphasizing the role of the PC in odour processing (Fig. 1h). We next examined whether odour-modulated neurons also exist in the human MTL. Whereas early, pioneering multiunit recordings in humans did not provide evidence for odour-specific neurons in the human amygdala[29], we identified a substantial fraction of amygdala neurons exhibiting odour-modulated firing (Fig. 1h). Moreover, we observed a significant set of odour-modulated neurons in the EC and hippocampus (Fig. 1h). Odour-modulated neurons were reliably identified across the participants (Extended Data Fig. 4a). Peri-stimulus time histograms (PSTHs) (Fig. 1j) demonstrate prominent peaks in firing rate after odour onset among odour-modulated neurons in the PC, amygdala, EC and hippocampus, whereas no such increase was observed in the parahippocampal cortex (PHC).

Sniffing odourless air alone has been shown to activate the PC in animal models[30] and in human imaging studies[31,32]. To disentangle putatively mechanosensitive and breathing-related effects from actual chemosensory responses, we included an odourless control. Exposure to odourless controls alone increased firing of odour-modulated neurons, albeit to a significantly lower degree than in response to odours (Fig. 1i). Such differences were most prominent in the PC, but also statistically significant in the amygdala, EC and hippocampus (Fig. 1i). Increased firing rates for odours compared to odourless controls were consistently observed when accounting for participants and sessions (Extended Data Table 1a). Odour-modulated neurons were likewise identified after excluding the odourless control (Extended Data Fig. 4b,c). Respiratory measurements confirmed that odour-modulated neurons were driven by odour-specific characteristics rather than by variability in respiration (Extended Data Fig. 5). Together, our findings firmly establish the existence of odour-modulated neurons both in the human PC and MTL.

## Neuronal activity decodes odour identity

The lack of human single-neuron recordings during olfactory processing has thus far hindered studying the underlying neuronal population codes at high spatial (that is, cellular) and temporal resolution. We therefore assessed how effectively odour identity is represented by neurons in different regions, performing decoding analysis on spiking data[33] (Fig. 2a). Odour identity was predicted from neuronal spiking with high degrees of accuracy in the PC, amygdala, EC and hippocampus (Fig. 2b). Subsampling equal numbers of neurons per region demonstrated the highest decoding performance in the PC, followed by the amygdala and EC (Fig. 2b). Increasing the number of neurons included in decoding further improved performance (Fig. 2c).

Notably, odour identity was reliably decoded by only a small number of neurons, especially in the PC. When systematically varying the decoding time window, odour-identity decoding was fastest in the PC and amygdala. By contrast, approximately a 1 s time window was required to reach above-chance decoding accuracy in the EC and hippocampus (Fig. 2d). Significant odour-identity decoding was observed across the recording sessions (Fig. 2e) and participants (Extended Data Fig. 6a). In conclusion, our results demonstrate effective neuronal odour-identity coding in humans across multiple brain regions involved in odour processing.

## Odour representations vary in sparseness

We next addressed the sparseness of the human olfactory code. To this end, we compared odour representations across regions based on their population sparseness index[34]. Sparseness differed significantly across regions (Fig. 3a). As extracellular recordings tend to omit very sparse neurons in the spike sorting[35], absolute sparseness values are challenging to interpret. Nonetheless, we can compare sparseness across regions given that the same recording and spike-sorting techniques were used. The amygdala and hippocampus showed the sparsest odour coding (Fig. 3a). The population code in the PC was significantly less sparse than that in the MTL areas. Consistent results were obtained when analysing population sparseness separately for each recording session (Extended Data Fig. 7a). Our findings indicate that the degree of sparseness varies significantly along the human olfactory pathway, with the amygdala and hippocampus showing the highest degree of sparseness.

## Neuronal repetition suppression to odours

We also investigated whether and how repeated presentations of the same odour affect responses of odour-modulated neurons. During our paradigm, each odour was presented eight times in a pseudorandom order with an average interpresentation interval for the same odour of approximately 5 min (5.18 ± 0.05 min). Despite this substantial interval, we observed decreasing response activity after repeated presentations in the PC, amygdala and hippocampus (Fig. 3b). This effect was not caused by decreased inhalation (Extended Data Fig. 7b,c). Odour-modulated neurons in the EC showed a decreasing trend that did not reach significance. Repetition suppression was reliably found when factoring in individual participants and sessions (Extended Data Table 1b). Repetition suppression was also observed when, instead of including only odour-modulated neurons, all recorded neurons were considered (Extended Data Fig. 7d). Notably, the response strength reduction in the PC showed a substantial first-trial effect (Fig. 3b and Extended Data Fig. 7e). Together, our analyses reveal differences in sparseness across central odour-processing areas, in conjunction with central repetition suppression.

## Amygdala neurons encode odour valence

The central role of the amygdala in emotional processing is well established[36,37], and rodent studies have revealed valence coding in amygdala neurons[38]. However, animals cannot directly report subjective preferences, and odour-valence coding remains unclear at the individual-neuron level in humans. Consequently, we investigated whether amygdala neurons encode subjective odour preferences and valence. On the basis of individual odour ratings, we compared neuronal responses of odour-modulated neurons to odours rated as liked versus disliked (Fig. 4a). Figure 4b,c shows an example amygdala neuron that responded preferentially to liked odours. Overall, the responses of odour-modulated neurons in the amygdala were significantly greater for liked versus disliked odours (Fig. 4d). No significant difference was observed in other regions. Increased activity of

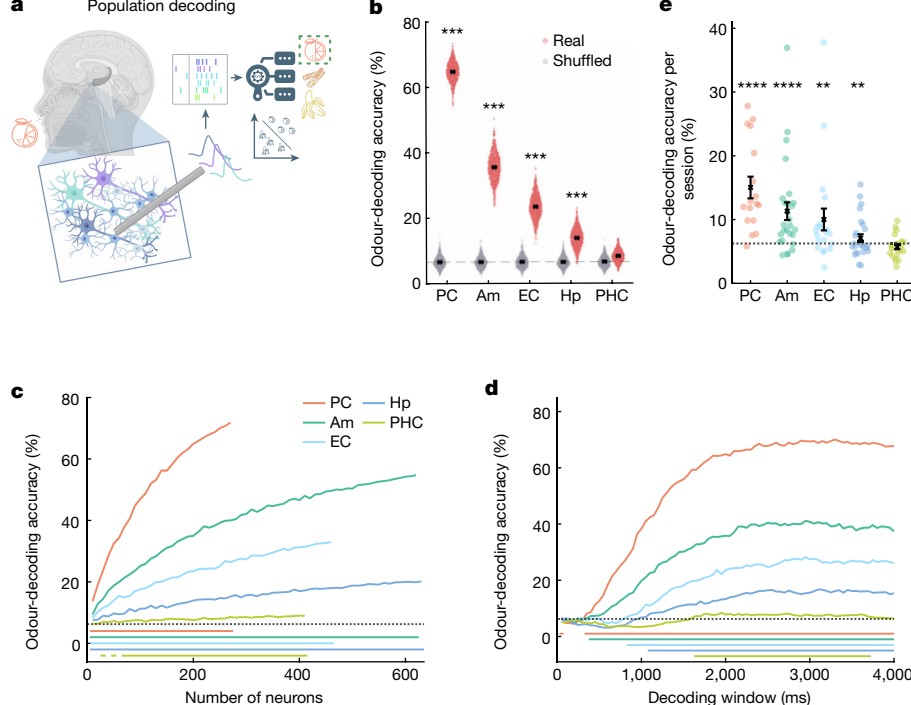

**Fig. 2 | Neuronal activity decodes odour identity. a**, Odour-identity decoding: neuronal spiking was used to train decoders to predict odour identity (here, the scent of orange). **b**, The odour-identity decoding accuracy per region. Each red dot shows the decoding performance based on 200 randomly drawn neurons (1,000 subsampling runs). The decoding performance (mean ± s.e.m.) across subsampling runs is shown in black. The grey dots indicate the decoding performance on label-permuted data. The chance level (6.25%) is indicated by the dashed horizontal line. Significance was calculated based on the percentile of mean decoding performance of the real data within the surrogate distribution (PC, $P < 0.001$; amygdala, $P < 0.001$; EC, $P < 0.001$; hippocampus, $P < 0.001$; PHC, $P = 0.16$; label permutation test with $n = 1,000$ permutations). s.e.m. margins in **b**–**d** are barely visible. **c**, Odour-identity decoding (mean ± s.e.m.) as a function of the number of neurons included (100 subsampling runs). The horizontal bars below the dashed line (chance level) indicate neuron counts with significant odour-identity decoding ($P < 0.05$, right-sided Wilcoxon signed-rank tests against chance, with Bonferroni correction for different neuron counts).

**d**, Odour-identity decoding (mean ± s.e.m.) as a function of the decoding time window beginning at odour onset (200 randomly drawn neurons, 100 subsampling runs). The horizontal bars below the dashed line (chance level) indicate the times of significant decoding performance ($P < 0.05$, right-sided Wilcoxon signed-rank test against chance, with Bonferroni correction for 80 time windows; beginning of sustained significant decoding: PC, 350 ms; amygdala, 400 ms; EC, 850 ms; hippocampus, 1,100 ms; PHC, 1,650 ms). **e**, The odour decoding performance (mean ± s.e.m., black) per recording session and region (coloured dots). Despite the limited and variable neuron counts per session, odour identity could be decoded significantly above chance (6.25%, dashed line) in the PC, amygdala, EC and hippocampus (PC, 14 out of $n = 17$ sessions showed significant decoding compared to 1,000 odour-label-permuted data, $P < 10^{-10}$; amygdala, 13 out of $n = 27$, $P = 1.3 \times 10^{-10}$; EC, 5 out of $n = 21$, $P = 0.0032$; hippocampus, 6 out of $n = 27$, $P = 0.0019$; PHC, 1 out of $n = 24$, $P = 0.71$; right-sided binomial test, $P_{chance} = 0.05$, regions with ≥2 neurons). Diagrams were created using BioRender (**a**) and Noun Project (**a**).

odour-modulated neurons in the amygdala in response to liked odours was also evident when correcting for session and participant-specific differences (Extended Data Table 1c). Using published valence ratings[39] of the standardized odours used in our study, we sought to correlate general valence ratings with the activity of odour-modulated neurons in the amygdala. Here we found a significant correlation of firing rate with valence across recordings (Fig. 4e).

## Hippocampus predicts odour identification

Successful odour identification requires odour perception, recognition and recall of the semantic odour label. The MTL has been suggested to have an essential role in these processes[40], although the underlying neuronal mechanisms remain largely unexplored. Thus, we next investigated whether single-neuron activity is linked to behavioural odour identification performance (Fig. 4f). We found that correct odour identification is accompanied by an overall increase in the firing rate of odour-modulated neurons (two-sided Wilcoxon signed-rank, $n = 406$, $Z = 2.36$, $P = 0.018$). We next investigated whether neuronal odour representations relate to behavioural identification performance. We correlated odour-decoding accuracy in each recording session and region with behavioural odour-identification

performance and found a significant positive correlation exclusively in the hippocampus (Fig. 4g). This correlation was consistently observed across the participants (Extended Data Fig. 6b). Moreover, odour identification performance at the behavioural level was correlated with higher proportions of odour-modulated neurons in the hippocampus and EC (Extended Data Fig. 6c). We next analysed whether neuronal odour representations reflected chemical odour identity (presented odours) rather than subjective perception (selected odour labels). Decoding revealed a dissociation between PC and hippocampus, with PC neurons coding preferably for chemical odour identity, and hippocampal neurons predicting perceived odour identity (Fig. 4h). Collectively, our findings reveal distinct roles of amygdala neurons in odour-valence coding and hippocampal neurons in odour identification.

## Olfactory/visual cross-modal integration

Natural environments require humans and other species to constantly integrate visual and olfactory sensory cues. How visual and olfactory information is integrated at the level of individual neurons is still unexplored in humans. We recorded from neurons along the human olfactory pathway to explore representations of congruent visual and

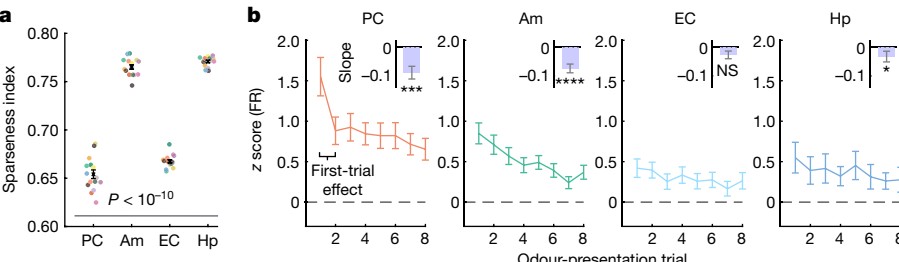

**Fig. 3 | Odour representations vary in sparseness and are suppressed by repetition. a**, The population sparseness index for each of the 15 odours across regions containing odour-modulated neurons (odours are colour coded as in Fig. 1; mean ± s.e.m. (black)). The sparseness of odour coding significantly differed across regions (one-way ANOVA, $F_{3,56} = 505$, $P < 10^{-10}$). PC exhibited a less sparse odour code than MTL regions ($P < 0.01$ for all pairwise comparisons, except for amygdala versus hippocampus, for which $P = 0.47$, after applying Tukey's honestly significant difference procedure). **b**, The average response strength (mean ± s.e.m.) of odour-modulated neurons for repeated odour presentations across regions containing odour-modulated neurons. Insets: the response slopes per region (mean ± s.e.m.). Significance was calculated based on a two-sided Wilcoxon signed-rank test against a constant response strength, that is, a slope of zero (PC, $n = 99$ neurons, $Z = -3.4$, $P = 0.00081$; amygdala, $n = 130$, $Z = -4.5$, $P = 6.9 \times 10^{-6}$; EC, $n = 74$, $Z = -1.7$, $P = 0.087$; hippocampus, $n = 74$, $Z = -2.4$, $P = 0.019$). Firing of PC neurons substantially decreased from the first to the second odour presentation (two-sided Wilcoxon signed-rank tests comparing firing rates of the first versus second trial in PC: $n = 99$ neurons, $Z = 4.9$, $P = 8.6 \times 10^{-7}$; Extended Data Fig. 7e). NS, not significant.

olfactory stimuli (Fig. 5a). For this purpose, the participants completed an additional visual task after our olfactory paradigm (20 out of 27 sessions), during which they repeatedly viewed images of objects, each of which corresponded to one of the odours in our panel (for example, orange odour and image of an orange). This enabled us to compare neuronal activity after exposure to congruent images and odours within the same population of neurons. Image-modulated neurons (Fig. 5b) were identified analogously to odour-modulated neurons. Across regions, we found more neurons modulated by odours than by images, with a significant overlap between both populations (Fig. 5c). Specifically, significantly more PC and amygdala neurons were odour modulated than image modulated, signifying their central role in odour processing. Nonetheless, a significant fraction of PC neurons was image modulated (35 neurons out of 277, $P = 5.7 \times 10^{-7}$, two-sided binomial test, $n = 277$ neurons, $P_{chance} = 0.05$), that is, they changed firing based on the image identity (Fig. 5b). The ability of PC neurons to encode odour-related image identity was further confirmed by decoding analysis (Fig. 5d and Extended Data Fig. 6d). Notably, neuronal activity in the PC predicted odour-related image identity more accurately than in any of the MTL regions, demonstrating that human PC neurons are not exclusively driven by olfaction, but also encode information from other sensory modalities. To determine whether there is a unified code for olfactory and visual stimuli, we trained a decoder on odours and tested its performance on images, and vice versa. The results showed that neuronal coding in the amygdala and PC generalizes across odours and images, suggesting cross-sensory representations in these regions (Fig. 5e,f and Extended Data Fig. 6e,f). Notably, in this analysis, identity decoding in the PC only generalized when training on odours and testing on images, and not vice versa, whereas the amygdala exhibited cross-modal coding in both cases.

In the human MTL, concept cells have been identified that respond with a high degree of invariance to representations of a specific concept (for example, a picture as well as the written and spoken name of a person or an object)[41,42]. In our recordings, we identified neurons exhibiting concept cell-like characteristics. For example, a neuron in the amygdala increased firing selectively in response to the image of a banana (Fig. 5g). Notably, the same neuron also responded to the odour of banana and the written word 'banana', indicating semantic coding in the olfactory domain. Moreover, we recorded a PC neuron selectively responding to liquorice odour, the image of liquorice and the written word 'liquorice' (Fig. 5h). Notably, this neuron also responded to a second odour, anise, an odour that is typically associated with and contained in liquorice candy. These observations in the amygdala and PC suggest semantic representations of odours at early stages of olfactory processing. Collectively, our findings reveal encoding of odour-related visual information in human PC neurons, as well as multimodal odour representations in the human amygdala and PC.

## Discussion

Despite the importance of human olfaction in health and disease, our understanding of central odour coding relies primarily on animal models and human imaging studies. Although highly informative at the macroscopic level, functional human brain imaging lacks both the spatial and temporal resolution necessary to investigate the individual neuron and circuit coding logic underlying human olfactory processing. We are therefore facing a considerable knowledge gap in human olfactory research. Here we recorded from human single neurons in PC and MTL, providing insights into olfactory processing. Across both primary and secondary olfactory areas, we identified neurons that responded to odours and altered their firing based on odour identity. We observed suppressed response strength after repeated odour presentations at prolonged intervals beyond peripheral sensory adaptation. Analysis of population sparseness revealed a more distributed code in PC compared to MTL regions. Nonetheless, neuronal activity in both the PC and MTL could accurately decode odour identity. Our findings suggest that different MTL regions mediate distinct aspects of odour processing. Amygdala neurons encode odour valence, whereas hippocampal activity predicts odour identification performance. Notably, we show that human PC neurons efficiently encode odour-related image identity. Integrating data from both odour-related visual and olfactory stimuli, we identified neurons with the ability to represent a specific stimulus concept (for example, banana) in a cross-modal manner by responding to the scent, an image and the written name of a banana. The PC and amygdala in particular engage in cross-modal coding.

## Odour coding in the human PC and MTL

Given its central role in odour processing and the distributed non-chemotopic olfactory coding in the PC proposed by animal studies[10,11,17,19,20], we hypothesized a corresponding coding logic implemented by human PC neurons. We have demonstrated that a substantial fraction of PC neurons is modulated by odour identity. Moreover, activity in few PC neurons is sufficient to accurately decode chemical odour identity. The PC exhibits a more distributed odour code compared with MTL regions, indicating an increase in sparseness of odour representations along the human olfactory pathway. Tuning profiles

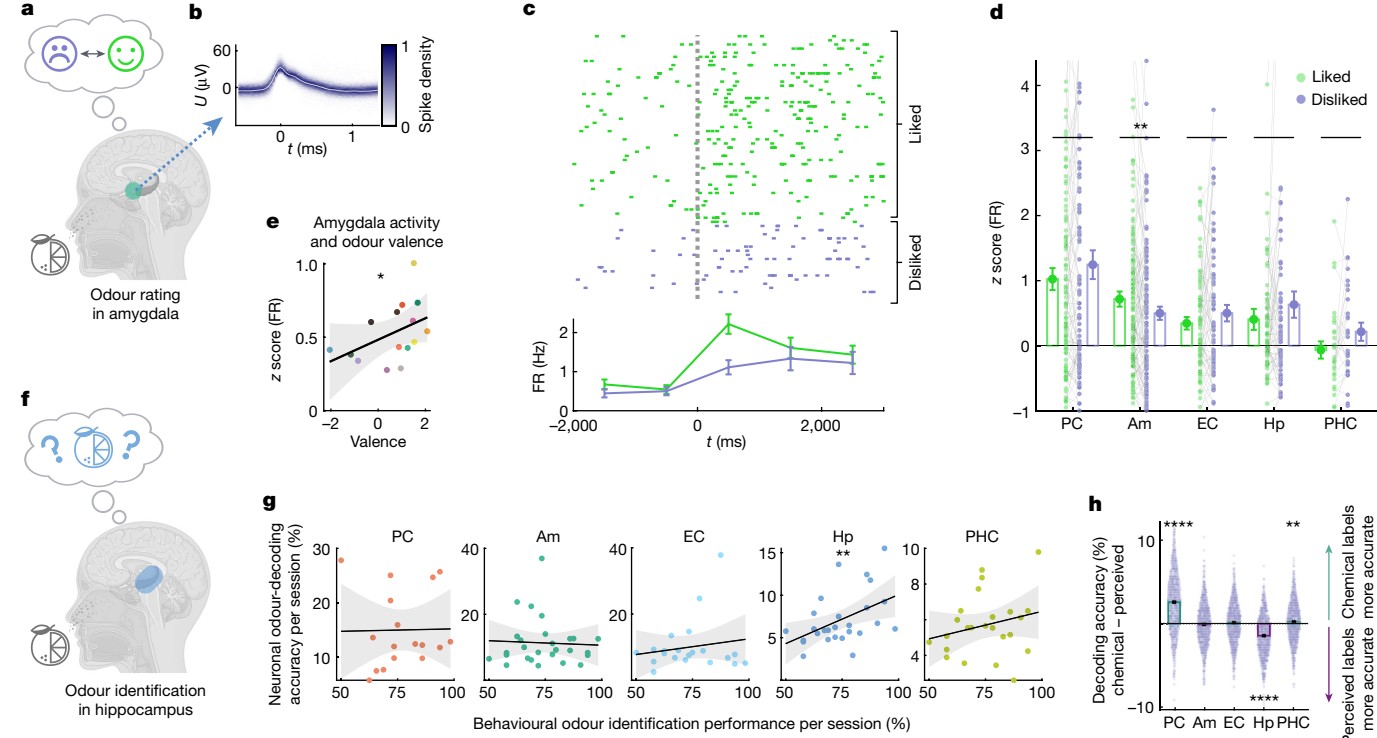

**Fig. 4 | Amygdala neurons encode odour valence and the hippocampus predicts behavioural odour-identification performance. a**, The participants rated odours as liked or disliked. **b**, Spike-shape density (mean ± s.d.) of an amygdala neuron. **c**, This neuron increased firing to liked versus disliked odours (two-sided Wilcoxon rank-sum test comparing the $z$-scored firing rates 0–2 s after odour onset; $n = 46$ versus $n = 18$, $Z = 2.1$, $P = 0.034$). Bottom, PSTH for liked and disliked odours (mean ± s.e.m., 1 s bins). **d**, Firing rates ($z$-scored, mean ± s.e.m.) of odour-modulated neurons in response to liked versus disliked odours. Only the amygdala exhibited a significant difference of subjective preference (two-sided Wilcoxon signed-rank tests; PC, $n = 99$ odour-modulated neurons, $Z = −0.76$, $P = 0.44$; amygdala, $n = 130$, $Z = 2.9$, $P = 0.004$; EC, $n = 74$, $Z = −0.37$, $P = 0.71$; hippocampus, $n = 74$, $Z = −0.61$, $P = 0.54$; PHC, $n = 29$, $Z = −1.6$, $P = 0.11$). The $y$ axis displays 95% of data. **e**, Averaged firing of odour-modulated amygdala neurons ($z$-scored) correlated with standard odour-valence ratings[39] (Spearman correlation, $n = 15$ odours, $r = 0.56$, $P = 0.03$, two-sided permutation test). This correlation was observed in a significant number of sessions (6 out of $n = 27$, $P = 0.002$) and participants (4 out of $n = 17$, $P = 0.009$, one-sided binomial test, $P_{chance} = 0.05$). Linear regressions (black) with 95% confidence

intervals (grey). **f**, Odour identification: the participants chose the odour label. **g**, Neuronal odour-decoding accuracy and behavioural odour-identification performance across regions and sessions (coloured dots). The decoding accuracy in the hippocampus was positively correlated with behavioural odour-identification performance across sessions (Spearman correlation, PC, $n = 17$ sessions, $r = 0.14$, $P = 0.59$; amygdala, $n = 27$, $r = 0.06$, $P = 0.75$; EC, $n = 21$, $r = −0.12$, $P = 0.62$; hippocampus, $n = 27$, $r = 0.50$, $P = 0.0076$; PHC, $n = 24$, $r = 0.19$, $P = 0.38$, two-sided permutation tests, regions with ≥2 neurons). Data are shown as in **e**. **h**, The difference in decoding accuracies based on chemical versus perceived (selected) odour identity. PC neurons decoded chemical odour identity more reliably, whereas hippocampal neurons predicted selected odour labels more accurately (PC, 75.8% chemical versus 22.1% perceived more accurate, $Z = 19$, $P < 10^{−10}$; amygdala: 45.6% versus 49.7%, $Z = −0.95$, $P = 0.34$; EC, 50% versus 45.2%, $Z = 1.7$, $P = 0.083$; hippocampus, 26.7% versus 69%, $Z = −16$, $P < 10^{−10}$; PHC, 52.2% versus 43.5%, $Z = 3$, $P = 0.0024$; two-sided Wilcoxon signed-rank tests across 1,000 subsampling runs). The $y$ axis displays 99% of data. Diagrams were created using BioRender (**a** and **f**) and Noun Project (**a** and **f**).

---

of PC neurons appear relatively broad as odour-modulated neurons frequently responded to several odours. Thus, similar to results from animal studies[19,20], the labelled-line organization of chemotopic information established in the human olfactory bulb is disrupted along the bulb-to-PC signalling axis.

Beyond the PC, we identified odour-modulated neurons in various MTL regions, including the amygdala, EC and hippocampus. Effective coding of odour information within these regions highlights their importance for central odour processing and formation of odour representations. Notably, decoding power decreased (and required both larger ensembles and longer integration periods) along the hierarchy of the olfactory processing pathway. Neural representations of odours emerged first in the PC and amygdala and only approximately 500 ms later in the EC. This delay in EC odour coding could support a connectivity scheme in which, in contrast to rodents[6,7], human olfactory bulb mitral cells might not directly project to the EC. Together, our findings resolve the long-standing question of whether and how individual neurons in human PC and MTL respond to odours[29], setting the stage for future studies to decipher the human olfactory code.

## Central odour repetition suppression

We are constantly surrounded by a variety of odours. Thus, detecting novel odours is of high behavioural relevance—for example, to identify potential hazards like smoke. We observed a decrement in response strength with repeated presentations (approximately every 5 min) of the same odour in the PC, amygdala and hippocampus. This temporal regime exceeds typical timescales consistent with olfactory sensory neuron adaptation[43], thereby favouring the interpretation that central habituation mechanisms are responsible for the observed response reduction. While olfactory habituation has been reported both in animals[44] and at macroscopic scale in human imaging studies[13,45], we demonstrate this phenomenon in humans at the single-neuron level. Notably, the response reduction in odour-modulated neurons qualitatively resembles that of visual responses in the human MTL[46]. The pronounced suppression observed in PC neurons between first and second stimulation is particularly marked. This may indicate high responsiveness of PC neurons to novel odours, consistent with the 'first-trial effect' observed in zebrafish[47], and could result from local inhibitory

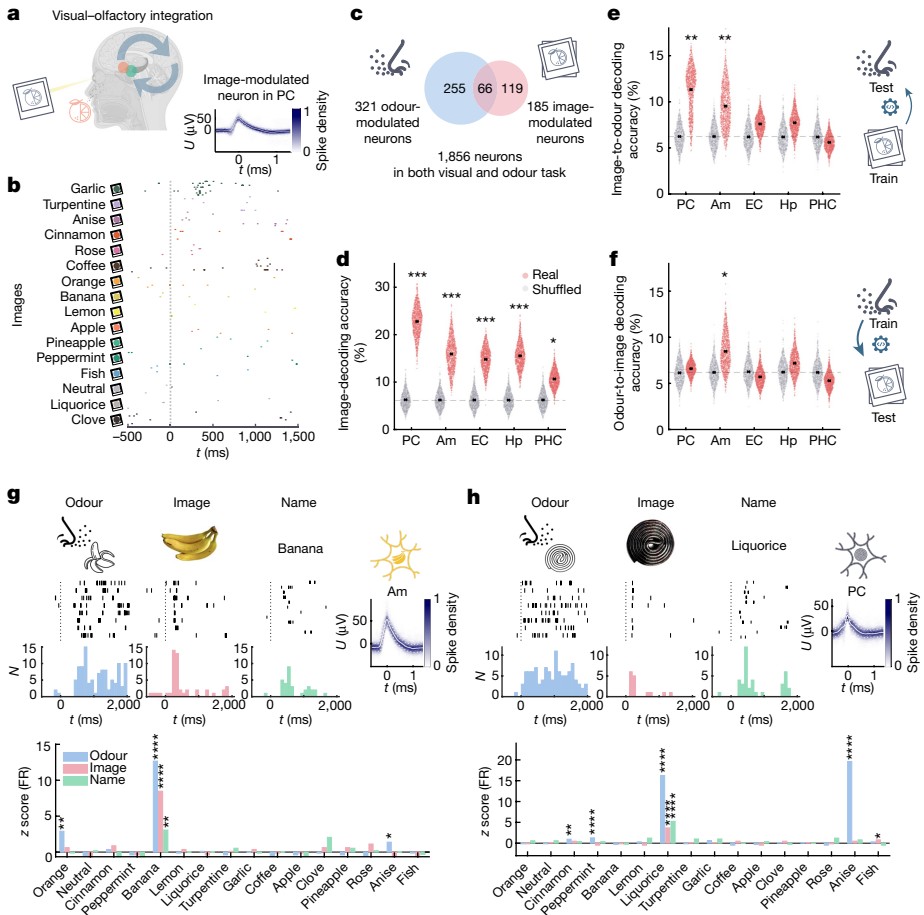

**Fig. 5 | Olfactory/visual cross-modal integration. a**, Cross-modal coding for visual and olfactory stimuli (orange scent and picture). **b**, Image-modulated PC neuron (one-way ANOVA of z-scored firing rates with image identity, $F_{15,112} = 11.98$, $P < 10^{-10}$). **c**, Population of image- and odour-modulated neurons. In total, 185 image-modulated neurons were identified ($P < 10^{-10}$, two-sided binomial test, $k = 185$, $n = 1,856$ neurons in olfactory and visual task, $P_{chance} = 0.05$). More neurons were odour modulated than image modulated (321 versus 185, two-proportion Z-test: $Z = 6.5$, $P < 10^{-10}$). Both populations showed significant overlap (66 neurons, two-sided binomial test, $P = 8.9 \times 10^{-8}$, $k = 66$, $n = 1,856$, $P_{chance} = (321/1,856) \times (185/1,856) = 0.017$). The PC and amygdala contained significantly more odour-modulated than image-modulated neurons (two-proportion Z-tests: PC, 99 versus 35 of $n = 277$ neurons, $Z = 6.3$, $P = 2.2 \times 10^{-10}$; amygdala, 99 versus 48 of $n = 479$, $Z = 4.6$, $P = 4.8 \times 10^{-6}$; EC, 36 versus 22 of $n = 301$, $Z = 1.9$, $P = 0.053$; hippocampus, 59 versus 49 of $n = 469$, $Z = 1$, $P = 0.31$; PHC: 28 versus 31 of $n = 330$, $Z = -0.41$, $P = 0.68$). **d**, The image-decoding performance based on neuronal activity was significant in all regions (statistical analysis was performed using a label permutation test with $n = 1,000$ permutations, as in Fig. 2b; for PC, amygdala, EC, hippocampus, all

$P < 0.001$; for PHC, $P = 0.018$). **e**,**f**, The decoding performance for cross-modal decoding trained on images and evaluated on odours (**d**) and vice versa (**e**) (image to odour: PC, $P = 0.002$; amygdala, $P = 0.007$; EC, $P = 0.14$; hippocampus, $P = 0.11$; PHC, $P = 0.68$; odour to image: PC, $P = 0.34$; amygdala, $P = 0.042$; EC, $P = 0.66$; hippocampus, $P = 0.22$; PHC, $P = 0.77$, label permutation test as in **d**). **g**, An amygdala neuron that increases firing in response to banana odour, a banana image and the written word 'banana' (right-sided Wilcoxon rank-sum tests, comparing the pre-odour baseline firing rates ($n = 128$, 2 s) with the firing rates after the onsets of odours ($n = 8$, 2 s), images ($n = 8$, 1 s) and non-target odour names ($n = 12$, 1 s) in the identification task; banana, $P_{odour} = 6.8 \times 10^{-8}$, $P_{image} = 1.4 \times 10^{-7}$, $P_{name} = 0.0073$; orange, $P_{odour} = 0.0029$; anise, $P_{odour} = 0.039$). **h**, A PC neuron that increases firing in response to the odour of liquorice and anise. The same neuron exhibited the most pronounced response to liquorice among images and names (liquorice, $P_{odour} = 3.2 \times 10^{-9}$, $P_{image} = 1.3 \times 10^{-6}$, $P_{name} = 5.4 \times 10^{-8}$; anise, $P_{odour} = 3.1 \times 10^{-9}$; cinnamon, $P_{odour} = 0.0014$; peppermint, $P_{odour} = 6.5 \times 10^{-5}$; fish, $P_{image} = 0.026$; statistical analysis was performed as described in **g**). Diagrams were created using BioRender (**a**) and Noun Project (**a**, **c** and **e**–**h**).

circuits specific to the PC[48]. The absence of a first-trial effect in other downstream regions indicates that olfactory information is processed in parallel and not merely relayed through the PC. The apparent lack of habituation at the earliest stages of human olfactory processing—the olfactory epithelium[43] and olfactory bulb[49]—furthermore suggests that the first-trial effect emerges predominantly at the PC level.

## Valence coding in the amygdala

On the basis of individual hedonic ratings, we demonstrated that odour-modulated amygdala neurons change firing depending on personal preferences and that amygdala firing correlates with reference odour-valence values[39]. Human imaging studies have demonstrated

amygdala activation by odours both with positive and negative valence[37,50]. However, their effects are not easily differentiated using univariate bulk-tissue-imaging methods[36], indicating local effects of odour-valence encoding[51]. Our single-neuron data suggest that odour-modulated neurons in the amygdala are involved in integrating odour identity and valence information. As positive valence was predominant in our study, future research should encompass odours that span the entire valence dimension to conclude whether our findings generalize. As both odour intensity and valence have been shown to influence the response of the human amygdala[37,51,52], future studies should also systematically vary odour intensity to investigate the interplay of valence and intensity coding. For this purpose, high-end olfactometers allowing for precise odour control will be essential.

## Hippocampal role in odour identification

Neurodegenerative diseases such as Parkinson's and Alzheimer's disease often first manifest with olfactory deficits, particularly concerning odour identification[53]. Our results link odour representations of hippocampal neurons directly with behavioural odour-identification performance, indicating that hippocampal degeneration may contribute to odour-identification deficits. Impaired behavioural odour identification performance could be a direct result of local neurodegeneration or could instead result indirectly from degeneration of upstream circuits (for example, olfactory bulb). Future research will have to explore causal contributions of odour-modulated neurons in odour identification.

## Multisensory odour representations

The PC is generally regarded as a primary olfactory area. However, with its three-layered architecture and immensely plastic recurrent connectivity, it resembles the structure of an association cortex[48,54]. Recent rodent studies have shown that neurons in the posterior PC precisely encode spatial information, suggesting a role in odour–place association[15]. Further evidence for multimodal processing of odour-related information in the PC stems from rodents[55] and human imaging studies[56,57]. Here we tested semantically coherent olfactory and visual stimuli to explore coding of PC neurons beyond olfactory perception. We identified that PC neurons decode not only odours, but also odour-related image identities. Thus, the PC not only processes olfactory stimuli, but also integrates top-down semantic information from higher cognitive areas. Notably, odour-related images were decoded more accurately in the PC than in the MTL. Future research will need to examine whether PC neurons specifically encode odour-related images, or whether they also process images of odourless objects. Our results further suggest PC involvement in multimodal, possibly even semantic integration. The lack of a specific odour-imagination task prevents us from delineating whether these multimodal representations are correlates of cross-modal integration or olfactory imagery[58]. While there is an ongoing debate how olfaction differs from other human senses, particularly with regard to olfactory imagery and the role of verbal descriptors[59,60], our findings suggest that conceptual neuronal coding schemes of olfactory information resemble those of other senses[42]. Assigning semantic odour labels is a uniquely human ability. Here we revealed that PC neurons preferably encode chemical odour identity, whereas hippocampal activity rather reflects subjectively perceived odours. This integrates well with our finding that hippocampal activity predicts behavioural odour identification, indicating that coherent internal and external odour representations facilitate semantic odour identification. While invariant responses of MTL concept neurons to visual (pictures or written text) and auditory (spoken words) stimuli have been described previously[42], chemosensory concept cells have not been identified to date. We observed neurons that generalize their response to congruent visual and olfactory stimuli. As demonstrated by cross-modal decoding analysis, amygdala neurons in particular generalize their coding between the olfactory and visual domain. Together, our findings demonstrate concept-based neuronal coding in human olfaction.

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

## Methods

### Sessions and participants

Data were collected at the Department of Epileptology at the University of Bonn Medical Center, Bonn, Germany. All of the patients in our study had drug-resistant epilepsy and underwent invasive seizure monitoring with the goal of subsequent neurosurgical resection of the seizure-generating focus. Overall, 27 sessions were recorded in 17 patients with epilepsy (12 female, 5 male; aged 22 to 60 years, mean ± s.d., 41.3 ± 11.7 years). Microwire bundles were implanted bilaterally to record single-neuron activity in the MTL and, in a subset of patients, also in the PC (17 sessions in 9 patients). All studies conformed with and were approved by the Medical Institutional Review Board of the University of Bonn, Germany (289/20). Each patient provided informed written consent.

### Human single-neuron recordings

Patients were implanted with Behnke-Fried depth electrodes (AdTech) (Extended Data Fig. 1a,b). These hollow rodlike electrodes have a diameter of 1.25 mm with 8 cylindrical clinical macroelectrodes (platinum–iridium). The innermost two macro contacts are spaced 3 mm apart, while the remaining contacts are equidistantly spaced. Through each electrode, a bundle of platinum–iridium microwires with a diameter of 40 μm was inserted. Each bundle contained eight insulated high-impedance (typically 200–500 kΩ)[61] recording wires and one low-impedance reference wire without insulation. Electrodes were implanted using a rigid stereotactic frame (Leksell, Elekta) with an orthogonal guide tube[62]. Electrode target locations were determined by clinical criteria and differed minimally within target regions across patients. This, along with the technical limitation of precisely localizing microwire positions, precluded us from targeting specific subregions, for example, individual subnuclei of the amygdala or specific hippocampal subfields. Electrode placement was controlled by intraoperative CT scans co-registering the head-fixed frame to pre-operative MRI planning scans. After skin incision at the electrode entry point, a hole for an anchor bolt was drilled, and the anchor bolt was screwed into the skull using the guide tube. Microwire bundles were preloaded into the macroelectrodes and trimmed by a single cut with either a scalpel or surgical scissors on a back table in the operation room, such that they protruded from the tip of the clinical electrode by 3 to 5 mm. Extended Data Fig. 1c displays SEM images of uncut and cut microwires for comparison. After preparation, microwire bundles were replaced by a guiding rod for implantation. After the insertion of the macroelectrode into its target position, the guiding rod was retracted and the microwire bundle was carefully inserted to avoid kinking or bending[62]. Local field potentials containing single-neuron activity were sampled at 32,768 Hz, band-pass filtered between 0.1 and 9,000 Hz, and amplified by a 256-channel ATLAS amplifier (Neuralynx) using Pegasus (v.2.1.1, Neuralynx). Spike extraction and sorting were performed using Combinato[63]. Spikes of negative voltage deflection were extracted and analysed. For illustration, spikes are depicted with inverted polarity. Automated artifact removal based on the DER algorithm[64] was applied to all sessions. Clustering of each channel was manually validated by an experienced rater, and artifacts were removed. As Combinato (used with the default parameters in this study) tends to overcluster the recorded unit data in automated mode, we manually merged clusters on the basis of their waveforms, cross correlograms and other firing characteristics. Single-unit recording quality and spike sorting was validated based on inter-spike-interval (ISI) violations, spike amplitudes and spike peak signal-to-noise (SNR), as well as cluster isolation distance (Extended Data Fig. 8). Electrode localization was performed based on co-registered CTs and MRIs using the LeGUI software package (v.1.2)[65] and electrode locations were visualized using Fieldtrip (v.213bc8bcb)[66] and the 'plot_ecog' function (https://github.com/s-michelmann/moment-by-moment-tracking/blob/master/plot_ecog.m). A total of 2,416 units was recorded (1,292 single units (SU)): 622 units (348 SU) in the amygdala, 464 units (256 SU) in the EC, 634 units (341 SU) in the hippocampus, 419 units (199 SU) in the PHC and 277 units (148 SU) in the PC.

### Odour stimuli and delivery protocol

As odour stimuli, we used standard pen-like Sniffin' Sticks from the Identification-16 test (Burghart Messtechnik). The participants sat in bed with a laptop on a tray in front of them while they were presented with 15 different odour stimuli, administered eight times in pseudorandom order. The pen containing leather was replaced by a blank odourless pen that served as control (26 of 27 recordings). Odour pens were presented approximately 2 cm below the nose, centred between both nostrils. The patients were verbally instructed on each trial to inhale on command ("Please inhale NOW!"). To ensure consistent odour sampling across trials, the participants were asked to inhale only once for each odour presentation and not sniff at their convenience. Odour pens were immediately removed after the first inhalation. This experimental protocol was devised to minimize odour-specific respiratory variability. The experimenter's (M.S.K.) direct supervision ensured adherence to the instructions throughout the experiment. Pens were opened only immediately before odour exposure. Simultaneous with the inhale command, the presentation time was logged and an odour was administered. In 13 out of 27 recording sessions, respiration was measured using thoracic and abdominal plethysmography belts (Extended Data Fig. 5a; SleepSense, Scientific Laboratory Products). Data from both belts were averaged and analysed using the Breathmetrics toolbox[67] (v.2.0, human respiratory belt default settings with sliding baseline correction). In the remaining 14 sessions, respiration belts could not by applied due to patient discomfort or noisy interference with the microwire recordings. Overall, the participants complied accurately with the experimental protocol, inhaling once during odour exposure and well timed to odour delivery (Fig. 1d and Extended Data Fig. 5c). Bilateral measurements of nasal airflow will allow future studies to precisely examine the interactions of neuronal activity and local oscillatory dynamics across the ipsilateral and contralateral hemispheres at a high temporal resolution. Standardized pen-like odour stimuli lack millisecond precision and exact control of odour concentrations that can be achieved with high-end olfactometers. However, this odour-delivery method proved to be both efficient and effective for presenting a wide range of odour stimuli in the clinical environment.

### Paradigm

During the first four presentation cycles, the patients were asked to rate whether they liked or disliked the odour (forced choice; Fig. 1e). In 64.8 ± 2.0% of trials, the participants reported to like the odour. Although liking and valence have been differentiated in some contexts[68], we use the term valence as a multifaceted concept that includes liking[69]. In the subsequent four presentation cycles, odours were to be identified by choosing the correct odour name out of four options (Fig. 1e). Written odour names (labels) were selected pseudorandomly from a list of the 15 odour stimuli plus the neutral, odourless control. Each odour label was used 4 times as the correct and 12 times as an incorrect choice option. Name options were sequentially added at 1 s intervals, allowing stimulus-specific assessment of neuronal activity to individual written odour-associated words (Fig. 5g,h). To avoid confounding cueing effects induced by previous presentation of semantically matching odours, we excluded trials from the analysis in which the odour word was the correct choice option (Fig. 5g,h). The participants identified the correct odour in 74.1 ± 1.5% of cases. The mean presentation time of odours was 2.31 ± 0.13 s, the mean inter-odour interval was 19.4 ± 0.4 s, with the same odours repeated on average every 5.18 ± 0.05 min. In 20 out of 27 recordings, immediately after the olfactory task, we additionally presented 16 pictures, each semantically corresponding to one of the odours, including a light grey screen to match the odourless control. Each picture was presented for 1 s, 8 times,

in pseudorandom order. This protocol enabled us to identify neurons responding to images that were semantically congruent to the odours presented in this study. The experimental tasks were implemented using MATLAB R2019a (MathWorks) and Psychtoolbox3[70–72].

## Statistics

All statistical analyses were conducted in MATLAB 2021a. Unless otherwise stated, nonparametric and two-sided statistical tests were applied with a $P$ value below an $\alpha$-level of 0.05 considered to be significant. The arithmetic mean was used to compute averages, and the error bars represent the s.e.m. or the s.d. as specified. Spearman's rank-order correlations were used for all correlational analyses with $P$ values estimated using MATLAB's 'corr' function. ANOVA was performed to determine significant differences between multiple groups using Tukey's honestly significant difference procedure to correct for multiple pairwise comparisons. The box plots in Fig. 1f were generated based on the built-in MATLAB function 'boxplot'; the central lines indicate the median, the box limits show the 25th and 75th percentiles, and the whiskers extend from the minimal to maximal values that are not considered outliers, which were defined by exceeding 1.5× the interquartile range. Statistical significance is indicated by asterisks in figures. Custom MATLAB codes were used to calculate binomial tests and to generate Venn diagrams (MATLAB Central File Exchange, M. Nelson 2023, v.2.0, https://www.mathworks.com/matlabcentral/fileexchange/248 13-mybinomtest-s-n-p-sided; Darik 2023, v.1.7, https://www.mathworks.com/matlabcentral/fileexchange/22282-venn).

## Odour-modulated neurons

To identify odour-modulated neurons, we first calculated a $z$ value for the firing rate during a response interval ([0, 2 s] after odour onset compared to [−5, 0 s] before odour onset) and performed a one-way ANOVA for odour identity. Neurons with a significant effect of odour identity across all 128 trials ($P < 0.05$) were termed odour-modulated neurons. Normalized PSTHs (Fig. 1j) were calculated by binning the spiking of each neuron (50 ms bins) and $z$-scoring all bins using the bins in the [−5, 0 s] baseline window before odour onset.

## Image-modulated neurons

In analogy to our definition of odour-modulated neurons, we identified image-modulated neurons based on a one-way ANOVA of the $z$-scored firing rates for image identity ([0, 1 s] after image onset compared to [−0.5, 0 s] before the image onset[27,73,74]). Neurons with a significant effect of image identity across all 128 trials ($P < 0.05$) were termed image-modulated neurons.

## Decoding analysis

All decoding analyses were performed using the Neural Decoding Toolbox[33] (v.1.0.4). In each region, spiking data were first binned within a [0, 2 s] time window after odour onset and a [0, 1 s] time window after image onset. We trained a maximum-correlation-coefficient classifier to predict odour or image identity, using 8 cross-validation data splits and 10 resample runs. To compare decoding performance across regions, an equal number of neurons ($n$ = 200) was subsampled in each decoding analysis. The decoding was repeated 1,000 times on random subsamples. Significance levels were estimated based on a surrogate distribution derived from decoding analysis on label-permuted data ($n_{perm}$ = 1,000). The percentile of the actual data mean within the surrogate distribution was used to estimate $P$ values. To evaluate the impact of the decoding time window (Fig. 2d), we repeated the decoding analysis, systematically varying the decoding time interval ranging from 50 ms up to 4,000 ms, with 50 ms increments and 100 subsampling runs. Moreover, we systematically varied the number of neurons included in the decoding analysis, starting with 10 neurons and increasing in steps of 10 (Fig. 2c). For cross-modal decoding, we trained the classifier on the image trials and tested it on the odour

trials (Fig. 5e) and vice versa (Fig. 5f) using the [0, 2 s] decoding time window. To ensure that our decoding results were not driven by systematic differences of the first compared to later trials, we repeated the decoding without the first trial and obtained overall consistent findings (Extended Data Fig. 9). In the population decoding, equal numbers of neurons are randomly sampled across recording sessions, enabling a balanced comparison of performance between regions irrespective of individual variations in neuronal yield. Comparing decoding performance of randomly sampled neurons within and across recording sessions yielded consistent results (Extended Data Fig. 9g), indicating that population decoding extrapolates well to larger populations of neurons. The odour-decoding performance for each session was estimated based on all recorded neurons per region with a minimum of 2 neurons, using all odour presentations, 8 cross-validation data splits and 1,000 resample runs. For each session and region, a surrogate distribution was estimated by repeating the decoding analysis 1,000 times on odour-label-permuted data, using 10 resample runs each. The percentile of the actual decoding performance within this surrogate distribution was used to estimate $P$ values. Decoding performances per participant were evaluated by averaging decoding performances across repeated sessions within anatomical target regions. To test whether neural activity predicted chemical odour identity better than perceived odour identity (that is, sometimes falsely selected odour labels), we used a decoding analysis during the odour-identification task (4 trials per odour). An equal number of neurons was randomly subsampled from recordings in which each odour was chosen at least twice. In each anatomical target region, 100 neurons were randomly subsampled 1,000 times, and a decoder was trained using two cross-validation data splits and ten resample runs. Decoders were trained based both on chemical odour identity and perceived odour identity (selected odour label) using the same neuronal populations. The differences between the two decoding accuracies were used to assess which labels were predicted more accurately by neuronal firing.

## Estimation of population sparseness

A widely used measure of population sparseness is the activity ratio $A_k$, defined as[75,76]

$$A_k = \frac{\left(\frac{1}{N}\sum_{i=1}^{N} x_i\right)^2}{\frac{1}{N}\sum_{i=1}^{N} x_i^2}$$

where $x_i$ is the mean response activity of the $i$th neuron to the stimulus $k$, and $N$ is the number of neurons. The overall sparseness of the population to a set of different stimuli was estimated by averaging across stimuli. We use the sparseness index $SI_k = (1 - A_k)/(1 - 1/N)$ to obtain a measure of sparseness ranging from 0 to 1, with higher values corresponding to a sparser code[34].

## Olfactory repetition suppression

Each odour was presented eight times. For each odour-modulated neuron, we calculated the mean $z$-scored firing rate for each odour presentation, resulting in eight firing-rate values per neuron. We then performed a linear regression for each neuron (firing rates versus odour presentation) and used the resulting slopes as a measure of change in the firing rate, following previous studies[46]. Slopes were calculated for each region and compared with a constant response strength (that is, a slope of 0) using a Wilcoxon signed-rank test.

## Mixed-effects models

Generalized linear mixed-effects models (GLMMs) were used to control for recordings across multiple sessions within and across participants. A GLMM was used for each fixed effect to predict trial-wise spike counts of odour-modulated neurons using MATLAB's 'fitglme' function. Brain regions and interactions were incorporated as fixed

effects. Participant identity and recording session per participant were included as random effects to account for their nested hierarchical nature[77]. Each fixed-effects regressor was incorporated as a random slope for both participant identity and participant-session nesting, and neuron identity was included with an individual intersect to account for participant–session–neuron nesting[78]. All random effects comprised an individual intersect. Likelihood ratio tests (MATLAB's 'compare' function) confirmed that the full models that we used with both random slopes and intercepts outperformed models incorporating only random intercepts. Poisson models were fitted based on the restricted maximum pseudo likelihood with a logarithmic link function.

## SEM analysis of microwires

For SEM analysis, two microwires from a new bundle were used. One microwire was trimmed using a scalpel, while the other remained uncut. For imaging, wires were shortened to approximately 8 mm in length and mounted onto aluminium stubs using conductive carbon tape. The samples were then sputter-coated with 15 nm of gold using a Quorum 150 R ES coating unit (Quorum Technologies) and imaged using the Everhart–Thornley secondary electron detector in a Zeiss Sigma 300 (Zeiss) Field Emission Gun SEM operated at 2 kV. In total, five images of two scalpel-trimmed microwires and four images of two untrimmed microwires were obtained.

## Reporting summary

Further information on research design is available in the Nature Portfolio Reporting Summary linked to this article.

## Data availability

Data supporting the central findings of this study and needed to reproduce the main figures in this manuscript are publicly available at GitHub (https://github.com/marcelkehl/HumanOdorRepresentations). Reference valence ratings of the standardized odours in our study were reported previously[39] (Fig. 4e).

## Code availability

Codes used to generate the main figures and reproduce the central results of this study are publicly available at GitHub (https://github.com/marcelkehl/HumanOdorRepresentations).

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

**Acknowledgements** We thank all of the patients for their participation; A. H. D. Mostafa for help with spike-sorting; and E. Johnson at the Dunn School EM Facility at Oxford for assistance in obtaining the SEM images. This research was supported by grants from the DFG (MO 930/4-2, MO 930/15-1, SFB 1089, SPP 2205, SPP 2411), BMBF (031L0197B) (F.M.) and a NRW Network Grant (iBehave) (F.M. and M. Spehr). Figures 1a, 2a, 4a,f and 5a and Extended Data Fig. 5a were created using BioRender. Icons from the Noun Project (https://thenounproject.com) were obtained under a NounPro subscription.

**Author contributions** M.S.K. and F.M. designed the study. M.S.K. implemented the study. R.S. and F.M. recruited patients. V.B., M. Schneider and F.M. implanted the electrodes. M.S.K., F.M. and M. Spehr planned and designed the analyses with contributions from K.O.; M.S.K. performed all experiments. M.S.K collected and analysed all data. S.M. validated the code. M.S.K., M. Spehr and F.M. wrote the paper with contributions from K.O. and S.M. All of the authors discussed the results and commented on the manuscript.

**Competing interests** K.O. is currently employed by dsm-firmenich. The company had no influence on the study design or interpretation of the results and did not provide any financial support. The other authors declare no competing interests.

**Additional information**
**Correspondence and requests for materials** should be addressed to Marc Spehr or Florian Mormann.

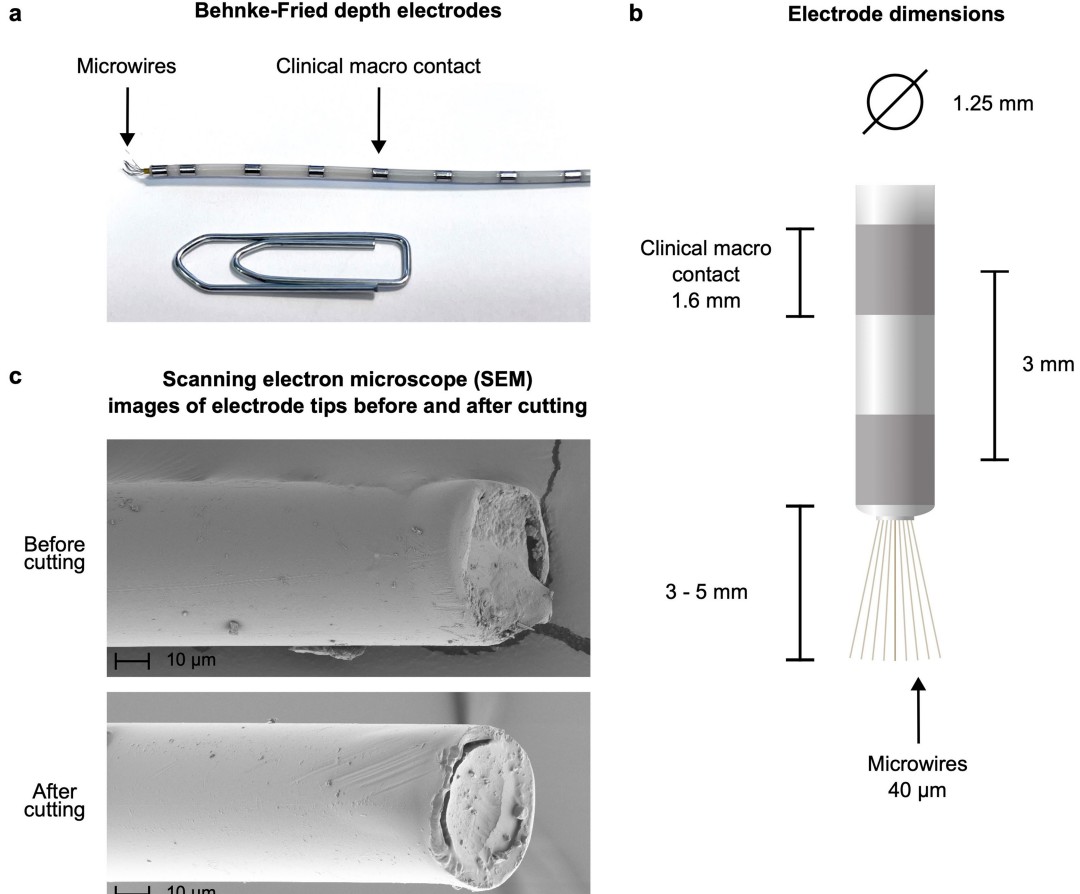

**a** Behnke-Fried depth electrodes

Microwires Clinical macro contact

**b** Electrode dimensions

⌀ 1.25 mm

Clinical macro contact 1.6 mm

3 mm

3 - 5 mm

Microwires 40 µm

**c** Scanning electron microscope (SEM) images of electrode tips before and after cutting

Before cutting

10 µm

After cutting

10 µm

**Extended Data Fig. 1 | Characteristics of Behnke-Fried depth electrodes used for single-neuron recordings in the human PC and MTL. a**, Behnke-Fried depth electrode. Microwires inserted through the shaft of the hollow clinical macro electrode protrude from the tip of the electrode. The electrode features eight cylindrical clinical platin-iridium contacts. The two innermost contacts are 3 mm apart, while the remaining contacts are equidistantly spaced along the electrode. **b**, Illustration of the electrode geometry and dimensions. **c**, Scanning electron microscopy images of the tip of a microwire before (top) and after cutting (bottom).

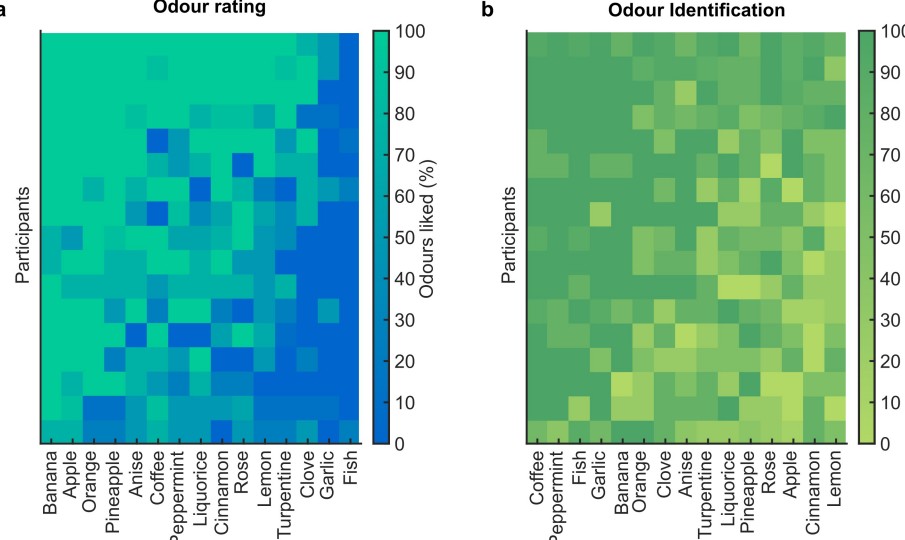

**Extended Data Fig. 2 | Odour valence ratings and identification performance for each participant. a**, Mean odour ratings for each participant and odour. Odours are sorted from most to least liked (left to right) and participants are organized by average valence ratings (top to bottom). **b**, Average behavioural odour identification performance for each participant and odour. Odours are sorted from most to the least accurately identified (left to right) and participants are organized by their mean identification performance (top to bottom).

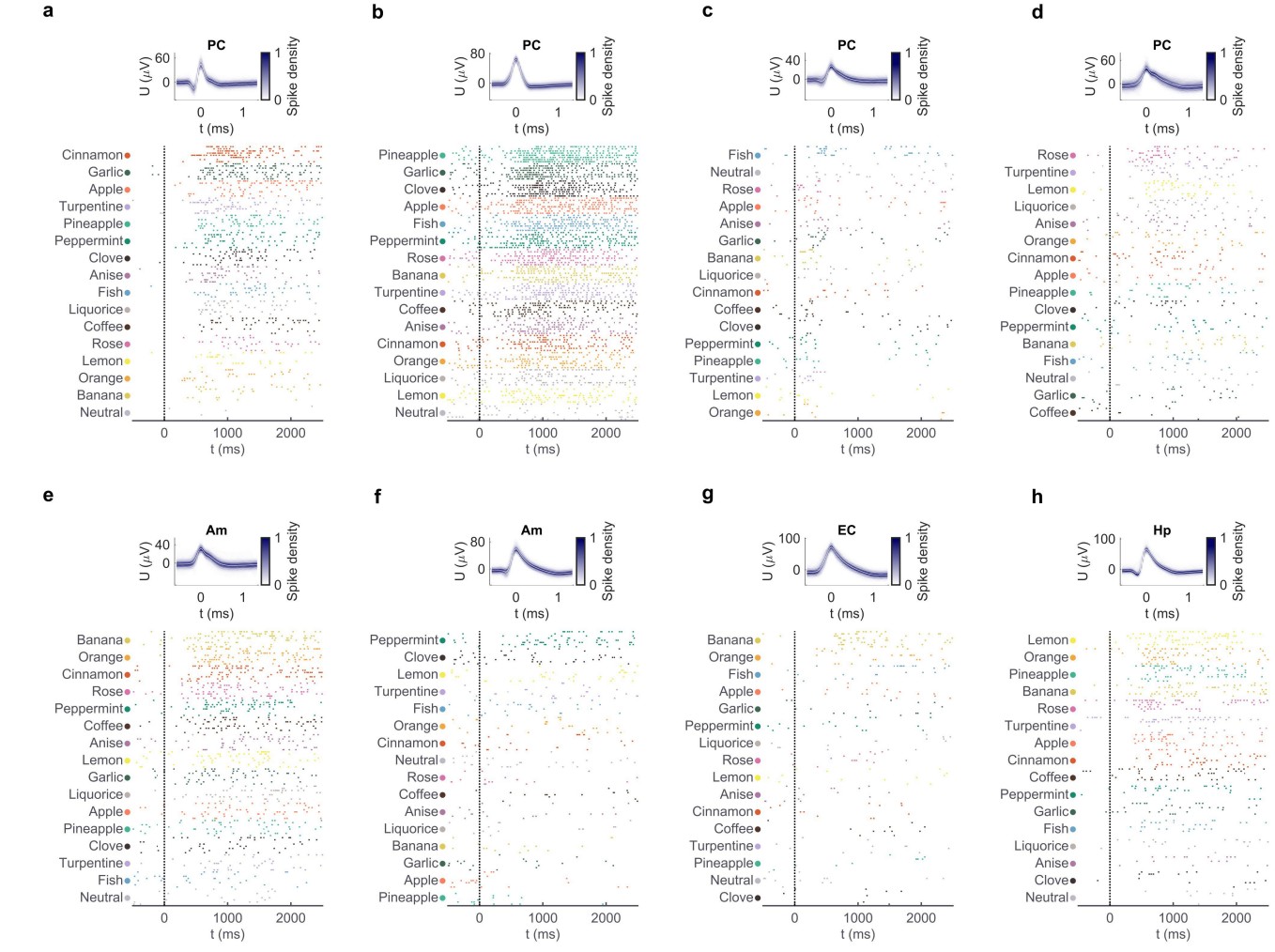

**Extended Data Fig. 3 | Odour-modulated neurons in the PC and MTL.**
Examples of odour-modulated neurons across recording sites exhibiting significantly different firing rates in response to distinct odours (one-way ANOVA of z-scored firing rates with odour identity, $n = 128$ trials). Spike shape density plots of each neuron are shown in the top of each panel, with mean ± s.d. in white. **a**, PC neuron: $F_{15,112} = 9.4$, $P < 10^{-10}$; **b**, PC neuron: $F_{15,112} = 9.8$, $P < 10^{-10}$; **c**, PC neuron: $F_{15,112} = 2.0$, $P = 0.02$; **d**, PC neuron: $F_{15,112} = 4.0$, $P = 7.8 \cdot 10^{-6}$; **e**, amygdala neuron: $F_{15,112} = 5.5$; $P = 3.0 \cdot 10^{-8}$; **f**, amygdala neuron: $F_{15,112} = 5.7$, $P = 1.4 \cdot 10^{-8}$; **g**, EC neuron: $F_{15,112} = 6.8$; $P = 3.1 \cdot 10^{-10}$; **h**, hippocampus neuron: $F_{15,112} = 3.4$; $P = 0.00012$. PC, piriform cortex; Am, amygdala; EC, entorhinal cortex; Hp, hippocampus.

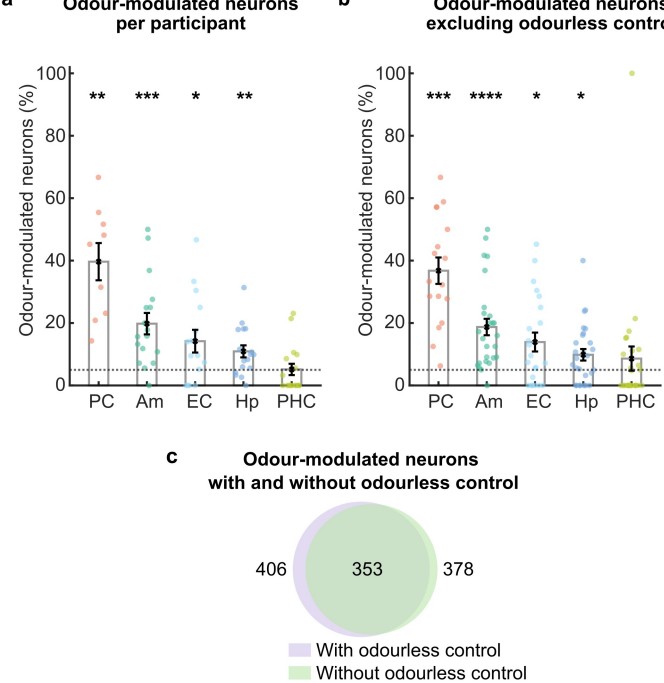

**a** Odour-modulated neurons per participant

**b** Odour-modulated neurons excluding odourless control

**c** Odour-modulated neurons with and without odourless control

406 | 353 | 378

With odourless control
Without odourless control

**Extended Data Fig. 4 | Odour-modulated neurons are reliably identified across participants and without odourless controls. a**, Same as Fig. 1h, but averaged across recording sessions per participant. Proportions of odour-modulated neurons (mean ± s.e.m.) across regions for each participant. Significant proportions of odour-modulated neurons were found in PC, amygdala, EC and hippocampus across participants (PC: 39.7 ± 6%, $n = 9$ participants, $P = 0.002$; amygdala: 19.8 ± 3.4%, $n = 17$, $P = 0.00033$; EC: 14.2 ± 3.6%, $n = 15$, $P = 0.027$; hippocampus: 10.9 ± 1.9%, $n = 17$, $P = 0.0043$; PHC: 5.14 ± 1.8%, $n = 17$, $P = 0.79$; one-sided Wilcoxon signed-rank against chance). Chance level (5%) indicated by the horizontal dashed line (see also Fig. 1h). **b**, Same as Fig. 1h, but excluding the odourless control. Distribution of odour-modulated neurons after omitting the neutral odour stimuli for the definition of odour-modulated neurons (PC: 36.8 ± 4.2%, $n = 17$ sessions, $P = 0.00016$; amygdala: 18.7 ± 2.6%, $n = 27$, $P = 1.2 \cdot 10^{-5}$; EC: 13.9 ± 3%, $n = 22$, $P = 0.017$; hippocampus: 9.83 ± 1.8%, $n = 27$, $P = 0.011$; PHC: 8.61 ± 3.9%, $n = 26$, $P = 0.42$; one-sided Wilcoxon signed-rank against chance). **c**, Population of odour-modulated neurons identified with and without the odourless control showed a highly significant overlap ($P < 10^{-10}$ in a two-sided binomial test with $k = 353$, $n = 2,416$ neurons and $P_{chance} = (406/2,416) \cdot (378/2,416)$). ****$P < 0.0001$, ***$P < 0.001$, **$P < 0.01$, *$P < 0.05$.

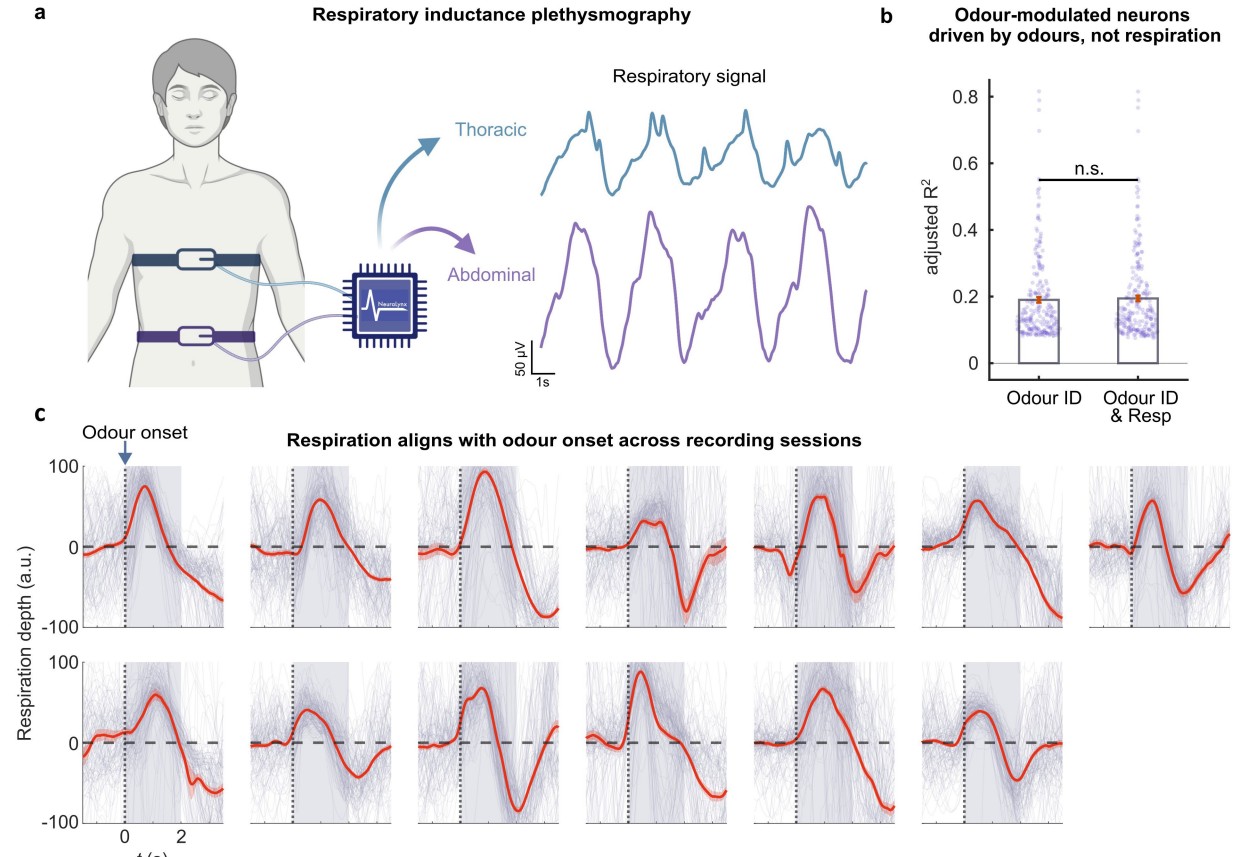

**a** Respiratory inductance plethysmography

Respiratory signal

Thoracic

Abdominal

**b** Odour-modulated neurons driven by odours, not respiration

n.s.

Odour ID

Odour ID & Resp

**c** Odour onset

Respiration aligns with odour onset across recording sessions

Respiration depth (a.u.)

*t* (s)

**Extended Data Fig. 5 | Odour identity, not respiration, drives odour-modulated neurons. a**, Respiration was measured with thoracic (upper, turquoise) and abdominal (lower, lilac) inductive plethysmography belts. Respiration signals were amplified and recorded using the Neuralynx ATLAS system, ensuring reliable temporal synchronization with neural recordings. **b**, Performance (adjusted $R^2$) of linear regression models, predicting neuronal firing (z-scores) based on odour identity, or odour identity combined with respiration (inhalation depth). Adding respiratory information to odour identity did not significantly improve the model predictions of firing rates of odour-modulated neurons (odour identity & respiration ($R^2 = 0.194 \pm 0.008$) versus odour identity alone ($R^2 = 0.190 \pm 0.008$), $n = 240$ odour-modulated neurons with respiratory recordings, $Z = 0.85$, $P = 0.39$ two-sided Wilcoxon signed-rank). Thus, odour-modulated neurons are primarily driven by odour-specific differences and not variations in respiration. **c**, Averaged odour-locked respiratory signals for each individual recording session (mean ± s.e.m., 13 sessions with $n = 128$ trials each). Participants consistently inhaled once (single peak) during the first 2 seconds after odour onset (grey shaded area), the analysis time window used for identification of odour-modulated neurons. n.s. = not significant. Diagrams were created using BioRender (**a**) and Noun Project (**a**).

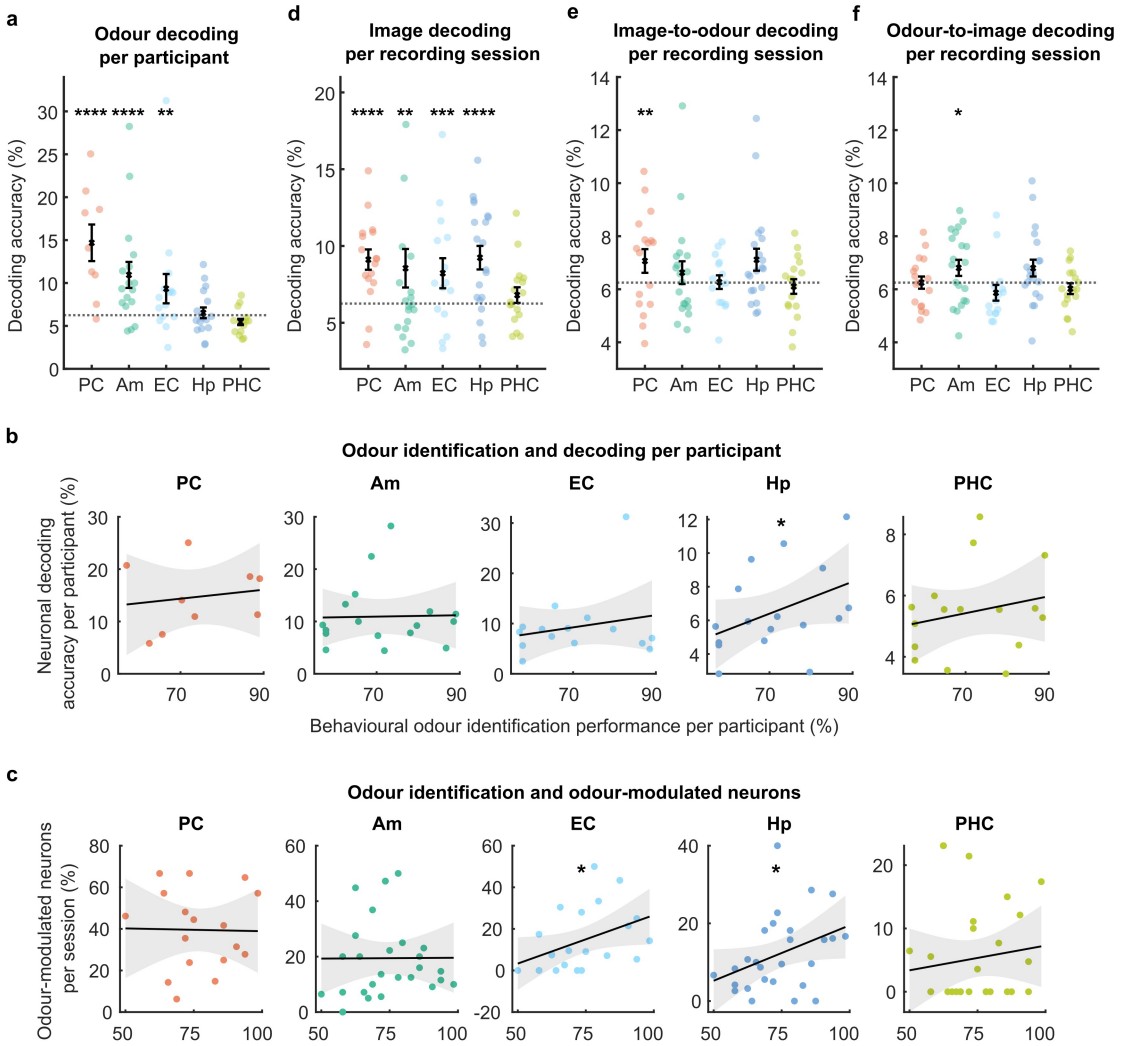

**Extended Data Fig. 6 | Decoding across individual recording sessions and participants. a**, Odour-decoding performance per participant and region. Averaging the decoding performance across all sessions per participant (mean ± s.e.m., black) demonstrated significant odour identity decoding in PC, amygdala, and EC (PC: 7 out of $n$ = 9 participants, $P$ = 2.6·10$^{-8}$; amygdala: 10 of $n$ = 17, $P$ = 1.4·10$^{-9}$; EC: 4 of $n$ = 15, $P$ = 0.0055; hippocampus: 3 of $n$ = 17, $P$ = 0.050; PHC: 0 of $n$ = 16, $P$ = 1; right-sided binomial test with $P_{chance}$ = 0.05). See also Fig. 2b. **b**, Odour-decoding accuracy and behavioural odour-identification performance across regions and participants, averaged across sessions for each participant (coloured dots). Decoding accuracy in the hippocampus positively correlated with odour-identification performance across participants (Spearman correlation, PC: $n$ = 9 participants, $r$ = 0.15, $P$ = 0.71; amygdala: $n$ = 17, $r$ = 0.10, $P$ = 0.71; EC: $n$ = 15, $r$ = 0.01, $P$ = 0.96; hippocampus: $n$ = 17, $r$ = 0.50, $P$ = 0.043; PHC: $n$ = 16, $r$ = 0.15, $P$ = 0.58, two-sided permutation test). Linear regressions (black) with 95%-confidence intervals (grey). **c**, Odour identification improves with more odour-modulated neurons in the hippocampus and EC. Percentage of odour-modulated neurons and performance for each recording session for different regions. Percentage of odour-modulated neurons in the EC and hippocampus is positively correlated with individual performance in the odour identification task (Spearman correlation, PC: $n$ = 17 sessions, $r$ = −0.04, $P$ = 0.89; amygdala: $n$ = 27, $r$ = 0.15, $P$ = 0.44; EC: $n$ = 22, $r$ = 0.49,

$P$ = 0.022; hippocampus: $n$ = 27, $r$ = 0.38, $P$ = 0.049; PHC: $n$ = 26, $r$ = 0.15, $P$ = 0.47, two-sided permutation test). Linear regressions (black) with 95%-confidence intervals (grey). **d**, Image-decoding accuracy (mean ± s.e.m., black) per recording session and region (coloured dots). Despite the limited and variable neuron count per session, image identity could be decoded significantly above chance (6.25%, dashed horizontal line) across sessions in PC, amygdala, EC, and hippocampus (PC: 7 out of $n$ = 17 sessions showed significant decoding compared to 1,000 image-label-permuted data, $P$ = 9.7·10$^{-6}$; amygdala: 5 out of $n$ = 20, $P$ = 0.0026; EC: 5 out of $n$ = 15, $P$ = 0.00061; hippocampus: 10 out of $n$ = 20, $P$ = 1.1·10$^{-8}$; PHC: 2 out of $n$ = 17, $P$ = 0.21; right-sided binomial test with $P_{chance}$ = 0.05, regions with ≥ 2 neurons in recordings with both olfactory and visual task). **e-f**, Cross-modal decoding per session trained on images and evaluated on odours (e), and vice versa (f), revealed significant cross-modal coding in PC and amygdala (Image-to-odour: PC: 4 out of $n$ = 17 sessions, $P$ = 0.0088; amygdala: 2 out of $n$ = 20, $P$ = 0.26; EC: 0 out of $n$ = 15, $P$ = 1; hippocampus: 2 out of $n$ = 20, $P$ = 0.26; PHC: 0 out of $n$ = 17, $P$ = 1; Odour-to-image: PC: 0 out of $n$ = 17, $P$ = 1; amygdala: 4 out of $n$ = 20, $P$ = 0.016; EC: 1 out of $n$ = 15, $P$ = 0.54; hippocampus: 2 out of $n$ = 20, $P$ = 0.26; PHC: 0 out of $n$ = 17, $P$ = 1; right-sided binomial test with $P_{chance}$ = 0.05, regions with ≥ 2 neurons in recordings with both olfactory and visual task as in (d)). ****$P$ < 0.0001, ***$P$ < 0.001, **$P$ < 0.01, *$P$ < 0.05.

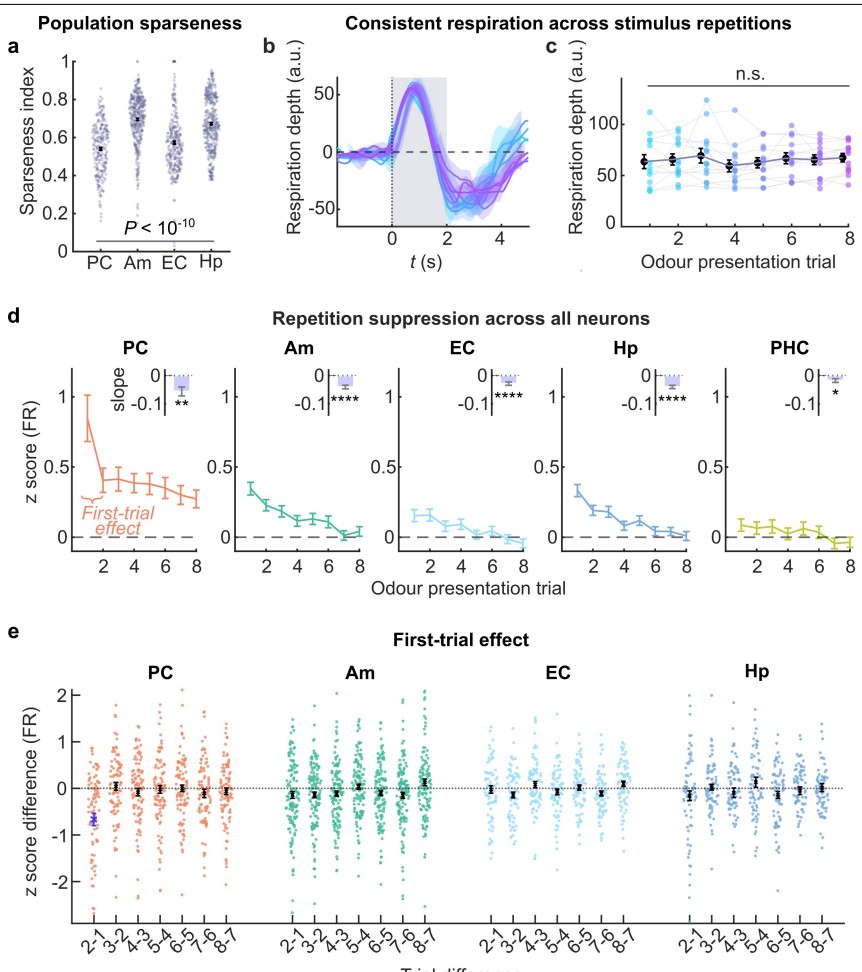

**Extended Data Fig. 7 | Population sparseness of odours per recording session and repetition suppression across all neurons. a**, Population sparseness index in response to odours for each recording session and odour (mean ± s.e.m. in black). Sparseness significantly differed across recording sites (one-way ANOVA, $F_{3,1367} = 90.1, P < 10^{-10}$). All pairwise tests significant ($P < 0.05$) following Tukey's honestly significant difference procedure, except the pairwise comparison of amygdala and hippocampus ($P = 0.11$). **b**, Average respiratory traces (mean ± s.e.m.) for each odour presentation (trials 1 to 8) across 13 recording sessions. **c**, Averaged inhalation depth (mean ± s.e.m., black) for each odour presentation (1 to 8) and recording session (coloured dots). Inhalation depth was consistent across odour repetitions (one-way ANOVA, $F_{7,96} = 0.3, P = 0.95, n = 13$ recording sessions with 8 trials each). **d**, Average response strength for repeated odour presentations across all recorded neurons in each anatomical region (mean ± s.e.m.). Odour repetitions are approximately 5 min apart. Insets depict the mean response slopes per region (mean ± s.e.m.). Significance is based on a two-sided Wilcoxon signed-rank

against a slope of zero (PC: $n = 276$ neurons, $Z = -3.1, P = 0.002$; amygdala: $n = 617, Z = -6.6, P < 10^{-10}$; EC: $n = 464, Z = -4.1, P = 4.2 \cdot 10^{-5}$; hippocampus: $n = 633, Z = -6.5, P = 1.0 \cdot 10^{-10}$; PHC: $n = 418, Z = -2.4, P = 0.018$, neurons with a non-zero pre-odour baseline firing rate). **e**, First-trial effect in the human piriform cortex. Changes in firing rates (z-scores, mean ± s.e.m. in black) of odour-modulated neurons between consecutive trials. For each region, we calculated the differences of firing rate between successive trials (i.e., 2nd-1st, 3rd-2nd,..., 8th-7th trial). Firing rate changes were significantly different across trials and regions (one-way ANOVA, $F_{27,2611} = 3.8, P = 1.5 \cdot 10^{-10}, n = 377$ neurons). PC neurons showed the most pronounced decline in firing rate from first to second trial, as indicated by the blue cross and error bar. All 27 pairwise comparisons (blue cross versus each of the remaining crosses) were statistically significant ($P < 0.05$) after Tukey's correction for multiple comparisons across all $n = 378$ (binomial coefficient for selecting 2 out of 27) pair-wise comparisons. The y-axis is truncated to display 99% of the data to improve visibility. ****$P < 0.0001$, **$P < 0.01$, *$P < 0.05$, n.s. = not significant.

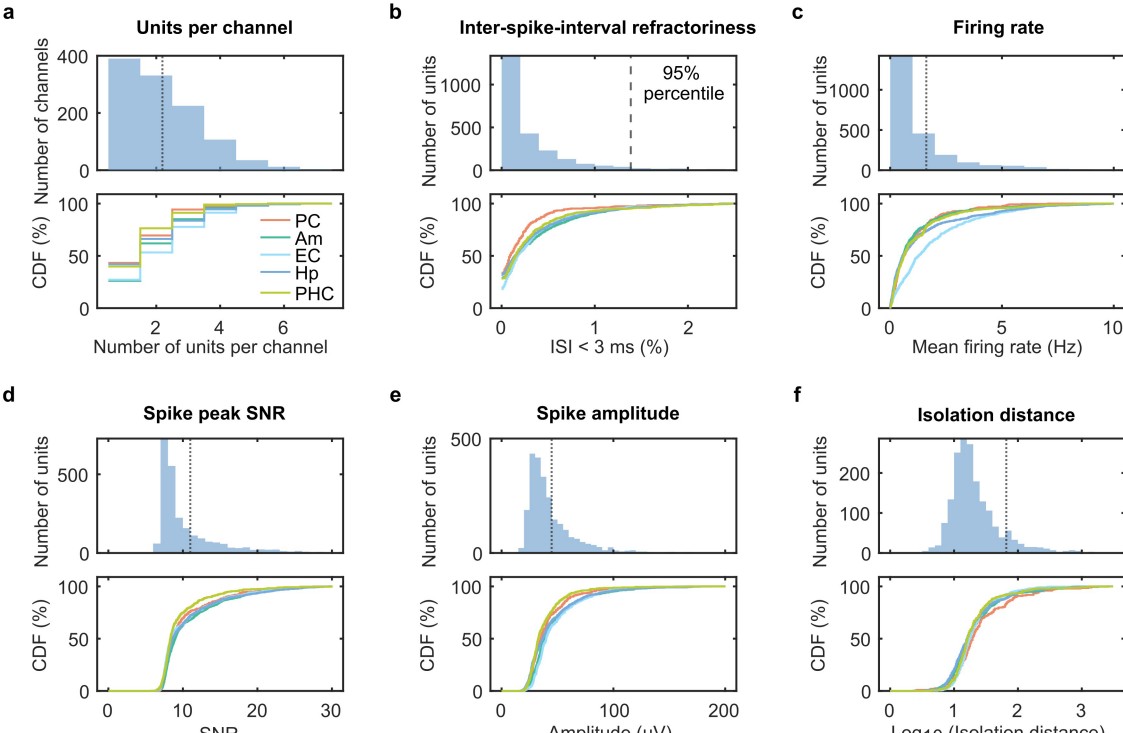

**Extended Data Fig. 8 | Spike-sorting and recording-quality metrics. a**, After automated spike sorting and manual verification, we identified $n$ = 2,416 units, with an average of 2.19 ± 0.04 (mean ± s.e.m., dotted vertical line) units per channel. Only channels with at least one recorded unit were included. Cumulative density functions (CDF) per brain region are shown as coloured solid lines in the lower panels. **b**, Proportions of Inter-spike intervals (ISI) shorter than 3 ms. Units exhibited an average proportion of (mean ± s.e.m.) 0.36 ± 0.01% of ISI intervals below 3 ms. More than 95% of all units showed less than 1.4% of ISIs below 3 ms (dashed vertical line). **c**, Distribution of mean firing rates (mean ± s.e.m.: 1.62 ± 0.05 Hz, dotted vertical line). **d**, Spike peak amplitude SNR (mean ± s.e.m.: 11 ± 0.1, dotted vertical line). Peak SNR was calculated by dividing the peak amplitude by the standard deviation of the background activity, estimated based on the median absolute deviation (MAD) as SD = MAD/0.6745[63]. **e**, Mean spike peak amplitude distribution (mean ± s.e.m.: 44.8 ± 0.5µV, dotted vertical line). **f**, Isolation distance (mean ± s.e.m.: 66 ± 12, for the 1786 clusters for which this measure could be calculated).

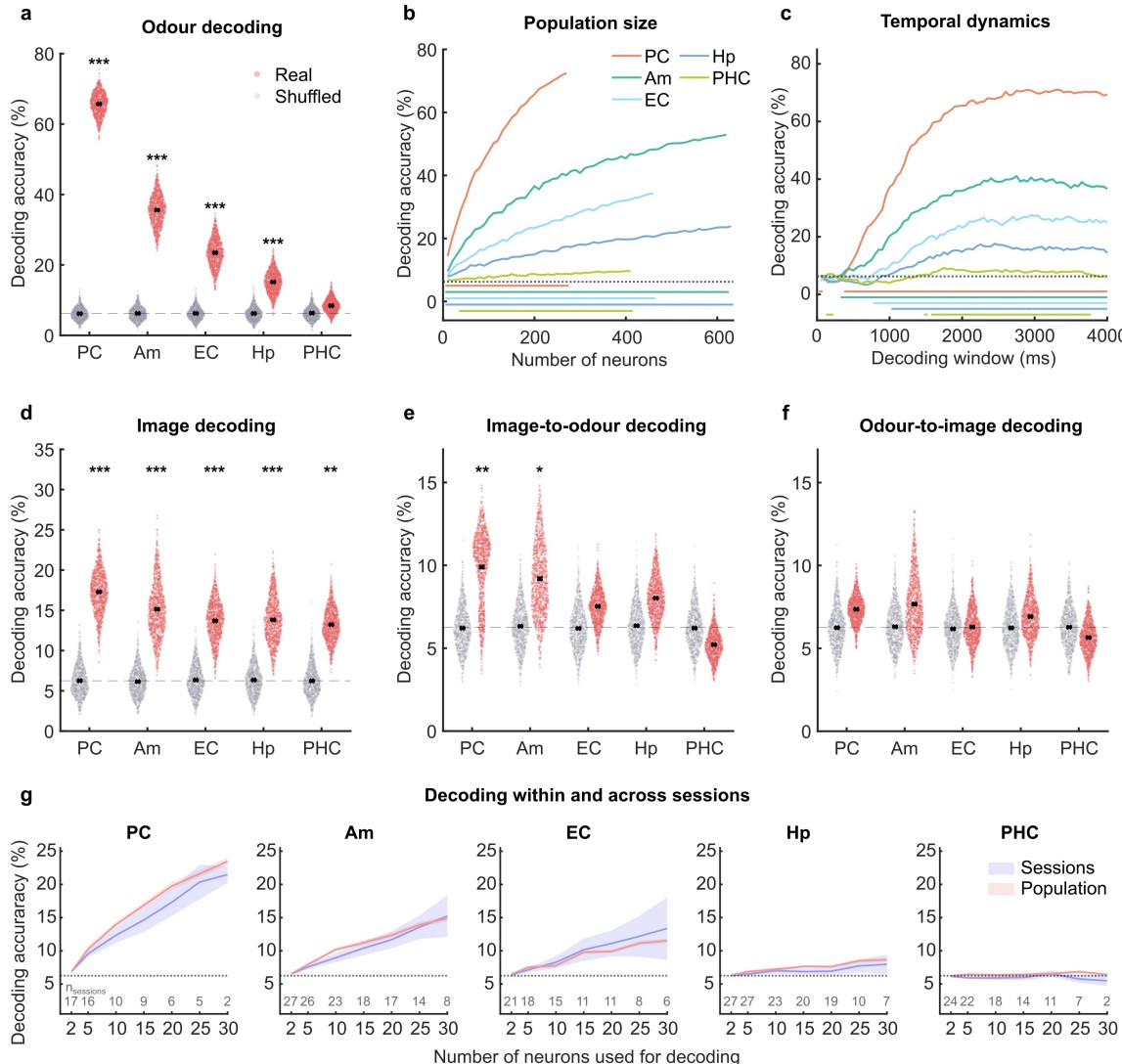

**Extended Data Fig. 9 | Replication of the decoding analysis after excluding the first trial. a**, Odour-identity decoding accuracy based on neuronal activity separated by region. Each red dot in the distributions shows the decoding performance based on 200 randomly drawn neurons (1,000 subsampling runs). Mean decoding performance and s.e.m. across subsampling runs are shown in black. Grey dots indicate decoding performance on label-permuted data. The dashed horizontal line indicates chance level (6.25%). Significance based on percentile of mean decoding performance of the real data in the surrogate distribution (PC: $P < 0.001$; amygdala: $P < 0.001$; EC: $P < 0.001$; hippocampus: $P < 0.001$; PHC: $P = 0.12$, label permutation test with $n = 1,000$ permutations). **b**, Performance of odour-identity decoding (mean ± s.e.m.) as a function of the number of neurons included in the decoding analysis using 100 subsampling runs. Horizontal bars indicate neuron counts for which decoding performance significantly exceeded chance ($P < 0.05$, right-sided Wilcoxon signed-rank against chance after Bonferroni correction for different neuron counts). **c**, Performance of odour-identity decoding (mean ± s.e.m.) as a function of the decoding time-window beginning at odour onset using 200 randomly drawn neurons and 100 subsampling runs. Horizontal bars indicate times where decoding performance significantly exceeded chance ($P < 0.05$, right-sided Wilcoxon signed-rank against chance after Bonferroni correction for 80 decoding time windows; beginning of sustained significant decoding:

PC: 400 ms; amygdala: 350 ms; EC: 800 ms; hippocampus: 1,050 ms; PHC: 1,600 ms). **d**, Image-identity decoding accuracy based on neuronal activity separated by region, depicted as in (a). All regions exhibited significant decoding of image identities (PC: $P < 0.001$; amygdala: $P < 0.001$; EC: $P < 0.001$; hippocampus: $P < 0.001$; PHC: $P = 0.003$, label permutation test with $n = 1,000$ permutations, as in (a)). Decoding accuracy in PC surpassed all other regions. **e-f**, Decoding accuracy (as in a) for a cross-modal decoding analysis trained on images and evaluated on odours (e), and vice versa (f). (Image-to-odour: PC: $P = 0.008$; amygdala: $P = 0.018$; EC: $P = 0.15$; hippocampus: $P = 0.1$; PHC: $P = 0.79$; Odour-to-image: PC: $P = 0.17$; amygdala: $P = 0.14$; EC: $P = 0.45$; hippocampus: $P = 0.28$; PHC: $P = 0.7$, label permutation test with $n = 1,000$ permutations, as in (a)). PC and amygdala reached substantially higher decoding accuracies than any of the other regions in both cross-modal decoding analyses. **g**, Odour-decoding accuracy as a function of the number of neurons used for decoding, sampled across participants (red, mean ± s.e.m.) or within participants (blue, mean ± s.e.m. across sessions), as in Fig. 2c (100 times randomly subsampled with 8 cross-validation data splits and 10 resample runs). Chance level (6.25%) shown as dashed horizontal line. Note that with 8-32 microwires per anatomical target region, it is rarely possible to simultaneously record the activity of 30 or more neurons per participant. ***$P < 0.001$, **$P < 0.01$, *$P < 0.05$.

# Extended Data Table 1 | Generalized linear mixed-effects models across participants and sessions

## Extended Data Table 1a | Odours vs. odourless control

| Fixed Effect | Estimate | Std. Error | t-value | P-value | 95%-CI |
|---|---|---|---|---|---|
| Intercept | 0.14 | 0.34 | 0.43 | 0.67 | [-0.51, 0.80] |
| Region-PC | 0.59 | 0.38 | 1.56 | 0.12 | [-0.15, 1.32] |
| Region-Am | 0.55 | 0.45 | 1.23 | 0.22 | [-0.33, 1.44] |
| Region-EC | 0.96 | 0.48 | 2.01 | 0.04 | [ 0.02, 1.90] |
| Region-Hp | 0.47 | 0.52 | 0.92 | 0.36 | [-0.54, 1.48] |
| Control | 0.13 | 0.05 | 2.54 | 0.01 | [ 0.03, 0.23] |
| Region-PC × Control | -0.46 | 0.05 | -10 | >0.0001 | [-0.55, -0.37] |
| Region-Am × Control | -0.2 | 0.04 | -4.75 | >0.0001 | [-0.29, -0.12] |
| Region-EC × Control | -0.21 | 0.05 | -4.56 | >0.0001 | [-0.30, -0.12] |
| Region-Hp × Control | -0.23 | 0.05 | -5.04 | >0.0001 | [-0.32, -0.14] |

## Extended Data Table 1b | Odour repetition suppression

| Fixed Effect | Estimate | Std. Error | t-value | P-value | 95%-CI |
|---|---|---|---|---|---|
| Intercept | 0.08 | 0.34 | 0.24 | 0.81 | [-0.58, 0.74] |
| Region-PC | 0.85 | 0.38 | 2.24 | 0.03 | [ 0.11, 1.60] |
| Region-Am | 0.68 | 0.47 | 1.44 | 0.15 | [-0.25, 1.61] |
| Region-EC | 1.03 | 0.49 | 2.1 | 0.04 | [ 0.07, 1.99] |
| Region-Hp | 0.59 | 0.54 | 1.1 | 0.27 | [-0.46, 1.65] |
| Repetition | 0.02 | 0.01 | 2.63 | 0.01 | [ 0.00, 0.03] |
| Region-PC × Repetition | -0.06 | 0.005 | -12.44 | >0.0001 | [-0.07, -0.05] |
| Region-Am × Repetition | -0.04 | 0.005 | -8.14 | >0.0001 | [-0.05, -0.03] |
| Region-EC × Repetition | -0.03 | 0.005 | -6.16 | >0.0001 | [-0.04, -0.02] |
| Region-Hp × Repetition | -0.04 | 0.005 | -7.72 | >0.0001 | [-0.05, -0.03] |

## Extended Data Table 1c | Subjective odour valence coding

| Fixed Effect | Estimate | Std. Error | t-value | P-value | 95%-CI |
|---|---|---|---|---|---|
| Intercept | 0.03 | 0.36 | 0.07 | 0.94 | [-0.67, 0.72] |
| Region-PC | 0.71 | 0.38 | 1.87 | 0.06 | [-0.04, 1.46] |
| Region-Am | 0.76 | 0.43 | 1.77 | 0.08 | [-0.08, 1.60] |
| Region-EC | 1.06 | 0.48 | 2.19 | 0.03 | [ 0.11, 2.00] |
| Region-Hp | 0.66 | 0.48 | 1.37 | 0.17 | [-0.29, 1.61] |
| Liked | -0.13 | 0.05 | -2.34 | 0.02 | [-0.23, -0.02] |
| Region-PC × Liked | 0.06 | 0.03 | 1.94 | 0.052 | [ 0.00, 0.13] |
| Region-Am × Liked | 0.15 | 0.03 | 4.63 | >0.0001 | [ 0.08, 0.21] |
| Region-EC × Liked | 0.04 | 0.03 | 1.26 | 0.21 | [-0.02, 0.11] |
| Region-Hp × Liked | 0.01 | 0.03 | 0.33 | 0.74 | [-0.05, 0.08] |

Results of general linear-mixed effects models (GLMMs) for firing rates of odour-modulated neurons in response to odours as compared to odourless controls (a), repeated odour presentations (b), and subjective valence ratings (c). Models include different brain regions and account for neurons recorded across participants and sessions as well as their nested structure. **a**, Odours elicited significantly stronger activity of odour-modulated neurons than odourless controls in the PC, amygdala and hippocampus across patients and sessions. (GLMM predicting spike counts (SC) based on odour versus odourless control, brain region, and their interaction. Model: SC ~1 + Control x Region + (Control|ParticipantID)+(Region|ParticipantID)+ (control|ParticipantID:SessionID)+(Region|ParticipantID:SessionID)+(1|ParticipantID:SessionID:UnitID), using the odour condition and PHC as reference). **b**, Repeated odour presentations led to reduced firing of odour-modulated neurons specifically in the PC, amygdala, EC and hippocampus across patients and sessions. (GLMM predicting spike counts for repeated odour presentations (Rep), brain region and their interaction. Model: SC ~1 + Rep x Region + (Rep|ParticipantID)+(Region|ParticipantID)+(Rep|ParticipantID:SessionID)+(Region|ParticipantID:SessionID)+ (1|ParticipantID:SessionID:UnitID), using PHC as reference). **c**, Behavioural valence ratings predicted firing of odour-modulated neurons especially in the amygdala across patients and sessions (GLMM predicting spike counts based on valence (liked vs disliked), region and their interaction. Model: SC ~1 + Liked x Region + (Liked|ParticipantID)+(Region|ParticipantID)+ (Liked|ParticipantID:SessionID)+(Region|ParticipantID:SessionID)+(1|ParticipantID:SessionID:UnitID), with disliked and PHC as reference).

# Reporting Summary

## Statistics

For all statistical analyses, confirm that the following items are present in the figure legend, table legend, main text, or Methods section.

| n/a | Confirmed | |
|---|---|---|
| ☐ | ☒ | The exact sample size (*n*) for each experimental group/condition, given as a discrete number and unit of measurement |
| ☐ | ☒ | A statement on whether measurements were taken from distinct samples or whether the same sample was measured repeatedly |
| ☐ | ☒ | The statistical test(s) used AND whether they are one- or two-sided<br>*Only common tests should be described solely by name; describe more complex techniques in the Methods section.* |
| ☒ | ☐ | A description of all covariates tested |
| ☐ | ☒ | A description of any assumptions or corrections, such as tests of normality and adjustment for multiple comparisons |
| ☐ | ☒ | A full description of the statistical parameters including central tendency (e.g. means) or other basic estimates (e.g. regression coefficient) AND variation (e.g. standard deviation) or associated estimates of uncertainty (e.g. confidence intervals) |
| ☐ | ☒ | For null hypothesis testing, the test statistic (e.g. $F$, $t$, $r$) with confidence intervals, effect sizes, degrees of freedom and $P$ value noted<br>*Give P values as exact values whenever suitable.* |
| ☒ | ☐ | For Bayesian analysis, information on the choice of priors and Markov chain Monte Carlo settings |
| ☐ | ☒ | For hierarchical and complex designs, identification of the appropriate level for tests and full reporting of outcomes |
| ☒ | ☐ | Estimates of effect sizes (e.g. Cohen's $d$, Pearson's $r$), indicating how they were calculated |

*Our web collection on statistics for biologists contains articles on many of the points above.*

## Software and code

Policy information about availability of computer code

| Data collection | *Neurophysiological data were recorded using Behnke-Fried depth electrodes (AdTech, Racine, WI) equipped with microwire bundles protruding from the tip of the electrodes (3-5 mm). Data were amplified and recorded using a 256-channel ATLAS amplifier (Neuralynx, Bozeman, MT) and the Pegasus software (version 2.1.1, Neuralynx, Bozeman, MT).* |
|---|---|

| Data analysis | *Statistical analyses were conducted in MATLAB 2021a including the Statistics and Machine Learning Toolbox (The MathWorks, Natick, MA)* |
| --- | --- |
| | *The experimental paradigm was implemented with Psychtoolbox3 (www.psychtoolbox.org)* |
| | *Spike extraction and sorting was performed using Combinato (no version, https://github.com/jniediek/combinato)* |
| | *Electrode localization was performed using the LeGUI software package (Davis et al. 2021, version 1.2, DOI: 10.3389/fnins.2021.769872, https://github.com/Rolston-Lab/LeGUI)* |
| | *Decoding analyses were performed using the Neural Decoding Toolbox (Meyers 2013, version 1.0.4, DOI: 10.3389/fninf.2013.00008, https://www.readout.info/)* |
| | *Respiratory signals analyzed with Breathmetrics toolbox (Noto et al. 2018, version 2.0, DOI: 10.1093/chemse/bjy045, https://github.com/zelanolab/breathmetrics)* |
| | *Matlab toolboxes for visualizing Venn diagrams (version 1.7, https://de.mathworks.com/matlabcentral/fileexchange/22282-venn)* |
| | *Matlab toolboxes for exact binomial tests (version 2.0, https://www.mathworks.com/matlabcentral/fileexchange/24813-mybinomtest-s-n-p-sided)* |
| | *Electrode visualization based on Fieldtrip (Oostenveld et al. 2011, version 213bc8bcb, DOI: 10.1155/2011/156869)* |
| | *and the Matlab function 'plot_ecog' (https://github.com/s-michelmann/moment-by-moment-tracking/blob/master/plot_ecog.m)* |
| | |
| | *Custom MATLAB code to reproduce the main figures and analysis of this study are publicly available on GitHub: https://github.com/marcelkehl/HumanOdorRepresentations.* |

For manuscripts utilizing custom algorithms or software that are central to the research but not yet described in published literature, software must be made available to editors and reviewers. We strongly encourage code deposition in a community repository (e.g. GitHub). See the Nature Portfolio guidelines for submitting code & software for further information.

# Data

Policy information about availability of data

All manuscripts must include a data availability statement. This statement should provide the following information, where applicable:

- Accession codes, unique identifiers, or web links for publicly available datasets
- A description of any restrictions on data availability
- For clinical datasets or third party data, please ensure that the statement adheres to our policy

Data to reproduce the main figures and analysis of this study are publicly available on GitHub:
https://github.com/marcelkehl/HumanOdorRepresentations

Reference valence ratings of the standardized odors in our study are available from Toet et al. 2020 (https://doi.org/10.1007/s12078-019-09275-7)

# Research involving human participants, their data, or biological material

Policy information about studies with human participants or human data. See also policy information about sex, gender (identity/presentation), and sexual orientation and race, ethnicity and racism.

| Reporting on sex and gender | *Our study included 17 participants, of whom 12 were female and 5 were male, according to the clinical reports.* |
| --- | --- |
| | *Our study did not include sex- or gender-specific analyses.* |

| Reporting on race, ethnicity, or other socially relevant groupings | *No socially constructed variables were used or analyzed in this study.* |
| --- | --- |

| Population characteristics | *The population included 17 participants (12 female, 5 male) aged 22 to 60 years (mean ± SD: 41.3 ± 11.7 years).* |
| --- | --- |
| | *In this study, no characteristics of the population were included as covariates.* |

| Recruitment | *This study comprises a rare-opportunity sample, based on the treatment of drug-resistant epilepsy patients undergoing invasive seizure monitoring at the Department of Epileptology at the University of Bonn Medical Center, Germany.* |
| --- | --- |
| | *Patients were offered to participate in research including single-neuron recordings only after they opted for invasive epilepsy diagnostics in consultation with their treating physicians. Informed written consent for microwire recordings was obtained from all participants. Patients were informed that they could withdraw from research at any time, and without any impact on their clinical care. Patients who consented to microwire implantation were subsequently invited to participate in this study. All patients gave informed written consent to participate in this study in accordance with the Medical Institutional Review Board of the University of Bonn, Germany. Only patients over the age of 18 years were recruited.* |
| | *None of the participants reported subjective olfactory dysfunction or acute respiratory tract infection.* |

| Ethics oversight | *Informed written consent was provided by each patient. All studies conformed with and were approved by the Medical Institutional Review Board of the University of Bonn, Germany (License No. 289/20).* |
| --- | --- |

Note that full information on the approval of the study protocol must also be provided in the manuscript.

# Field-specific reporting

Please select the one below that is the best fit for your research. If you are not sure, read the appropriate sections before making your selection.

☒ Life sciences        ☐ Behavioural & social sciences        ☐ Ecological, evolutionary & environmental sciences

# Life sciences study design

All studies must disclose on these points even when the disclosure is negative.

| | |
|---|---|
| Sample size | Our analysis is based on a total of 2,416 units recorded in 27 recording sessions from 17 epilepsy patients suffering from drug-resistant epilepsy and undergoing invasive seizure monitoring with the goal of subsequent neurosurgical resection.<br>No a-priori sample-size calculation was performed. Our comprehensive dataset easily complies with or exceeds current standards in the field of human single unit recordings (e.g., Jamali et al., Nature, 2024; Qasim et al., Cell, 2021) |
| Data exclusions | No data were excluded from the analysis |
| Replication | All experiments were performed across multiple patients using equivalent electrode and recording techniques, and the same anatomical target regions. Here, we investigated the neuronal coding of odors in the human brain. The central findings of our study were replicable across recording sessions and participants.<br>Replication in healthy subjects is not applicable owing to intervention constraints of the invasive recording techniques. |
| Randomization | There was no experimental group assignment in our study. |
| Blinding | The study did not include allocation of subjects to experimental groups so no blinding applies. |

# Behavioural & social sciences study design

All studies must disclose on these points even when the disclosure is negative.

| | |
|---|---|
| Study description | Briefly describe the study type including whether data are quantitative, qualitative, or mixed-methods (e.g. qualitative cross-sectional, quantitative experimental, mixed-methods case study). |
| Research sample | State the research sample (e.g. Harvard university undergraduates, villagers in rural India) and provide relevant demographic information (e.g. age, sex) and indicate whether the sample is representative. Provide a rationale for the study sample chosen. For studies involving existing datasets, please describe the dataset and source. |
| Sampling strategy | Describe the sampling procedure (e.g. random, snowball, stratified, convenience). Describe the statistical methods that were used to predetermine sample size OR if no sample-size calculation was performed, describe how sample sizes were chosen and provide a rationale for why these sample sizes are sufficient. For qualitative data, please indicate whether data saturation was considered, and what criteria were used to decide that no further sampling was needed. |
| Data collection | Provide details about the data collection procedure, including the instruments or devices used to record the data (e.g. pen and paper, computer, eye tracker, video or audio equipment) whether anyone was present besides the participant(s) and the researcher, and whether the researcher was blind to experimental condition and/or the study hypothesis during data collection. |
| Timing | Indicate the start and stop dates of data collection. If there is a gap between collection periods, state the dates for each sample cohort. |
| Data exclusions | If no data were excluded from the analyses, state so OR if data were excluded, provide the exact number of exclusions and the rationale behind them, indicating whether exclusion criteria were pre-established. |
| Non-participation | State how many participants dropped out/declined participation and the reason(s) given OR provide response rate OR state that no participants dropped out/declined participation. |
| Randomization | If participants were not allocated into experimental groups, state so OR describe how participants were allocated to groups, and if allocation was not random, describe how covariates were controlled. |

# Ecological, evolutionary & environmental sciences study design

All studies must disclose on these points even when the disclosure is negative.

| | |
|---|---|
| Study description | Briefly describe the study. For quantitative data include treatment factors and interactions, design structure (e.g. factorial, nested, hierarchical), nature and number of experimental units and replicates. |
| Research sample | Describe the research sample (e.g. a group of tagged Passer domesticus, all Stenocereus thurberi within Organ Pipe Cactus National Monument), and provide a rationale for the sample choice. When relevant, describe the organism taxa, source, sex, age range and any manipulations. State what population the sample is meant to represent when applicable. For studies involving existing datasets, describe the data and its source. |
| Sampling strategy | Note the sampling procedure. Describe the statistical methods that were used to predetermine sample size OR if no sample-size calculation was performed, describe how sample sizes were chosen and provide a rationale for why these sample sizes are sufficient. |

| | |
|---|---|
| Data collection | *Describe the data collection procedure, including who recorded the data and how.* |
| Timing and spatial scale | *Indicate the start and stop dates of data collection, noting the frequency and periodicity of sampling and providing a rationale for these choices. If there is a gap between collection periods, state the dates for each sample cohort. Specify the spatial scale from which the data are taken* |
| Data exclusions | *If no data were excluded from the analyses, state so OR if data were excluded, describe the exclusions and the rationale behind them, indicating whether exclusion criteria were pre-established.* |
| Reproducibility | *Describe the measures taken to verify the reproducibility of experimental findings. For each experiment, note whether any attempts to repeat the experiment failed OR state that all attempts to repeat the experiment were successful.* |
| Randomization | *Describe how samples/organisms/participants were allocated into groups. If allocation was not random, describe how covariates were controlled. If this is not relevant to your study, explain why.* |
| Blinding | *Describe the extent of blinding used during data acquisition and analysis. If blinding was not possible, describe why OR explain why blinding was not relevant to your study.* |

Did the study involve field work?   ☐ Yes   ☐ No

## Field work, collection and transport

| | |
|---|---|
| Field conditions | *Describe the study conditions for field work, providing relevant parameters (e.g. temperature, rainfall).* |
| Location | *State the location of the sampling or experiment, providing relevant parameters (e.g. latitude and longitude, elevation, water depth).* |
| Access & import/export | *Describe the efforts you have made to access habitats and to collect and import/export your samples in a responsible manner and in compliance with local, national and international laws, noting any permits that were obtained (give the name of the issuing authority, the date of issue, and any identifying information).* |
| Disturbance | *Describe any disturbance caused by the study and how it was minimized.* |

# Reporting for specific materials, systems and methods

We require information from authors about some types of materials, experimental systems and methods used in many studies. Here, indicate whether each material, system or method listed is relevant to your study. If you are not sure if a list item applies to your research, read the appropriate section before selecting a response.

### Materials & experimental systems

| n/a | Involved in the study |
|---|---|
| ☒ ☐ | Antibodies |
| ☒ ☐ | Eukaryotic cell lines |
| ☒ ☐ | Palaeontology and archaeology |
| ☒ ☐ | Animals and other organisms |
| ☒ ☐ | Clinical data |
| ☒ ☐ | Dual use research of concern |
| ☒ ☐ | Plants |

### Methods

| n/a | Involved in the study |
|---|---|
| ☒ ☐ | ChIP-seq |
| ☒ ☐ | Flow cytometry |
| ☒ ☐ | MRI-based neuroimaging |

## Antibodies

| | |
|---|---|
| Antibodies used | *Describe all antibodies used in the study; as applicable, provide supplier name, catalog number, clone name, and lot number.* |
| Validation | *Describe the validation of each primary antibody for the species and application, noting any validation statements on the manufacturer's website, relevant citations, antibody profiles in online databases, or data provided in the manuscript.* |

## Eukaryotic cell lines

Policy information about cell lines and Sex and Gender in Research

| | |
|---|---|
| Cell line source(s) | *State the source of each cell line used and the sex of all primary cell lines and cells derived from human participants or vertebrate models.* |

| Authentication | Describe the authentication procedures for each cell line used OR declare that none of the cell lines used were authenticated. |
| Mycoplasma contamination | Confirm that all cell lines tested negative for mycoplasma contamination OR describe the results of the testing for mycoplasma contamination OR declare that the cell lines were not tested for mycoplasma contamination. |
| Commonly misidentified lines (See ICLAC register) | Name any commonly misidentified cell lines used in the study and provide a rationale for their use. |

## Palaeontology and Archaeology

| Specimen provenance | Provide provenance information for specimens and describe permits that were obtained for the work (including the name of the issuing authority, the date of issue, and any identifying information). Permits should encompass collection and, where applicable, export. |
| Specimen deposition | Indicate where the specimens have been deposited to permit free access by other researchers. |
| Dating methods | If new dates are provided, describe how they were obtained (e.g. collection, storage, sample pretreatment and measurement), where they were obtained (i.e. lab name), the calibration program and the protocol for quality assurance OR state that no new dates are provided. |

☐ Tick this box to confirm that the raw and calibrated dates are available in the paper or in Supplementary Information.

| Ethics oversight | Identify the organization(s) that approved or provided guidance on the study protocol, OR state that no ethical approval or guidance was required and explain why not. |

Note that full information on the approval of the study protocol must also be provided in the manuscript.

## Animals and other research organisms

Policy information about studies involving animals; ARRIVE guidelines recommended for reporting animal research, and Sex and Gender in Research

| Laboratory animals | For laboratory animals, report species, strain and age OR state that the study did not involve laboratory animals. |
| Wild animals | Provide details on animals observed in or captured in the field; report species and age where possible. Describe how animals were caught and transported and what happened to captive animals after the study (if killed, explain why and describe method; if released, say where and when) OR state that the study did not involve wild animals. |
| Reporting on sex | Indicate if findings apply to only one sex; describe whether sex was considered in study design, methods used for assigning sex. Provide data disaggregated for sex where this information has been collected in the source data as appropriate; provide overall numbers in this Reporting Summary. Please state if this information has not been collected. Report sex-based analyses where performed, justify reasons for lack of sex-based analysis. |
| Field-collected samples | For laboratory work with field-collected samples, describe all relevant parameters such as housing, maintenance, temperature, photoperiod and end-of-experiment protocol OR state that the study did not involve samples collected from the field. |
| Ethics oversight | Identify the organization(s) that approved or provided guidance on the study protocol, OR state that no ethical approval or guidance was required and explain why not. |

Note that full information on the approval of the study protocol must also be provided in the manuscript.

## Clinical data

Policy information about clinical studies

All manuscripts should comply with the ICMJE guidelines for publication of clinical research and a completed CONSORT checklist must be included with all submissions.

| Clinical trial registration | Provide the trial registration number from ClinicalTrials.gov or an equivalent agency. |
| Study protocol | Note where the full trial protocol can be accessed OR if not available, explain why. |
| Data collection | Describe the settings and locales of data collection, noting the time periods of recruitment and data collection. |
| Outcomes | Describe how you pre-defined primary and secondary outcome measures and how you assessed these measures. |

# Dual use research of concern

Policy information about dual use research of concern

## Hazards

Could the accidental, deliberate or reckless misuse of agents or technologies generated in the work, or the application of information presented in the manuscript, pose a threat to:

No | Yes

☐ ☐ Public health

☐ ☐ National security

☐ ☐ Crops and/or livestock

☐ ☐ Ecosystems

☐ ☐ Any other significant area

## Experiments of concern

Does the work involve any of these experiments of concern:

No | Yes

☐ ☐ Demonstrate how to render a vaccine ineffective

☐ ☐ Confer resistance to therapeutically useful antibiotics or antiviral agents

☐ ☐ Enhance the virulence of a pathogen or render a nonpathogen virulent

☐ ☐ Increase transmissibility of a pathogen

☐ ☐ Alter the host range of a pathogen

☐ ☐ Enable evasion of diagnostic/detection modalities

☐ ☐ Enable the weaponization of a biological agent or toxin

☐ ☐ Any other potentially harmful combination of experiments and agents

# Plants

Seed stocks | *Report on the source of all seed stocks or other plant material used. If applicable, state the seed stock centre and catalogue number. If plant specimens were collected from the field, describe the collection location, date and sampling procedures.*

Novel plant genotypes | *Describe the methods by which all novel plant genotypes were produced. This includes those generated by transgenic approaches, gene editing, chemical/radiation-based mutagenesis and hybridization. For transgenic lines, describe the transformation method, the number of independent lines analyzed and the generation upon which experiments were performed. For gene-edited lines, describe the editor used, the endogenous sequence targeted for editing, the targeting guide RNA sequence (if applicable) and how the editor was applied.*

Authentication | *Describe any authentication procedures for each seed stock used or novel genotype generated. Describe any experiments used to assess the effect of a mutation and, where applicable, how potential secondary effects (e.g. second site T-DNA insertions, mosiacism, off-target gene editing) were examined.*

# ChIP-seq

## Data deposition

☐ Confirm that both raw and final processed data have been deposited in a public database such as GEO.

☐ Confirm that you have deposited or provided access to graph files (e.g. BED files) for the called peaks.

Data access links
*May remain private before publication.* | *For "Initial submission" or "Revised version" documents, provide reviewer access links.  For your "Final submission" document, provide a link to the deposited data.*

Files in database submission | *Provide a list of all files available in the database submission.*

Genome browser session
(e.g. UCSC) | *Provide a link to an anonymized genome browser session for "Initial submission" and "Revised version" documents only, to enable peer review.  Write "no longer applicable" for "Final submission" documents.*

## Methodology

**Replicates**
*Describe the experimental replicates, specifying number, type and replicate agreement.*

**Sequencing depth**
*Describe the sequencing depth for each experiment, providing the total number of reads, uniquely mapped reads, length of reads and whether they were paired- or single-end.*

**Antibodies**
*Describe the antibodies used for the ChIP-seq experiments; as applicable, provide supplier name, catalog number, clone name, and lot number.*

**Peak calling parameters**
*Specify the command line program and parameters used for read mapping and peak calling, including the ChIP, control and index files used.*

**Data quality**
*Describe the methods used to ensure data quality in full detail, including how many peaks are at FDR 5% and above 5-fold enrichment.*

**Software**
*Describe the software used to collect and analyze the ChIP-seq data. For custom code that has been deposited into a community repository, provide accession details.*

# Flow Cytometry

## Plots

Confirm that:

☐ The axis labels state the marker and fluorochrome used (e.g. CD4-FITC).

☐ The axis scales are clearly visible. Include numbers along axes only for bottom left plot of group (a 'group' is an analysis of identical markers).

☐ All plots are contour plots with outliers or pseudocolor plots.

☐ A numerical value for number of cells or percentage (with statistics) is provided.

## Methodology

**Sample preparation**
*Describe the sample preparation, detailing the biological source of the cells and any tissue processing steps used.*

**Instrument**
*Identify the instrument used for data collection, specifying make and model number.*

**Software**
*Describe the software used to collect and analyze the flow cytometry data. For custom code that has been deposited into a community repository, provide accession details.*

**Cell population abundance**
*Describe the abundance of the relevant cell populations within post-sort fractions, providing details on the purity of the samples and how it was determined.*

**Gating strategy**
*Describe the gating strategy used for all relevant experiments, specifying the preliminary FSC/SSC gates of the starting cell population, indicating where boundaries between "positive" and "negative" staining cell populations are defined.*

☐ Tick this box to confirm that a figure exemplifying the gating strategy is provided in the Supplementary Information.

# Magnetic resonance imaging

## Experimental design

**Design type**
*Indicate task or resting state; event-related or block design.*

**Design specifications**
*Specify the number of blocks, trials or experimental units per session and/or subject, and specify the length of each trial or block (if trials are blocked) and interval between trials.*

**Behavioral performance measures**
*State number and/or type of variables recorded (e.g. correct button press, response time) and what statistics were used to establish that the subjects were performing the task as expected (e.g. mean, range, and/or standard deviation across subjects).*

## Acquisition

**Imaging type(s)**
> Specify: functional, structural, diffusion, perfusion.

**Field strength**
> Specify in Tesla

**Sequence & imaging parameters**
> Specify the pulse sequence type (gradient echo, spin echo, etc.), imaging type (EPI, spiral, etc.), field of view, matrix size, slice thickness, orientation and TE/TR/flip angle.

**Area of acquisition**
> State whether a whole brain scan was used OR define the area of acquisition, describing how the region was determined.

**Diffusion MRI**  ☐ Used   ☐ Not used

## Preprocessing

**Preprocessing software**
> Provide detail on software version and revision number and on specific parameters (model/functions, brain extraction, segmentation, smoothing kernel size, etc.).

**Normalization**
> If data were normalized/standardized, describe the approach(es): specify linear or non-linear and define image types used for transformation OR indicate that data were not normalized and explain rationale for lack of normalization.

**Normalization template**
> Describe the template used for normalization/transformation, specifying subject space or group standardized space (e.g. original Talairach, MNI305, ICBM152) OR indicate that the data were not normalized.

**Noise and artifact removal**
> Describe your procedure(s) for artifact and structured noise removal, specifying motion parameters, tissue signals and physiological signals (heart rate, respiration).

**Volume censoring**
> Define your software and/or method and criteria for volume censoring, and state the extent of such censoring.

## Statistical modeling & inference

**Model type and settings**
> Specify type (mass univariate, multivariate, RSA, predictive, etc.) and describe essential details of the model at the first and second levels (e.g. fixed, random or mixed effects; drift or auto-correlation).

**Effect(s) tested**
> Define precise effect in terms of the task or stimulus conditions instead of psychological concepts and indicate whether ANOVA or factorial designs were used.

**Specify type of analysis:**  ☐ Whole brain   ☐ ROI-based   ☐ Both

**Statistic type for inference**

(See Eklund et al. 2016)
> Specify voxel-wise or cluster-wise and report all relevant parameters for cluster-wise methods.

**Correction**
> Describe the type of correction and how it is obtained for multiple comparisons (e.g. FWE, FDR, permutation or Monte Carlo).

## Models & analysis

n/a | Involved in the study
☐ ☐ Functional and/or effective connectivity
☐ ☐ Graph analysis
☐ ☐ Multivariate modeling or predictive analysis

**Functional and/or effective connectivity**
> Report the measures of dependence used and the model details (e.g. Pearson correlation, partial correlation, mutual information).

**Graph analysis**
> Report the dependent variable and connectivity measure, specifying weighted graph or binarized graph, subject- or group-level, and the global and/or node summaries used (e.g. clustering coefficient, efficiency, etc.).

**Multivariate modeling and predictive analysis**
> Specify independent variables, features extraction and dimension reduction, model, training and evaluation metrics.

