## [Peer Review file · Nature]

Manuscript Title: Single-Neuron Representations of Odors in the Human Brain

Reviewer Comments & Author Rebuttals

Reviewer Reports on the Initial Version:

Referees' comments:

Referee #1 (Remarks to the Author):

Notes on Kehl et al; Nature 2023

In this exciting manuscript, the authors record from populations of single neurons in humans. They focus primarily on the piriform cortex (PC), and compared these recordings with recordings in other medial temporal lobe (MTL) areas, including the amygdala, hippocampus, entorhinal cortex (EC), and parahippocampal cortex (PHC).

The central goal of sensory neuroscience is to relate neural activity to perception. The obvious power of this study is the ability to ask the subject what they are smelling and relate that directly to the spiking activity of populations of neurons in different parts of the brain.

The first three figures of this paper show nicely that the odor responses they recorded in humans are generally consistent with what people have observed in rodents. This alone is a very important contribution. However, the authors then really take advantage of recording in humans. Their major novel findings are:

1. Neurons in the amygdala fire more strongly for liked than disliked odors, suggesting that the amygdala is encoding valence. The authors are careful to state in the methods that valence is a loaded term that means different things to different people. I think their use of valence here is fine, with that proviso.
2. They showed that accurate reporting, which they call “behavioral identification” correlates with increased decoding accuracy in the hippocampus only.
3. All areas show visual responses, with better image decoding in the PC than the other areas. Although the idea of multimodal responses in any brain area is not hugely surprising, to me, that decoding visual responses is more accurate in the PC than the amygdala or hippocampus is both novel and very important.
4. The authors made the remarkable observation that they can decode in PC and amygdala when training on image responses and testing on odor responses, and only in the amygdala when training on the odor responses and testing on the image responses.

A strength of this manuscript is that the authors find that one area is better at one type of representation and another area is better at a different one. It is not just, for example, that they got

better recordings in the PC, and therefore that that area was best at all decoding analyses. This serves as a good internal control and adds robustness to their findings.

Overall, this is a hugely important study that should be seen by the widest audience possible. It not only validates a decade of work in animals by showing similar results in humans, but it goes way beyond that because by registering single neuron spiking activity with direct perceptual reports, which cannot be done in animals, the authors have revealed a remarkable level of complexity PC activity, such as semantic representations. There is an impressively large dataset, the recordings appear of high quality, and the analyses and statistics are appropriate. I have only a few small points I would like clarified.

- Many of the areas the authors recorded from contain multiple functionally distinct subregions, e.g., “the amygdala” and “the hippocampus”. Obviously, recordings in humans entail less control than one has with animal experiments and there will likely be considerable variability across subjects. Fig. 1b notwithstanding, the authors should acknowledge and state this specifically.

- Fig. 4g: Why are decoding accuracies so low in the PC and amygdala? The y axis values in Fig. 4g align with the decoding accuracies reported in Fig. 2b in all other areas, but they are much lower in the amygdala and especially in the PC.

- Other than the references to “banana” in one amygdala cell (L. 271) and “liquorice” one PC cell (L. 273), and Fig. 5g, there is no discussion of when or how the “words” or “names” were presented, or what the responses in any of the brain areas to the words. In the methods (L. 433-), they describe how they presented the images, but how when or how the words/names were presented. Please either elaborate (preferred) or eliminate (if necessary).

- The observation that there is a pronounced first-trial effect in the PCx but not in the other areas is important because it precludes a simple model in which odor information from the bulb is first received and processed in the PC, and then routed from the PC to the various MTL areas. If so, a large first-trial effect would be expected in these areas too. Also, there is a fairly large first-trial effect in the zebrafish olfactory bulb of the Jacobson et al paper they cite. Is there also a first trial-effect in the human olfactory bulb? If so, then it is even more interesting that this is not seen in the other MTL regions. The authors do not mention this point but may want add a short discussion of this as I think provides important insight into the processing of olfactory information through the human brain.

- Related to this, did the authors include the first trials in their decoding analyses? They do not explicitly state that they did not. Given the large first-trial effect in PC, it may make sense to discard these trials in the decoding analyses, and this could increase decoding accuracy in PC. Some question whether this is fair, but most people, including me, agree that it is. For cross-model studies, in particular, discarding first trials is probably more correct because they always presented the odors before the images. As the authors are asking whether a particular image activates the same neurons that a semantically congruent odor would, the first odor trial should be omitted to compare visual- or word-evoked responses more accurately with the response they would have got had they presented an odor on that trial instead.

Minor points:

L. 215: In this paragraph the authors are writing about how levels single-neuron activity relate to identification. In L. 217-8, they specifically say “behavior identification”. In L. 215 they say “identification performance”, which is vague. They should say “decoding accuracy” or something similar that makes clear that they are writing about classifier performance.

L. 264: “... only generalized from images to odors, but not visa versa...” I found this phrase much more confusing than I should have and suggest being more explicit. e.g., “... only generalized when training on odors and testing on images, and not visa versa”.

Referee #2 (Remarks to the Author):

This manuscript by Kehl and colleagues presents single-neuron responses to odors in human olfactory (piriform cortex, PC; amygdala, Am) and medial temporal lobe structures (entorhinal cortex, EC; hippocampus, Hp; parahippocampal gyrus, PHC). Neurons in these regions (except PHC) are modulated by odors, respond stronger to odors versus no odors, suppressed by repeated stimulation, and encode odor identity as well as the identity of odor-related images. Whereas the activity of amygdala neurons correlates with odor valence, odor encoding in the hippocampus is related to odor identification performance.

I very much enjoyed reading this manuscript. It presents exciting data and is well written. The novelty and significance of the contribution lies in the systematic study of single-neuron responses to odors in humans. I am not aware of a comparable study. For the most part, the findings mirror and confirm results from previous animal studies and imaging work in humans, which is reassuring. However, the data are somewhat limited by shortcomings related to the odor delivery methods and the lack of nasal airflow measurements.

Major comments

1. The odor delivery methods used here are not state-of-the-art. Odors were delivered in the form of (hand-held?) odorized pens. This may induce substantial variability in factors determining the amount of odor that is delivered on a given trial (e.g., length of exposure, distance to nose, etc.). Nasal inhalation was instructed verbally, and inhalation times were logged manually. This approach results variable delivery times and imprecise measurements. Well-controlled odor concentrations and precise delivery times are important for the analysis of the neuronal recording data. Ideally, odors are delivered using olfactometry and delivery times are determined based on measurements of nasal airflow.

2. Another reason to measure nasal airflow is that it is a major driver of neural activity in olfactory areas in rodents (e.g., Fontanini et al. 2003, *J Neurosci*) and humans (e.g., Zelano et al 2016, *J Neurosci*; Sobel et al. 1998, *Nature*). Moreover, sniffing is rapidly modulated by odor pleasantness (e.g., Rozenkrantz et al. 2015, *Curr Biol*) and intensity (Johnson, et al 2003, *J Neurophysiol*). Thus, without accurately measuring and controlling for the effects of nasal airflow, we don't know whether the current results are driven by odor identity or odor-related differences in sniffing and nasal airflow.

3. The main decoding analysis (Figure 2) collapses across neurons recorded in different sessions and from different subjects. This paints an inflated picture of the information content available in individual brains. More appropriate would be a session-wise approach, in which decoding is based on neurons collected within an individual sessions and statistical tests are run across sessions (accounting for subjects). The authors already have decoding accuracies for individual sessions (Figure 4g) and it would be important use these throughout the manuscript.

4. Odor-modulated neurons in Figure 1g are defined based on comparing firing rates between 16 odors. Given this includes the neutral odor, and responses are substantially higher for odor vs neutral, it would be interesting to see the percentage of neurons that are significantly modulated by the 15 actual odors (i.e., excluding the neutral condition)

5. The authors appear surprised by the finding that PC neurons respond to images associated with odors (e.g., in the abstract: "Unexpectedly, we uncover that piriform cortex neurons also encode image identities challenging traditional views of a strictly unimodal ('olfaction-only') role of the human piriform cortex"). I don't think these findings are all that surprising given previous work across species. For instance, the olfactory bulb and PC in rodents respond to visual stimuli associated with odors (e.g., Mandairon et al 2014, *Front Behav Neurosci*), and the human PC responds to visual cues that predict odors (Schultz et al 2017, *Scientific Reports*), odor-associated words (Gonzales et al 2006, *Neuroimage*) and imagined odors (e.g., Djordjevic et al. 2006, *Neuroimage*). Moreover, given that images (and words) used in the current study were strongly associated with specific odors, I also don't think the findings provide evidence that PC processes non-olfactory information. For this, it would be necessary to show that PC neurons respond to images of odorless objects.

6. There is considerable across-subject variability in odor perception (e.g., Mainland et al. 2014, *Nature Neuroscience*; Keller et al. 2017, *Science*) and associated neural responses (e.g., Sagar et al. 2023, *Nature Neuroscience*). Averaging valence ratings and identification performance per odor across participants (e.g., Figure 1e) and relating averages to neural activity (e.g., Figure 4e) is thus not very informative. It would be important to present and analyze behavioral and neural data for individual subjects.

7. Many of the statistical analyses are run across sessions ($n=27$, e.g., Figure 1g, Figure 4g), and it is unclear how the authors controlled for non-independent data points (sessions from the same participants). Alternatively, the authors could average across sessions per subject and then run

statistical analyses across subjects.

8. Other statistical analyses are computed across neurons (e.g., Figure 1h, Figure 3b, Figure 4d) by collapsing across subjects and sessions. Again, neurons from the same subject/session are not independent observations, and it would be important to rather use mixed effects models that control for effects of sessions and subjects.

Minor comments:

1. The authors state that olfactory bulb projects to the EC (line 48). Unlike in rodents, direct projections from the olfactory bulb to EC are rare in non-human primates (Carmichael et al. 1994, *J Comp Neurol*) and seemingly absent in humans (Allison 1954, *J Anat*; Echevarria-Cooper et al. 2022, *J Neurosci*). The authors may want to revise the introduction and adjust their diagram in Figure 1a. Moreover, I think their finding of a 500 ms delay between odor encoding in PC and EC (Figure 2d) may provide interesting electrophysiological support for the lack of a direct projection from the bulb to EC in humans.

2. How can we interpret the sparseness of a neural population if this population does not encode information about a stimulus? Specially, what does it mean that the sparseness of PHC is between PC and amygdala although PHC does not encode odors?

3. The authors cite two navigation studies (Poo et al. 2009, *Nature*; Raithel et al. 2023, *Curr Biol*) as evidence for non-olfactory processing in olfactory cortex. Because these studies focused on odor-based navigation, I don't think they support the authors' argument. Rather, they show that PC represents odor-associated spatial information, which is in line with the current (and previous) findings of PC responding to odor-associated visual information.

Referee #3 (Remarks to the Author):

In Kehl et al the authors present intracranial recordings from patients in olfactory and related areas. Single-unit recordings in humans – despite a lot of recent developments – are still incredibly rare and a unique opportunity to explore mechanistic aspects of exciting human-specific questions.

Kehl et al use quite established “Behnke-Fried” electrodes where a lead of macroelectrodes is implanted as indicated for clinical diagnostics. After implantation a central guide is withdrawn and a set of 8 microelectrode wires inserted through the central space for neuroscience investigation beyond the tips. These small (“high impedance”) wires offer the opportunity to record localized neural activity and potentially multi-unit activity (MUA) or single-unit activity (SUA).

Prior work in human olfactory areas has been largely based on functional imaging (e.g. the Sobel and Gottfried labs) or using microelectrode field potential work (including recent work by the Gottfried lab as cited in the manuscript). To my knowledge, no SUA has been reported in PCX – although many studies pioneered by Fried and many others have used similar opportunities in patients with pharmacologically

intractable epilepsy to record in e.g. hippocampal or various MTL areas with non-olfactory stimuli. This includes e.g. the work by Fried and colleagues describing concept neurons, i.e. neurons that respond to specific concepts rather than visual or other stimuli (famously to celebrities as images, texts, in different constellations etc).

Two aspects are truly novel about the work by Kehl et al. Firstly, this might be the first SUA recording in humans with olfactory stimuli (and in olfactory areas). Secondly, the study offers the opportunity to provide a major puzzle piece to the long-standing question of whether olfaction is a “special sense”, whether olfactory imagery exists and how olfaction interacts with other senses (that are more verbal in humans, more consciously processed, relayed by thalamus).

While the SUA aspect might not on its own constitute a landmark (a number of SUA recordings in humans have been presented in the literature under slightly different conditions), the second point has the potential to be truly a fundamental and transformative one.

I am thus overall very excited about the study. However, this excitement is dampened by a few major concerns:

(1) Critical evaluation of spike identification and spike sorting:

The SUA aspect is critical (not only in its own right but also to provide certainty that e.g. single neurons can respond to odors and multimodal stimuli / imagined odors (see below). Here the authors fall short in detail and critical assessment. The example units displayed in the figures exclusively show positive or bipolar characteristics and appear to have slower waveforms compared to the vast majority of published recordings, e.g. from cortical areas in human patients (Chung, 2022 *Neuron*) and piriform cortex in rodents (Schoonover 2021, *Nature*). Positive spikes indeed can be recorded from neurons, especially near apical dendrites, but these have orders of magnitudes lower amplitude than negative spikes originating from the soma or axon initial segment (Schomburg 2012, *J Nsci*)

This needs to be discussed in the paper in detail, and spike sorting quality metrics such as ISI violations and clustering metrics (e.g. isolation distance or nearest neighbor metrics) should be carefully presented in detail to confirm that the signals are indeed from well-isolated neuronal units. Clarifying these questions is necessary to interpret the data. As is, I am not fully convinced that the reported small amplitude events are unequivocally single units (I do realize that some other published studies with Behnke-Fried electrodes do not show compelling single-unit isolation either – but this should not be the measure to use here, in particular as the key point emphasized repeatedly in the paper is the recording of single units). Spike sorting needs to be discussed in much more detail as the Combinato method used is for example often thought to be prone to over-merging.

In this vein – while Behnke-Fried electrodes have a long history of use, they still need to be described in more detail concerning microwire properties (size, impedance, insulation) – e.g. in my experience microwires are often being trimmed manually before recording and this seems not to be mentioned in the text. Some visual detail, e.g. SEM images of the wires would help to create an appreciation of recording properties.

(2) The identification of odor responses in PCx and other areas is perfectly in line with various animal studies. While commendable to repeat those in humans they do not provide any new insight and I would recommend significantly reducing the description of those. The same holds true for the valence relationship of Amygdala SUA – which is very similar to what has been observed in animal models and in

lower-resolution recordings in humans. All these aspects could and should be condensed to give more room to the SUA validation (see above) and the analysis of the highly exciting multimodal responses.

(3) Multimodal / mental imagery responses: This to me is the key point where human recordings truly add unique value to our understanding of the brain – but as is it is analyzed and presented only quite briefly and buried in Figure 5.

The authors call the units they observe multimodal or concept units. Isn't it a possible interpretation that patients imagined smells when seeing a picture or reading a descriptor? Concept neurons have been described several times (Fried et al) and seeing them in different sensory modalities might not be considered to be fundamentally novel. However, olfaction is special in that verbal descriptors, mental imagery is significantly more in doubt – I think the authors are missing an opportunity to make this point (especially discussions in philosophy and imagery e.g. RICHARD J. STEVENSON and TREVOR I. CASE *Psychonomic Bulletin & Review* 2005, 12 (2), 244-264 Olfactory imagery: A review or Young, B.D. (2019) Olfactory imagery: is exactly what it smells like. *Philosophical Studies*. <https://doi.org/10.1007/s11098-019-01371-4> - both argue for olfactory imagery and olfaction not being a special sense whereas some other studies e.g. Crowder, R. G., & Schab, F. R. (1995). Imagery for odors. In R. G. Crowder & F. R. Schab (Eds.), *Memory for odors* (pp. 93–107). Mahwah, NJ: Erlbaum. OR Herz, R. S. (2000). Verbal coding in olfactory versus nonolfactory cognition. *Memory and Cognition*, 28, 957–964.

<https://doi.org/10.3758/BF03209343>.

arguing against that point)

My take is that the data presented here could go a long way towards answering this very fundamental question. Ideally, the authors would devise tasks where imagery is separable from direct sensory inputs. Knowing how laborious and logistically challenging these recordings are I definitely do not want to exclude that such information can be extracted from existing data.

Minor points:

(1) I am confused by the lack of correlation of Behavioral performance and Decoding accuracy anywhere but in Hippocampus – I'd expect from animal work to see significant correlations also e.g. in piriform. This could be a relevant finding – or due to e.g. different signal-to-noise from recordings in different brain areas? Generally, a discussion how to compare recording quality in different brain areas would be helpful – or tuning down the comparative analyses

(2) The suppression is potentially interesting as well – however, I am not sure the conclusion that this is due to central effects is valid. Repeated exposure to odors in animal models results in suppression for a variety of reasons – notably a change in sampling behavior. Did the authors measure inhalation / sniffing in the patients? As the authors surely know, changes in sniff patterns have a profound impact on neural activity across the olfactory pathway in e.g. mice and rats and novel odors result in increased sniff rate. Thus, this could be one of several contributing factors. At the least, this needs to be discussed, better addressed by sniff recordings as done routinely in humans by Hummels, Sobel and others

(3) Several statements in the text tend to ignore animal literature – for example “the single neuron basis of olfactory valence coding is unexplored” might be true for human work but ample animal work has addressed exactly this (reviewed e.g. in Pignatelli Beyeler *Curr Opin Behav Sci*. 2019 Apr; 26: 97–106.).

Moreover, Winston Gottfried, Nolan J Neurosci 2005 should be cited in the context of valence coding in general. The authors should more carefully consider the animal literature and assess more directly where their findings with putative SU in humans adds, contradicts, or complements the literature. This includes a more balanced perspective on the unimodal interpretation of piriform – my understanding of the rodent literature is that this is not the commonly held perspective anymore. To be fair, the authors do mention e.g. Poo et al and Schoorover et al – but the statement in the abstract that their work challenges “traditional views of a strictly unimodal (‘olfaction-only’) role” is overstated (the authors mention “human piriform cortex” but this is one example where after recent rodent work this perspective has also changed in the human olfactory neuroscience community). Again, this should not take away anything from the highly exciting findings of multimodal responses in piriform that might indicate cellular correlates of olfactory imagery but need to be phrased in a more balanced way.

(4) In my opinion, the authors overemphasize the importance of olfaction for human health and disease. The study as discussed above does have the potential to address fundamental questions in human neurophilosophy – so no need to overemphasize the health consequences of hyposmia.

Author Rebuttals to Initial Comments:

Reply to the Referees' Comments

We thank the three reviewers for their helpful and expert comments and respond below in detail to each of the points raised. We are also grateful to the editor for giving us the opportunity to respond with a revision. Based on the insightful comments from all three reviewers, we have extensively revised our paper and have added new experimental data and analyses (now shown in additional figure panels, i.e. new/revise Figs. 1c,d & 4h as well as Extended Data Figs. 1, 2, 4-9).

To address the Referees' comments, we now submit a revised manuscript that substantially extends our findings, as described below in a point-by-point reply (marked in **dark blue font**) to the reviews (in *italics*). Any changes to the manuscript text are denoted in **red font**.

Comments to Referee 1:

In this exciting manuscript, the authors record from populations of single neurons in humans. They focus primarily on the piriform cortex (PC), and compared these recordings with recordings in other medial temporal lobe (MTL) areas, including the amygdala, hippocampus, entorhinal cortex (EC), and parahippocampal cortex (PHC).

The central goal of sensory neuroscience is to relate neural activity to perception. The obvious power of this study is the ability to ask the subject what they are smelling and relate that directly to the spiking activity of populations of neurons in different parts of the brain.

The first three figures of this paper show nicely that the odor responses they recorded in humans are generally consistent with what people have observed in rodents. This alone is a very important contribution. However, the authors then really take advantage of recording in humans. Their major novel findings are:

- 1. Neurons in the amygdala fire more strongly for liked than disliked odors, suggesting that the amygdala is encoding valence. The authors are careful to state in the methods that valence is a loaded term that means different things to different people. I think their use of valence here is fine, with that proviso.*
- 2. They showed that accurate reporting, which they call "behavioral identification" correlates with increased decoding accuracy in the hippocampus only.*
- 3. All areas show visual responses, with better image decoding in the PC than the other areas. Although the idea of multimodal responses in any brain area is not hugely surprising, to me, that decoding visual responses is more accurate in the PC than the amygdala or hippocampus is both novel and very important.*
- 4. The authors made the remarkable observation that they can decode in PC and amygdala when training on image responses and testing on odor responses, and only in the amygdala when training on the odor responses and testing on the image responses.*

A strength of this manuscript is that the authors find that one area is better at one type of representation and another area is better at a different one. It is not just, for example, that they got better recordings in the PC, and therefore that that area was best at all decoding analyses. This serves as a good internal control and adds robustness to their findings.

Overall, this is a hugely important study that should be seen by the widest audience possible. It not only validates a decade of work in animals by showing similar results in humans, but it goes way beyond that because by registering single neuron spiking activity with direct perceptual reports, which cannot be done in animals, the authors have revealed a remarkable level of complexity PC activity, such as semantic representations. There is an impressively large dataset, the recordings appear of high quality, and the analyses and statistics are appropriate. I have only a few small points I would like clarified.

We thank the reviewer for their positive evaluation of the significance of our study and its broad relevance to biological science, as well as their instructive comments which have helped us to improve our manuscript.

1) Many of the areas the authors recorded from contain multiple functionally distinct subregions, e.g., “the amygdala” and “the hippocampus”. Obviously, recordings in humans entail less control than one has with animal experiments and there will likely be considerable variability across subjects. Fig. 1b notwithstanding, the authors should acknowledge and state this specifically.

We thank the reviewer for raising this point. We agree and, accordingly, have added the following statement in the Methods section (p. 14; ll. 21-25): “Electrode target locations were determined by clinical criteria and differed minimally within target regions across patients. This, along with the technical limitation of precisely localizing microwire positions, precluded us from targeting specific subregions, e.g., individual subnuclei of the amygdala or specific hippocampal subfields. Electrode placement was controlled by intraoperative CT scans co-registering the head-fixed frame to pre-operative MRI planning scans.”

2) Fig. 4g: Why are decoding accuracies so low in the PC and amygdala? The y axis values in Fig. 4g align with the decoding accuracies reported in Fig. 2b in all other areas, but they are much lower in the amygdala and especially in the PC.

The reviewer raises an important point that requires clarification. We apologize for the somewhat vague description. The decoding analysis in Fig. 4g was performed per recording session. For our main decoding analysis (Fig. 2b), an equal number of 200 neurons per region was randomly selected across recording sessions. Since fewer neurons were recorded in an individual session, we observed lower decoding accuracies (as expected). In the revised manuscript, we have changed the axis labels in Fig. 4g to highlight that decoding performance was measured for individual sessions. The y-axis now reads “Neuronal decoding accuracy per session [%]” and the x-axis label now reads “Behavioral odor identification performance per session [%]” (Reviewer Fig. R1.1).

Fig. R1.1 (Fig. 4g): Hippocampus predicts odor identification performance.

Accordingly, we also edited the corresponding caption, which now reads (p. 8; ll. 16-21): “Neuronal odor-decoding accuracy and behavioral odor-identification performance across regions and sessions (colored dots). Decoding accuracy in the hippocampus positively correlated with behavioral odor-identification performance across sessions (Spearman correlation, PC: $N=17$, $r=0.14$, $P=0.59$; Am: $N=27$, $r=0.06$, $P=0.75$; EC: $N=21$, $r=-0.12$, $P=0.62$; Hp: $N=27$, $r=0.50$, $P=0.0076$; PHC: $N=24$, $r=0.19$, $P=0.38$). Decoding was performed per recording session for sessions with at least 2 recorded neurons per region. Solid black lines represent linear regressions with shaded grey areas depicting 95%-confidence intervals.”

3) Other than the references to “banana” in one amygdala cell (L. 271) and “liquorice” one PC cell (L. 273), and Fig. 5g, there is no discussion of when or how the “words” or “names” were presented, or what the responses in any of the brain areas to the words. In the methods (L. 433-), they describe how they presented the images, but how when or how the words/names were presented. Please either elaborate (preferred) or eliminate (if necessary).

This is a valid point. We apologize for not properly explaining how words/names were presented. In our experiments, we used the written choice options presented during the odor identification task (Fig. 1e) as written stimuli. These odor-associated words (e.g., “banana”) were presented sequentially at four different screen positions at one-second intervals and remained on the screen. This allowed us to align neuronal activity to the onset of each new word. Within the 64 odor-identification trials per session, each odor-associated word served 4 times as the correct and 12 times as an incorrect choice option. Notably, to avoid confounding cueing effects of prior odor presentation, we excluded the 4 correct choice options from the word analysis shown in Fig. 5g & h. We have now added more detailed information to the description of the paradigm (p. 15, l. 40 - p. 16, l. 4): “Written odor names (labels) were selected pseudo-randomly from a list of the 15 odor stimuli plus the neutral, odorless control. Each odor label served 4 times as the correct and 12 times as an incorrect choice option. Name options were sequentially added at 1-second intervals, allowing stimulus-specific assessment of neuronal activity to individual written odor-associated words (Fig. 5g & h). To avoid confounding cueing effects induced by prior presentation of semantically matching odors, we excluded trials from the analysis, in which the odor word served as the correct choice option (Fig. 5g & h).”

4) The observation that there is a pronounced first-trial effect in the PCx but not in the other areas is important because it precludes a simple model in which odor information from the bulb is first received and processed in the PC, and then routed from the PC to the various MTL areas. If so, a large first-trial

effect would be expected in these areas too. Also, there is a fairly large first-trial effect in the zebrafish olfactory bulb of the Jacobson et al paper they cite. Is there also a first trial-effect in the human olfactory bulb? If so, then it is even more interesting that this is not seen in the other MTL regions. The authors do not mention this point but may want add a short discussion of this as I think provides important insight into the processing of olfactory information through the human brain.

We thank the reviewer for this insightful comment. We agree that the specificity of the first-trial effect has important implications for the parallel processing of olfactory information and have added this to our discussion (p. 12; ll. 14-15): “The absence of a first-trial effect in other downstream regions indicates that olfactory information is processed in parallel and not merely relayed through the PC.” While direct recordings from the human olfactory bulb to the best of our knowledge have never been performed, indirect measures using an Electrobulbogram (EBG), i.e. electrodes placed at the root of the nose, have been used to non-invasively record olfactory-bulb activity during odor habituation (Iravani et al., 2020). Here, the authors did not find decreased responses upon repeated odor stimulation, while parallel scalp EEG and perceptual reports were consistent with habituation. Similarly, recordings from the human olfactory epithelium within the nasal cavity – the earliest step of olfactory processing – also showed no response decrement with repeated odor stimulation although participants’ intensity ratings did (Mignot et al., 2022). These findings suggest that the first-trial effect emerges only after olfactory bulb processing in humans. Our result of a PC-specific first-trial effect suggests that the PC is the first central stage of odor response suppression, indicating rapid experience-dependent neural plasticity. We included this in our discussion of the first-trial effect (p. 12; ll. 14-18): “The apparent lack of habituation at the earliest stages of human olfactory processing - the olfactory epithelium (Mignot et al., 2022) and olfactory bulb (Iravani et al., 2020) - furthermore suggests that the first-trial effect emerges predominantly at the PC level.”

5) Related to this, did the authors include the first trials in their decoding analyses? They do not explicitly state that they did not. Given the large first-trial effect in PC, it may make sense to discard these trials in the decoding analyses, and this could increase decoding accuracy in PC. Some question whether this is fair, but most people, including me, agree that it is. For cross-model studies, in particular, discarding first trials is probably more correct because they always presented the odors before the images. As the authors are asking whether a particular image activates the same neurons that a semantically congruent odor would, the first odor trial should be omitted to compare visual- or word-evoked responses more accurately with the response they would have got had they presented an odor on that trial instead.

Following the reviewer’s suggestion, we repeated all decoding analyses after omitting the first odor trial. We successfully reproduced our decoding results with only slightly reduced decoding accuracies. We summarized the findings in the figure below (Fig. R1.2). Specifically, the most reliable and fastest odor decoding was still observed in PC and amygdala, followed by EC and hippocampus (Fig. R1.2a-c). Moreover, image-identity decoding remained more reliable in PC than in any of the MTL regions (Fig. R1.2d). Cross-modal coding also demonstrated overall consistent effects (Fig. R1.2e-f); however, the slightly reduced decoding accuracies led to a drop below statistical significance in the amygdala for odor-to-image decoding, given the conservative permutation test used here. These results are not unexpected given that first-trial omission reduced (i) the data set by ~12% (1 out of 8 trials) and (ii) the classifier training data by ~14% (1 out of 7). Nonetheless, decoding results are overall consistent,

highlighting the robustness of the effects observed. In our manuscript we now state (p. 17; ll. 15-18): “To ensure that our decoding results were not driven by systematic differences of the first compared to later trials, we repeated the decoding without the first trial and obtained overall consistent findings (Extended Data Fig. 9).”

Fig. R1.2 (Extended Data Fig. 9): Replication of the decoding analysis after excluding the first trial.

a, Odor-identity decoding accuracy based on neuronal activity separated by region. Each red dot in the distributions shows the decoding performance based on 200 randomly drawn neurons (1000 subsampling runs). Mean decoding performance and SEM across subsampling runs are shown in black. Grey dots indicate decoding performance on label-permuted data. The dashed horizontal line indicates chance level (6.25%). Significance estimated based on percentile of mean decoding performance of the real data in the surrogate distribution (PC: $P < 0.001$; Am: $P < 0.001$; EC: $P < 0.001$; Hp: $P < 0.001$; PHC: $P = 0.12$). **b**, Performance of odor-identity decoding (mean \pm SEM) as a function of the number of neurons included in the decoding analysis, using 100 subsampling runs. Horizontal bars indicate neuron counts for which decoding performance significantly exceeded chance ($P < 0.05$, right-sided Wilcoxon signed-rank against chance after Bonferroni correction for different neuron counts). **c**, Performance of odor-identity decoding (mean \pm SEM) as a function of the decoding time-window beginning at odor onset, using 200 randomly drawn neurons and 100 subsampling runs. Horizontal bars indicate times where decoding performance significantly exceeded chance ($P < 0.05$, right-sided Wilcoxon signed-rank against chance after Bonferroni correction for $N = 80$ decoding time windows; Beginning of sustained significant decoding: PC: 400 ms; Am: 350 ms; EC: 800 ms; Hp: 1050 ms; PHC: 1600 ms). **d**, Image-identity decoding accuracy based on neuronal activity separated by region, depicted as in (a). All regions exhibited significant decoding of image identities (PC: $P < 0.001$; Am: $P < 0.001$; EC: $P < 0.001$; Hp: $P < 0.001$; PHC: $P = 0.003$). Decoding accuracy in PC surpassed all other regions. **e-f**, Decoding accuracy (as in a) for a cross-modal decoding analysis trained on images and evaluated on odors (e), and *vice versa* (f). (Image-to-odor: PC: $P = 0.008$; Am: $P = 0.018$; EC: $P = 0.15$; Hp: $P = 0.1$; PHC: $P = 0.79$; Odor-to-image: PC: $P = 0.17$; Am: $P = 0.14$; EC: $P = 0.45$; Hp: $P = 0.28$; PHC: $P = 0.7$). PC and amygdala reached significantly higher decoding accuracies than any of the other regions in both cross-modal decoding analyses.

Minor points:

L. 215: In this paragraph the authors are writing about how levels single-neuron activity relate to identification. In L. 217-8, they specifically say “behavior identification”. In L. 215 they say

“identification performance”, which is vague. They should say “decoding accuracy” or something similar that makes clear that they are writing about classifier performance.

Reviewer 1 is correct to point out our inconsistent wording. Throughout the revised manuscript, we now consistently use the terms ‘behavioral identification performance’ and ‘neuronal decoding accuracy’ for behavioral and decoding results, respectively.

*L. 264: “... only generalized from images to odors, but not *visa versa*...” I found this phrase much more confusing than I should have and suggest being more explicit. e.g., “... only generalized when training on odors and testing on images, and not *visa versa*”.*

We fully agree and have changed this sentence (p. 9; ll. 23-25): *“Interestingly, in this analysis, identity decoding in PC only generalized when training on odors and testing on images, and not *vice versa*, whereas the amygdala exhibited cross-modal coding in both cases.”*

Comments to referee 2:

This manuscript by Kehl and colleagues presents single-neuron responses to odors in human olfactory (piriform cortex, PC; amygdala, Am) and medial temporal lobe structures (entorhinal cortex, EC; hippocampus, Hp; parahippocampal gyrus, PHC). Neurons in these regions (except PHC) are modulated by odors, respond stronger to odors versus no odors, suppressed by repeated stimulation, and encode odor identity as well as the identity of odor-related images. Whereas the activity of amygdala neurons correlates with odor valence, odor encoding in the hippocampus is related to odor identification performance.

I very much enjoyed reading this manuscript. It presents exciting data and is well written. The novelty and significance of the contribution lies in the systematic study of single-neuron responses to odors in humans. I am not aware of a comparable study. For the most part, the findings mirror and confirm results from previous animal studies and imaging work in humans, which is reassuring. However, the data are somewhat limited by shortcomings related to the odor delivery methods and the lack of nasal airflow measurements.

We thank the reviewer for the positive overall evaluation of the significance of our study. We have added extensive analyses to verify appropriate odor delivery as detailed below. Specifically, we have analyzed the respiratory belt data we recorded along with the neuronal data in about half of the sessions.

1) The odor delivery methods used here are not state-of-the-art. Odors were delivered in the form of (hand-held?) odorized pens. This may induce substantial variability in factors determining the amount of odor that is delivered on a given trial (e.g., length of exposure, distance to nose, etc.). Nasal inhalation was instructed verbally, and inhalation times were logged manually. This approach results variable delivery times and imprecise measurements. Well-controlled odor concentrations and precise delivery times are important for the analysis of the neuronal recording data. Ideally, odors are delivered using olfactometry and delivery times are determined based on measurements of nasal airflow.

Reviewer 2 brings up two important points, i.e., (i) stimulus delivery and (ii) measurement of respiration. We fully agree that both are critical issues. While the latter (“*measurements of nasal airflow*”) is discussed in detail in our response to concern #2 (see below), we appreciate the opportunity to comment here on our choice of odor-delivery method and the rationale behind it.

We apologize for not providing sufficient detail of the odor-delivery paradigm. We acknowledge, of course, that for most studies (in both humans and rodents) olfactometry is the state of the art. However, we deliberately opted for the use of odor pens as they offer several advantages in the unique clinical setting in which our experiments are performed. We believe that the benefits, which we detail below, outweigh the obvious disadvantages such as the lack of millisecond temporal control. A) Conducting research in a clinical environment poses unique challenges. This is particularly evident when working with epilepsy patients, who are confined to their hospital rooms and undergo 24/7 video surveillance and EEG monitoring. Accordingly, any experiments must accommodate to the patients’ clinical schedule and needs. Here, flexibility and minimal burden are key. Not only do we need to set up and perform experiments on short notice and within limited time windows, but we must also retain the patients’ willingness to participate, despite their highly vulnerable state. Thus, to

ensure that our research does not impose any undue burden, odor pens are ideal as they provide a minimally intrusive way of stimulus delivery. In fact, this way, most patients enjoyed participating.

B) A key feature of our study design is comparative analysis across stimulus space, requiring randomized stimulation with multiple (15 + 1) odors. While multi-channel olfactometers obviously exist, they are usually bulky, and their setup is time-consuming (calibration etc.), rendering them unfavorable for use in our clinical setting. Also, their electrical noise can interfere with the electrophysiological recordings because patient safety prohibits proper grounding of the electrophysiological amplifier. By contrast, odor pens offer a practical alternative by enabling time-efficient and intuitive odor delivery without the need for extensive setup, and do not interfere with the recordings. Other recent olfactory studies in epilepsy patients employed similar odor-delivery methods (i.e., hand-held squeeze bottles) (Yang et al., 2022; Zhou et al., 2019).

C) The odor pens used in our study are sourced from the Sniffin' Sticks odor identification test (Burghart Messtechnik, Wedel, Germany), which is a widely recognized and validated clinical test (see, e.g., (Oleszkiewicz et al., 2019) for a recent validation study with more than 9,000 subjects).

D) Given the obvious imprecisions inherent to the use of odor pens, we made an extra effort to improve temporal reproducibility in odor delivery. Upon odor presentation, patients were instructed to inhale just once. Moreover, the experimenter adhered to a standardized protocol, always presenting odor pens at approximately the same distance to the patient's nose and logging odor exposure via button press. Respiratory recordings during 13 of 27 sessions confirmed temporal alignment of odor delivery and inhalation (see response to point #2 below and Reviewer Fig. R2.1b-c). In addition, we note that none of our analyses address questions geared towards millisecond temporal resolution. Instead, we focus on longer time windows (e.g., 2-second periods for identification of odor-modulated neurons).

Overall, given (i) the unique clinical setting in which our study was conducted and (ii) the specific research objectives we pursue, odor pens provide a practical and effective means of stimulus delivery. Nonetheless, we acknowledge the method's limitations and, accordingly, we now explicitly state these in the revised manuscript (p. 15; ll. 30-33): "Standardized pen-like odor stimuli lack millisecond precision and exact control of odor concentrations that can be achieved with high-end olfactometers. This odor-delivery method, however, proved both efficient and effective for presenting a wide range of odor stimuli in the clinical environment." Furthermore, we now state in the discussion (p. 12; ll. 28-31): "As both odor intensity and valence have been shown to influence the response of the human amygdala (Anderson et al., 2003; Winston et al., 2005; Jin et al., 2015), future studies should also systematically vary odor intensity to investigate the interplay of valence and intensity coding. For this purpose, high-end olfactometers allowing for precise odor control will be essential."

Importantly, we recognize our original manuscript lacked sufficient detail in the description of the odor-delivery method. Accordingly, we have added a new section to the methods, which now reads (p. 15; ll. 13-19): "Odor pens were presented approximately 2 cm below the nose, centered between both nostrils. Patients were verbally instructed on each trial to inhale on command ('Please inhale NOW!'). To ensure consistent odor sampling across trials, participants were asked to inhale only once for each odor presentation and not sniff at their convenience. Odor pens were immediately removed after the first inhalation. This experimental protocol was devised to minimize odor-specific respiratory variability. The experimenter's (MSK) direct supervision ensured adherence to the instructions throughout the experiment. Pens were only opened immediately prior to odor exposure."

2. Another reason to measure nasal airflow is that it is a major driver of neural activity in olfactory areas in rodents (e.g., Fontanini et al. 2003, *J Neurosci*) and humans (e.g., Zelano et al 2016, *J Neurosci*; Sobel et al. 1998, *Nature*). Moreover, sniffing is rapidly modulated by odor pleasantness (e.g., Rozenkrantz et al. 2015, *Curr Biol*) and intensity (Johnson, et al 2003, *J Neurophysiol*). Thus, without accurately measuring and controlling for the effects of nasal airflow, we don't know whether the current results are driven by odor identity or odor-related differences in sniffing and nasal airflow.

We thank the reviewer for raising this critical point concerning the interplay between odor-specific and respiration-related effects. We agree that controlling for respiration is essential to interpret neural signatures of olfaction. Our experimental protocol was designed to ensure uniform breathing across trials, thus limiting odor sampling to a single inhalation (see point #1). Additionally, we recorded and analyzed respiratory signatures in 13 out of 27 sessions (Fig. R2.1). In line with other recent human olfactory studies (Bao et al., 2019; Bhutani et al., 2019; Sagar et al., 2023), respiration was measured with thoracic and abdominal inductive plethysmography belts (Fig. R2.1a). Respiratory signals from both belts were averaged and analyzed using the *Breathmetrics* toolbox (Noto et al., 2018). This revealed that inhalation onset closely aligned with the onset of odor delivery (Fig. R2.1c & R2.2). Results further confirmed that only a single inhalation occurred within the first 2 seconds (Fig. R2.1c & R2.2). Together, these data demonstrate both a high adherence to our experimental protocol and adequate temporal precision for our analyses.

Fig. R2.1 (Extended Data Fig. 5): Odor identity, not respiration, drives odor-modulated neurons.

a, Respiration was measured with thoracic (upper, green) and abdominal (lower, lilac) inductive plethysmography belts. Respiration signals were amplified and recorded using the Neuralynx ATLAS system, ensuring reliable temporal synchronization with neural recordings. **b**, Performance (adjusted R^2) of linear regression models, predicting neuronal firing (z-scores) based on respiration, odor identity, or both. Respiration (inhalation depth) alone failed to predict firing of odor-

modulated neurons above chance ($N=240$, $Z=-0.48$, $P=0.63$, Wilcoxon signed-rank across all odor-modulated neurons in recordings with respiratory monitoring). Odor identity predicted the firing of odor-modulated neurons significantly better than respiration (linear regression model for each odor-modulated unit to predict firing rates (z-scored): Respiration: $R^2=0.0040\pm 0.0013$; Odor identity: $R^2=0.19\pm 0.008$; Odor identity vs. respiration: $N=240$, $Z=13$, $P=1.2\cdot 10^{-40}$, Wilcoxon signed-rank). Adding respiratory information to odor identity did not significantly improved the model predictions of firing rates of odor-modulated neurons (odor identity & respiration: $R^2=0.19\pm 0.008$; odor identity & respiration vs. odor identity alone $N=240$, $Z=0.85$, $P=0.39$; odor identity & respiration vs. respiration: $N=240$, $Z=13$, $P=4.0\cdot 10^{-41}$, Wilcoxon signed-rank). Thus, odor-modulated neurons are primarily driven by odor-specific differences and not variations in respiration. **c**, Averaged odor-locked respiration across recording sessions (mean \pm SEM, 13 sessions). The initiation of inhalation was closely synchronized with odor presentation (dashed line). **c**, Odor-locked respiratory signals for each individual recording session. Participants consistently inhaled once (single peak) during the first 2 seconds after odor onset (gray shaded area), the analysis time window used for identification of odor-modulated neurons.

We furthermore added the mean respiratory trace across sessions (Fig. R2.2) as a panel to our outline of the experimental protocol in Fig. 1 and now state in the results (p. 4; ll. 8-9): “Respiratory measurements confirmed alignment of inhalation with odor presentation (Fig. 1d).”

Fig. R2.2 (Fig. 1d): Averaged odor-locked respiration.

Average respiratory depth (mean \pm SEM) demonstrating how single inhalations align to odor delivery ($N=13$ sessions, Extended Data Fig. 5).

We added the following description of these respiratory measurements to the methods section (p. 15; ll. 21-27): “In 13 of 27 recording sessions, respiration was measured using thoracic and abdominal plethysmography belts (Extended Data Fig. 5a; SleepSense, S.L.P. Scientific Laboratory Products, Elgin, IL). Data from both belts were averaged and analyzed using the *Breathmetrics* toolbox (Noto et al., 2018) (human respiratory belt default settings with sliding baseline correction). In the remaining 14 sessions, respiration belts could not be applied due to patient discomfort or noisy interference with the microwire recordings. Overall, participants complied accurately with the experimental protocol, inhaling once during odor exposure and well timed to odor delivery (Fig. 1d, Extended Data Fig. 5c).”

Following the reviewer’s comment, we asked whether the activity of odor-modulated neurons is driven by odor identity or by variations in breathing (Fig. R2.1b). To this end, we used linear regression models to predict the activity of odor-modulated neurons based on respiration (inhalation depth), chemical odor identity, and the combination of both. Strikingly, comparing the performance of these models (Fig. R2.1b) demonstrated that the activity of odor-modulated neurons is driven by odor-specific differences and not respiration. Adding respiration to the models yielded no significant improvement (Fig. R2.1b). We further validated our results by testing different measures of respiration

(mean inhalation, integrated absolute respiration, and time to peak inhalation). Odor identity predicted the firing of odor-modulated neurons significantly better than any of these measures, and none of these measures significantly improved performance in a combined regression model. We therefore conclude that odor-modulated neurons are primarily driven by odor identity and not variations in respiration. Accordingly, we added the following statement to our results (p. 4; ll. 35-36): **“Respiratory measurements confirmed that odor-modulated neurons were driven by odor-specific characteristics rather than by variability in respiration (Extended Data Fig. 5)”**

While our data firmly support the validity of our experimental protocol and odor delivery method, we acknowledge that direct measurements of nasal airflow is superior to plethysmography belts, allowing nostril-specific evaluations with high temporal precision. We therefore added the following statement to our discussion (p. 15; ll. 28-30): **“Bilateral measurements of nasal airflow will allow future studies to precisely explore interactions of neuronal activity and local oscillatory dynamics across the ipsilateral and contralateral hemispheres with high temporal resolution.”**

Importantly, while our results suggest that the activity of odor-modulated neurons is predominantly modulated by odor-specific features rather than variations of breathing, these findings do not preclude the important impact of respiration on neuronal activity (see, e.g., Fig. 1i). We would like to express our gratitude to the reviewer for raising this critical point. We included our findings in the revised manuscript (Fig. 1d & Extended Data Fig. 5).

3. The main decoding analysis (Figure 2) collapses across neurons recorded in different sessions and from different subjects. This paints an inflated picture of the information content available in individual brains. More appropriate would be a session-wise approach, in which decoding is based on neurons collected within an individual sessions and statistical tests are run across sessions (accounting for subjects). The authors already have decoding accuracies for individual sessions (Figure 4g) and it would be important use these throughout the manuscript.

We appreciate the reviewer’s thoughtful comments regarding our statistical analyses across neurons, subjects and sessions. Pooling neurons across session and conducting statistical analyses on these aggregated data is the widely accepted standard in the field of human single-unit recordings, e.g. (Hayat et al., 2022; Jamali et al., 2021; Kamiński et al., 2017; Kolibius et al., 2023; Kunz et al., 2021; Kutter et al., 2018; Minxha et al., 2020; Quian Quiroga et al., 2009; Rey et al., 2020; Rutishauser et al., 2010, 2015; Schonhaut et al., 2020; Tong et al., 2021). Likewise, decoding analyses are generally performed across a pseudo-population of neurons pooled across recordings, e.g. (Jamali et al., 2021; Kamiński et al., 2017; Minxha et al., 2020; Rutishauser et al., 2015; Zheng et al., 2022). Our approach of randomly subsampling equal numbers of neurons for each subregion ensures that any regional differences are determined by inherent features rather than differences in neuronal yield or exact electrode position within the anatomical target region. It is also worth noting that influential physiological rodent work in olfaction also pooled across recordings and utilized pseudo-populations for decoding (Bolding et al., 2020; Bolding & Franks, 2017; Iurilli & Datta, 2017; Miura et al., 2012; Poo et al., 2022; Roland et al., 2017).

We nevertheless acknowledge the reviewer's point and agree that it is informative to present individual contributions of each recording session and subject. Consequently, we have repeated the

decoding analysis for each recording session separately and assessed the decoding performance on a per-subject basis by averaging decoding accuracy across sessions (Fig. R2.3a-b). These analyses paint a picture that is strikingly consistent with our previous results. We added these figures to our manuscript (Extended Data Fig. 6) and now state (p. 5; ll. 6-7): "Odor-identity decoding was consistently observed in individual recording sessions and participants (Extended data Fig. 6a & b)." We also added the following paragraph to the methods section (p. 17; ll. 18-24): "The odor-decoding performance for each session was estimated based on all recorded neurons per region with a minimum of 2 neurons, using all odor presentations, 8 cross-validation data splits, and 1000 resample runs. For each session and region, a surrogate distribution was estimated by repeating the decoding analysis 1,000 times on odor-label-permuted data, using 10 resample runs each. The percentile of the actual decoding performance within this surrogate distribution was used to estimate *P* values. Decoding performances per participant were evaluated by averaging decoding performances across repeated sessions within anatomical target regions."

We thank the reviewer for pointing us in this direction, and we now include these controls as an Extended Data figure in our manuscript.

Fig. R2.3 (Extended Data Fig. 6 a-c): Decoding across individual recording sessions and participants.

a, Odor decoding performance per recording session and region. Note that the number of neurons varies between sessions and regions. Sessions with at least 2 neurons per region were included in this analysis. Despite the limited neuron count per session, odor identity could be decoded significantly above chance across sessions in PC, Am, EC, and Hp (PC: 14 out of 17 sessions showed significant decoding compared to odor-label permuted data, $P=3.3 \cdot 10^{-16}$; Am: 13 out of 27, $P=1.3 \cdot 10^{-10}$; EC: 5 out of 21, $P=0.0032$; Hp: 6 out of 27, $P=0.0019$; PHC: 1 out of 24, $P=0.71$; one-sided binomial test with $P_{chance}=0.05$). Chance level (6.25%) indicated by the horizontal dashed line (see also Fig. 2b). **b**, Decoding performance per participant and region.

Averaging the decoding performance across all sessions per participant demonstrated significant odor identity decoding in PC, Am, EC, and Hp (PC: 7 out of 9, $P=2.6 \cdot 10^{-8}$; Am: 10 out of 17, $P=1.4 \cdot 10^{-9}$; EC: 4 out of 15, $P=0.0055$; Hp: 3 out of 17, $P=0.05$; PHC: 0 out of 16, $P=1$; one-sided binomial test with $P_{chance}=0.05$). See also Fig. 2b. **c**, Odor-decoding accuracy and behavioral odor-identification performance across regions and participants, averaged across sessions for each participant (colored dots). Decoding accuracy in the hippocampus positively correlated with odor identification performance across participants (Spearman correlation, PC: $N=9$, $r=0.15$, $P=0.71$; Am: $N=17$, $r=0.10$, $P=0.71$; EC: $N=15$, $r=0.01$, $P=0.96$; Hp: $N=17$, $r=0.50$, $P=0.043$; PHC: $N=16$, $r=0.15$, $P=0.58$). Solid black lines represent linear regressions with shaded grey areas depicting 95%-confidence intervals.

4. Odor-modulated neurons in Figure 1g are defined based on comparing firing rates between 16 odors. Given this includes the neutral odor, and responses are substantially higher for odor vs neutral, it would be interesting to see the percentage of neurons that are significantly modulated by the 15 actual odors (i.e., excluding the neutral condition)

The reviewer raises an important point. We agree that excluding the neutral stimulus adds an important control. Following the reviewer's advice, we repeated the analysis without the odorless/neutral stimulus. In agreement with our previous findings, these new results demonstrate significant fractions of odor-modulated neurons across PC, amygdala, EC, and hippocampus (Fig. R2.4b). Striking overlap of odor-modulated neurons identified with and without the neutral odor (Fig. R2.4b-c) indicates that activity is not driven by the contrast between odors and odorless controls. We now include these novel controls (Fig. R2.4) in our manuscript (Extended Data Fig. 4) and state "Odor-modulated neurons were likewise identified after excluding the odorless control (Extended Data Fig. 4b & c)." in the results (p. 4; ll. 34-35).

Fig. R2.4 (Extended Data Fig. 4): Odor-modulated neurons are reliably identified across participants and without odorless controls.

a, Same as Fig. 1h, but averaged across recording sessions per participant. Proportions of odor-modulated neurons (mean \pm SEM) across regions for each participant. Significant proportions of odor-modulated neurons were found in PC, Am, EC and Hp across participants (PC: $39.7 \pm 6\%$, $N=9$, $P=0.002$; Am: $19.8 \pm 3.4\%$, $N=17$, $P=0.00033$; EC: $14.2 \pm 3.6\%$, $N=15$, $P=0.027$; Hp: $10.9 \pm 1.9\%$, $N=17$, $P=0.0043$; PHC: $5.14 \pm 1.8\%$, $N=17$, $P=0.79$; one-sided Wilcoxon signed-rank against chance). Chance level (5%) indicated by the horizontal dashed line (see also Fig. 1h). **b**, Same as Fig. 1h, but excluding the odorless control. Distribution of odor-modulated neurons after omitting the neutral odor stimuli for the definition of odor-modulated neurons (PC: $36.8 \pm 4.2\%$, $N=17$, $P=0.00016$; Am: $18.7 \pm 2.6\%$, $N=27$, $P=1.2 \cdot 10^{-5}$; EC: $13.9 \pm 3\%$, $N=22$, $P=0.017$; Hp: $9.83 \pm 1.8\%$, $N=27$, $P=0.011$; PHC: $8.61 \pm 3.9\%$, $N=26$, $P=0.42$; one-sided Wilcoxon signed-rank against chance). **c**, Population of odor-modulated neurons identified with and without the odorless control showed a highly significant overlap ($P < 10^{-100}$ in a two-sided binomial test with $k=353$, $N=2,416$ and $P_{\text{chance}}=(406/2,416) \cdot (378/2,416)$).

5. The authors appear surprised by the finding that PC neurons respond to images associated with odors (e.g., in the abstract: “Unexpectedly, we uncover that piriform cortex neurons also encode image identities challenging traditional views of a strictly unimodal (‘olfaction-only’) role of the human piriform cortex”). I don’t think these findings are all that surprising given previous work across species. For instance, the olfactory bulb and PC in rodents respond to visual stimuli associated with odors (e.g., Mandairon et al 2014, *Front Behav Neurosci*), and the human PC responds to visual cues that predict odors (Schultz et al 2017, *Scientific Reports*), odor-associated words (Gonzales et al 2006, *Neuroimage*) and imagined odors (e.g., Djordjevic et al. 2006, *Neuroimage*). Moreover, given that images (and words) used in the current study were strongly associated with specific odors, I also don’t think the findings provide evidence that PC processes non-olfactory information. For this, it would be necessary to show that PC neurons respond to images of odorless objects.

We agree that prior studies in rodents and human imaging studies are also pointing towards multimodal representations in the piriform cortex. We apologize for not stating this more clearly and have carefully revised our manuscript.

The abstract now states (p. 1; ll. 27-29): "Critically, we uncover that piriform cortex neurons reliably encode odor-related images, supporting a multimodal role of the human piriform cortex."

The results now read as follows (p. 9; ll. 17-19): "Strikingly, neuronal activity in PC predicted odor-related image identity more accurately than in any of the MTL regions, demonstrating that human PC neurons are not exclusively driven by olfaction, but also encode information from other sensory modalities." Moreover, we added (p. 9; ll. 36-38): "Collectively, our findings reveal encoding of odor-related visual information in human PC neurons, as well as multimodal odor representations in the human amygdala and PC."

We also revised our discussion (p. 13; ll. 4-12): "Recent rodent studies have shown that neurons in the posterior PC precisely encode spatial information, suggesting a role in odor-place association (Poo et al., 2022). Further evidence for multimodal processing of odor-related information in the PC stems from rodents (Mandairon et al., 2014) and human Imaging studies (Djordjevic et al., 2008; González et al., 2006; Schulze et al., 2017). Here, we tested semantically coherent olfactory and visual stimuli to explore coding of PC neurons beyond olfactory perception. We discovered that PC neurons not only decode odors, but also odor-related image identities. The PC therefore not only processes olfactory stimuli, but also integrates top-down semantic information from higher cognitive areas. Most strikingly, odor-related images were decoded more accurately in the PC than in the MTL. Future research will need to explore whether PC neurons specifically encode odor-related images, or whether they also process images of odorless objects."

6. There is considerable across-subject variability in odor perception (e.g., Mainland et al. 2014, Nature Neuroscience; Keller et al. 2017, Science) and associated neural responses (e.g., Sagar et al. 2023, Nature Neuroscience). Averaging valence ratings and identification performance per odor across participants (e.g., Figure 1e) and relating averages to neural activity (e.g., Figure 4e) is thus not very informative. It would be important to present and analyze behavioral and neural data for individual subjects.

We acknowledge the across-subject variability in odor perception, and that it would be important to provide additional details on individuals' behavioral measures (i.e., individuals' odor valence ratings and odor identification performance). We therefore now include both odor ratings and odor identification performances for each subject and odor in our study (Fig. R2.5). While we note some inter-subject variability, these results also demonstrate a reasonable degree of inter-subject consistency (e.g., valence ratings for fish and garlic or correct identification of coffee).

Fig. R2.5 (Extended Data Fig. 2): Odor valence ratings and identification performance for each participant.

a, Mean odor ratings for each participant and odor. Odors are sorted from most to least liked (left to right) and participants are organized by average valence ratings (top to bottom). **b**, Average behavioral odor identification performance for each participant and odor. Odors are sorted from most to the least accurately identified (left to right) and participants are organized by their mean identification performance (top to bottom).

We recognize the importance of across-subject variability in odor perception when linking neuronal activity to valence rating. Our main analysis in Fig. 4d accounts for this across-subject variability and reveals a distinct role of amygdala neurons in individual valence coding. Based on this, we next sought to determine whether the population activity in the amygdala reflects standardized valence ratings. Our study only involved binary valence ratings (liked vs. disliked), therefore we utilized more quantitative valence ratings for the same odor panel (Toet et al., 2020). Despite individual differences in odor perception, we observed a significant correlation between amygdala activity and population-wide valence ratings (Fig. 4e). This finding integrates well with prior research demonstrating that population-wide odor valence ratings can be predicted based on molecular features (Keller et al., 2017).

We apologize for not stating this clearly in the original manuscript and have changed the result section accordingly (p. 7; ll. 11-14): "Utilizing published valence ratings (Toet et al., 2020) of the standardized odors used in our study, we sought to correlate general valence ratings with the activity of odor-modulated neurons in the amygdala. Here, we found a significant correlation of firing rate with valence across recordings (Fig. 4e)."

A correlation between quantitative valence ratings of the odors (Toet et al., 2020) and firing rates of odor-modulated neurons in the amygdala was found in both sessions (6 out of 27, $P = 0.002$; two-sided binomial test) and participants (4 out of 17, $P = 0.009$). We added these novel findings to our manuscript (p. 8; ll. 13-15): "This correlation was observed in a significant number of sessions (6 of 27, $P=0.002$) and participants (4 of 17, $P=0.009$, one-sided binomial test with $P_{chance}=0.05$)."

7. Many of the statistical analyses are run across sessions (n=27, e.g., Figure 1g, Figure 4g), and it is unclear how the authors controlled for non-independent data points (sessions from the same participants). Alternatively, the authors could average across sessions per subject and then run statistical analyses across subjects.

We appreciate the reviewer pointing out the potential issue of non-independence across multiple sessions for the same individuals. Human single-neuron recordings are highly sensitive to the precise electrode position relative to the recorded neurons (Gold et al., 2006). Microscopic brain movements (e.g., pulse or motion related) contribute to microscopic electrode drifts across time. These drifts pose a major challenge for spike sorting in human single-unit data (Harris et al., 2016). It is therefore extremely difficult to record individual neurons over multiple hours in humans (Chaure & Rey, 2020; Niediek et al., 2016). Consequently, neurons recorded on separate days are generally treated as independent in the human single-unit literature (Hayat et al., 2022; Jamali et al., 2021; Kamiński et al., 2017; Kolibius et al., 2023; Kunz et al., 2021; Kutter et al., 2018; Minxha et al., 2020; Quian Quiroga et al., 2009; Rey et al., 2020; Rutishauser et al., 2010, 2015; Schonhaut et al., 2020; Tong et al., 2021). We followed this standard in the field. The time between sessions from a given patient was on average 5.7 ± 1.2 days, which makes it even more likely that recordings originate from different neurons across sessions.

Nonetheless, we acknowledge the reviewer's point and the value of demonstrating our findings on a per-subject basis. Following the reviewer's suggestion, we averaged across sessions for each subject. The results consistently reproduce the across-session findings (see Fig. R2.3 a & c), highlighting their robustness. Accordingly, we have added the following statement to our results (p. 4; ll. 22-23): "**Odor-modulated neurons were reliably identified across participants (Extended Data Fig. 4a).**". We have also included the following statement in the results (p. 7; ll. 22-25): "**We correlated odor-decoding accuracy in each recording session and region with behavioral odor-identification performance, and discovered a significant positive correlation exclusively in the hippocampus (Fig. 4g). This correlation was consistently observed across participants (Extended data Fig. 6c).**" We thank the reviewer for raising this point.

8. Other statistical analyses are computed across neurons (e.g., Figure 1h, Figure 3b, Figure 4d) by collapsing across subjects and sessions. Again, neurons from the same subject/session are not independent observations, and it would be important to rather use mixed effects models that control for effects of sessions and subjects.

As before, the reviewer addresses a fundamental challenge in human single neuron recordings. In human single-neuron studies, neurons recorded in a given patient across separate sessions are generally treated as independent and analyzed collectively, e.g., (Hayat et al., 2022; Jamali et al., 2021; Kamiński et al., 2017; Kolibius et al., 2023; Kunz et al., 2021; Kutter et al., 2018; Minxha et al., 2020; Quian Quiroga et al., 2009; Rey et al., 2020; Rutishauser et al., 2010, 2015; Schonhaut et al., 2020; Tong et al., 2021). Although this standard procedure seems reasonable due to electrode drifts, it would be preferable to consider the effects of recordings within and across subjects in our analysis. Following the reviewer's recommendation, we repeated our neuronal population analyses (Fig. 1i, 3b & 4d), leveraging generalized linear mixed-effects models to predict neuronal activity (Table R2.1). These models were applied to investigate the effects of odors vs. odorless controls (Fig. 1i), stimulus repetition (Fig. 3b), and subjective valence ratings (Fig. 4d). Critically, we included subject and session

as random effects to account for their inherently nested structure. We added the description of these models to our methods section (p. 18; ll. 12-24):

“Mixed-effects models

Generalized linear mixed-effects models (GLMMs) were used to control for recordings across multiple sessions within and across participants. A GLMM was utilized for each fixed effect to predict trial-wise spike counts of odor-modulated neurons using MATLAB's 'fitglme' function. Brain regions and interactions were incorporated as fixed effects. Participant identity and recording session per participant were included as random effects to account for their nested hierarchical nature (Aarts et al., 2014). Each fixed-effects regressor was incorporated as a random slope for both participant identity and participant-session nesting, and neuron identity was included with an individual intercept to account for participant-session-neuron nesting (Barr et al., 2013). All random effects comprised an individual intercept. Likelihood Ratio Tests (MATLAB's 'compare' function) confirmed that the full models we used with both random slopes and intercepts outperformed models incorporating only random intercepts. Poisson models were fitted based on the restricted maximum pseudo likelihood with a logarithmic link function.“

The results of these GLMMs, demonstrated in Table R2.1, validate our findings:

- Mirroring data presented in Fig. 1i, odorless controls elicit weaker neuronal firing than actual odor stimuli in the PC, Am, EC and Hp (Table R2.1a). We thus added the following sentence to the results (p. 4; ll. 32-34): “Increased firing rates for odors compared to odorless controls were consistently observed when accounting for participants and sessions (Supplementary Table S1a).“
- The effect of repetition suppression with repeated odor exposure (Fig. 3b) was reproduced by the mixed-effects model (Table R2.1b), with the effect in EC also reaching statistical significance. We therefore included the following sentence (p. 6; ll. 19-20): “Repetition suppression was reliably found when factoring in individual participants and sessions (Supplementary Table S1b).“
- Individual valence ratings in the odor-rating task (Fig. 4d) specifically influenced the activity of amygdala neurons (Table R2.1c). While disliked odors yielded an overall decrease in firing, amygdala neurons specifically showed an increased firing for liked odors. We therefore added the following statement to our results (p. 7; ll. 10-12): “Increased activity of odor-modulated neurons in the amygdala in response to liked odors was also evident when correcting for session and participant-specific differences (Supplementary Table S1c).“

Together, when accounting for subjects and sessions, these additional results highlight the robustness of our findings. We thank the reviewer for suggesting this statistical analysis, which we now added to our manuscript. In light of the established standards in the field of human single unit recordings, we propose to keep our original figures as main display items and add references to these controls in the corresponding sections as outlined above.

Comparison of odors and odorless controls

Fixed Effect	Estimate	Std. Error	t-value	p-value	95%-CI
Intercept	0.14	0.34	0.43	0.67	[-0.51, 0.80]
Region-PC	0.59	0.38	1.56	0.12	[-0.15, 1.32]
Region-Am	0.55	0.45	1.23	0.22	[-0.33, 1.44]
Region-EC	0.96	0.48	2.01	0.04	[0.02, 1.90]
Region-Hp	0.47	0.52	0.92	0.36	[-0.54, 1.48]
control	0.13	0.05	2.54	0.01	[0.03, 0.23]
Region-PC × Control	-0.46	0.05	-10.00	>0.0001	[-0.55, -0.37]
Region-Am × Control	-0.20	0.04	-4.75	>0.0001	[-0.29, -0.12]
Region-EC × Control	-0.21	0.05	-4.56	>0.0001	[-0.30, -0.12]
Region-Hp × Control	-0.23	0.05	-5.04	>0.0001	[-0.32, -0.14]

Odor repetition suppression

Fixed Effect	Estimate	Std. Error	t-value	p-value	95%-CI
Intercept	0.08	0.34	0.24	0.81	[-0.58, 0.74]
Region-PC	0.85	0.38	2.24	0.03	[0.11, 1.60]
Region-Am	0.68	0.47	1.44	0.15	[-0.25, 1.61]
Region-EC	1.03	0.49	2.10	0.04	[0.07, 1.99]
Region-Hp	0.59	0.54	1.10	0.27	[-0.46, 1.65]
Rep	0.02	0.01	2.63	0.01	[0.00, 0.03]
Region-PC × Rep	-0.06	0.005	-12.44	>0.0001	[-0.07, -0.05]
Region-Am × Rep	-0.04	0.005	-8.14	>0.0001	[-0.05, -0.03]
Region-EC × Rep	-0.03	0.005	-6.16	>0.0001	[-0.04, -0.02]
Region-Hp × Rep	-0.04	0.005	-7.72	>0.0001	[-0.05, -0.03]

Subjective valence coding

Fixed Effect	Estimate	Std. Error	t-value	p-value	95%-CI
Intercept	0.03	0.36	0.07	0.94	[-0.67, 0.72]
Region-PC	0.71	0.38	1.87	0.06	[-0.04, 1.46]
Region-Am	0.76	0.43	1.77	0.08	[-0.08, 1.60]
Region-EC	1.06	0.48	2.19	0.03	[0.11, 2.00]
Region-Hp	0.66	0.48	1.37	0.17	[-0.29, 1.61]
liked	-0.13	0.05	-2.34	0.02	[-0.23, -0.02]
Region-PC × Liked	0.06	0.03	1.94	0.052	[0.00, 0.13]
Region-Am × Liked	0.15	0.03	4.63	>0.0001	[0.08, 0.21]
Region-EC × Liked	0.04	0.03	1.26	0.21	[-0.02, 0.11]
Region-Hp × Liked	0.01	0.03	0.33	0.74	[-0.05, 0.08]

Table R2.1 (Supplementary Table 1): Generalized linear mixed-effects models across participants and sessions.

Results of general linear mixed-effects models (GLMMs) for firing rates of odor-modulated neurons in response to odors as compared to odorless controls (top), repeated odor presentations (middle), and subjective valence ratings (bottom). Models include different brain regions and account for neurons recorded across participants and sessions as well as their nested structure. **a**, Odors elicited significantly stronger activity of odor-modulated neurons than odorless controls in the PC, Am and Hp across patients and sessions. (GLMM predicting spike counts (SC) based on odor vs. odorless control, brain region, and their interaction. Model: $SC \sim 1 + \text{Control} \times \text{Region} + (\text{Control}|\text{ParticipantID}) + (\text{Region}|\text{ParticipantID}) + (\text{control}|\text{ParticipantID:SessionID}) + (\text{Region}|\text{ParticipantID:SessionID}) + (1|\text{ParticipantID:SessionID:UnitID})$, using the odor condition and PHC as reference). **b**, Repeated odor presentations led to reduced firing of odor-modulated neurons specifically in the PC, Am, EC and Hp across patients and sessions. (GLMM predicting spike counts for repeated odor presentations (Rep), brain region and their interaction. Model: $SC \sim 1 + \text{Rep} \times \text{Region} + (\text{Rep}|\text{ParticipantID}) + (\text{Region}|\text{ParticipantID}) + (\text{Rep}|\text{ParticipantID:SessionID}) + (\text{Region}|\text{ParticipantID:SessionID}) + (1|\text{ParticipantID:SessionID:UnitID})$, using PHC as reference). **c**, Behavioral valence ratings predicted firing of odor-modulated neurons especially in the amygdala across patients and sessions (GLMM predicting spike counts based on valence (liked vs disliked), region and their interaction. Model: $SC \sim 1 + \text{Liked} \times \text{Region} + (\text{Liked}|\text{ParticipantID}) + (\text{Region}|\text{ParticipantID}) + (\text{Liked}|\text{ParticipantID:SessionID}) + (\text{Region}|\text{ParticipantID:SessionID}) + (1|\text{ParticipantID:SessionID:UnitID})$, with disliked and PHC as reference).

Minor comments:

1. The authors state that olfactory bulb projects to the EC (line 48). Unlike in rodents, direct projections from the olfactory bulb to EC are rare in non-human primates (Carmichael et al. 1994, *J Comp Neurol*) and seemingly absent in humans (Allison 1954, *J Anat*; Echevarria-Cooper et al. 2022, *J Neurosci*). The authors may want to revise the introduction and adjust their diagram in Figure 1a. Moreover, I think their finding of a 500 ms delay between odor encoding in PC and EC (Figure 2d) may provide interesting electrophysiological support for the lack of a direct projection from the bulb to EC in humans.

We are thankful to the reviewer for raising this intriguing point. We adjusted the diagram in Fig. 1a using a dashed line to point towards the uncertainty of OB-to-EC projections in humans. Moreover, acknowledging the reviewer's argument, we now mention this point in the revised introduction, which now reads (p. 2; ll. 9-10): "Direct projections to the EC are established in rodents (Gretenkord et al., 2019; Sosulski et al., 2011) but have not yet been confirmed in humans (Allison, 1954; Echevarria-Cooper et al., 2022)." We have also added the following statement to the discussion (p. 11; ll. 35-38): "Neural representations of odors emerged first in the PC and amygdala and only approximately 500 ms later in EC. This delay in EC odor coding could support a connectivity scheme in which, unlike in rodents (Gretenkord et al., 2019; Sosulski et al., 2011), human olfactory bulb mitral cells might not directly project to the EC."

2. How can we interpret the sparseness of a neural population if this population does not encode information about a stimulus? Specially, what does it mean that the sparseness of PHC is between PC and amygdala although PHC does not encode odors?

Good point. While population sparseness can be calculated for any neuronal population and stimuli, it only yields meaningful results in populations encoding the stimulus identity. We therefore excluded the PHC from the sparseness analysis (Fig. R2.6) and thank the reviewer for raising this point.

Fig. R2.6 (Fig. 3 a): Odor representations vary in sparseness.

a, Population sparseness index for each of the 15 odors across regions containing odor-modulated neurons (odors color-coded as in Fig.1, mean ± SEM in black). Sparseness of odor coding significantly differed across regions (one-way ANOVA, $F(3,56)=505$, $P=1.7 \cdot 10^{-40}$). PC exhibited a less sparse odor code than MTL regions (all pairwise comparisons $P < 0.01$ except Am vs. Hp $P=0.46$, following Tukey's honestly significant difference procedure (Tukey, 1949)).

3. The authors cite two navigation studies (Poo et al. 2009, *Nature*; Raithel et al. 2023, *Curr Biol*) as evidence for non-olfactory processing in olfactory cortex. Because these studies focused on odor-based navigation, I don't think they support the authors' argument. Rather, they show that PC represents

odor-associated spatial information, which is in line with the current (and previous) findings of PC responding to odor-associated visual information.

We acknowledge that these studies focus on odor-related processes. Therefore, we have removed the corresponding sentence from our introduction: ~~Recent evidence suggested PC involvement in non-olfactory processes such as spatial navigation (Poo et al., 2022; Raithel et al., 2023).~~ Additionally, we have adapted our discussion to include this aspect as detailed in our response to the reviewer's major point #5 (see above).

Comments to referee 3:

In Kehl et al the authors present intracranial recordings from patients in olfactory and related areas. Single-unit recordings in humans – despite a lot of recent developments – are still incredibly rare and a unique opportunity to explore mechanistic aspects of exciting human-specific questions.

Kehl et al use quite established “Behnke-Fried” electrodes where a lead of macroelectrodes is implanted as indicated for clinical diagnostics. After implantation a central guide is withdrawn and a set of 8 microelectrode wires inserted through the central space for neuroscience investigation beyond the tips. These small (“high impedance”) wires offer the opportunity to record localized neural activity and potentially multi-unit activity (MUA) or single-unit activity (SUA). Prior work in human olfactory areas has been largely based on functional imaging (e.g. the Sobel and Gottfried labs) or using microelectrode field potential work (including recent work by the Gottfried lab as cited in the manuscript). To my knowledge, no SUA has been reported in PCX – although many studies pioneered by Fried and many others have used similar opportunities in patients with pharmacologically intractable epilepsy to record in e.g. hippocampal or various MTL areas with non-olfactory stimuli. This includes e.g. the work by Fried and colleagues describing concept neurons, i.e. neurons that respond to specific concepts rather than visual or other stimuli (famously to celebrities as images, texts, in different constellations etc).

Two aspects are truly novel about the work by Kehl et al.

Firstly, this might be the first SUA recording in humans with olfactory stimuli (and in olfactory areas). Secondly, the study offers the opportunity to provide a major puzzle piece to the long-standing question of whether olfaction is a “special sense”, whether olfactory imagery exists and how olfaction interacts with other senses (that are more verbal in humans, more consciously processed, relayed by thalamus).

While the SUA aspect might not on its own constitute a landmark (a number of SUA recordings in humans have been presented in the literature under slightly different conditions), the second point has the potential to be truly a fundamental and transformative one.

I am thus overall very excited about the study.

We thank the reviewer for the positive evaluation of the significance and potential of our study, as well as their instructive comments. Based on the reviewer’s suggestions, we improved our description of the recording techniques, verified our spike-sorting techniques, and further explored the properties of human odor representation, revealing regional variations in chemical vs. perceived odor representations.

However, this excitement is dampened by a few major concerns:

(1) Critical evaluation of spike identification and spike sorting: The SUA aspect is critical (not only in its own right but also to provide certainty that e.g. single neurons can respond to odors and multimodal stimuli / imagined odors (see below). Here the authors fall short in detail and critical assessment. The example units displayed in the figures exclusively show positive or bipolar characteristics and appear to have slower waveforms compared to the vast majority of published recordings, e.g. from cortical areas in human patients (Chung, 2022 Neuron) and piriform cortex in rodents (Schoonover 2021, Nature). Positive spikes indeed can be recorded from neurons, especially near apical dendrites, but these have orders of magnitudes lower amplitude than negative spikes originating from the soma or axon initial segment (Schomburg 2012, J Nsci) This needs to be discussed in the paper in detail, and spike sorting quality metrics such as ISI violations and clustering metrics (e.g. isolation distance or

nearest neighbor metrics) should be carefully presented in detail to confirm that the signals are indeed from well-isolated neuronal units. Clarifying these questions is necessary to interpret the data. As is, I am not fully convinced that the reported small amplitude events are unequivocally single units (I do realize that some other published studies with Behnke-Fried electrodes do not show compelling single-unit isolation either – but this should not be the measure to use here, in particular as the key point emphasized repeatedly in the paper is the recording of single units). Spike sorting needs to be discussed in much more detail as the Combinato method used is for example often thought to be prone to over-merging. In this vein – while Behnke-Fried electrodes have a long history of use, they still need to be described in more detail concerning microwire properties (size, impedance, insulation) – e.g. in my experience microwires are often being trimmed manually before recording and this seems not to be mentioned in the text. Some visual detail, e.g. SEM images of the wires would help to create an appreciation of recording properties.

We thank the reviewer for their detailed comments. We agree that our original manuscript was lacking detail in the description of recording techniques and data processing. Accordingly, we now provide detailed information on (i) spike polarity, (ii) spike-sorting quality metrics, and (iii) recording techniques.

Spike polarity: We sincerely apologize that our manuscript led the reviewer to assume we recorded positive spike deflections. We displayed all spike waveforms with inverted polarity, adhering to visualization standards established by the vast majority of publications in the field of human single-unit recordings, including numerous recent reports from other groups, e.g., (Kamiński et al., 2020; Kolibius et al., 2023; Umbach et al., 2020; Zheng et al., 2022), see also Reviewer Fig. R3.1). We regret to not have explicitly mentioned this and have now added the missing information to the Methods section (p. 14; ll. 36-37): **“Spikes of negative voltage deflection were extracted and analyzed. For illustration, spikes are depicted with inverted polarity.”** Additionally, we added the following sentence to the caption of Fig. 1 (p. 3; l. 19): **“Spike polarity inverted for visualization.”**

Spike shapes and spike-sorting quality

As pointed out by the reviewer, many human single-unit publications fail to provide details on spike sorting and spike waveforms. We agree that such details are important for validating recording quality. When comparing data quality among published datasets, it is important to use appropriate reference standards, which obviously differ, e.g., between Behnke-Fried monotrodes and Neuropixel high-density probes. Signals are additionally affected by differences in electrode geometry, scale and referencing, and exact spike shapes relay on filters applied during preprocessing. Importantly, electrodes with smaller surfaces record larger action potentials (Viswam et al., 2019). Note that the waveforms displayed in our figures are the filtered waveforms according to the standards usually used for Behnke-Fried electrodes. A comparison of spike shapes reported in our manuscript with those from other recent studies targeting the human MTL (Kamiński et al., 2020; Kunz et al., 2021; Umbach et al., 2020; Zheng et al., 2022) (Fig. R3.1), demonstrates overall consistent spike features (e.g., spike shape, amplitude, and width).

Reviewer Fig. R3.1: Example spike shapes in human MTL neurons.

Overview of the 13 units depicted in our manuscript along with representative examples of spike shapes recorded with Behnke-Fried electrodes in the human MTL reported in recent publications by other groups. Based on these published waveforms, our recording quality matches that of other groups recording in similar brain regions.

Although spike-sorting quality metrics are not reported in the majority of human single-unit studies, we acknowledge the importance of describing and critically evaluating our recording quality. Therefore, we now include a comprehensive overview of our data and spike-sorting quality metrics in the revised manuscript (Fig. R3.2). The methods now state (p. 15; ll. 2-4): **“Single-unit recording quality and spike sorting was validated based on ISI violations, spike amplitudes and spike peak signal-to-noise, as well as cluster isolation distance (Extended Data Fig. 8).”**

The unit yield in our recordings aligns well with other human single-unit datasets (e.g., (Cao et al., 2022; Faraut et al., 2018)). Importantly, ISI violations were extremely rare in our dataset ($0.36 \pm 0.01\%$ ISIs shorter than 3 ms across all units), indicating well-separated neuronal activity. Additionally, the peak signal-to-noise amplitudes of our spikes not only match but in many cases surpass those reported in the human MTL literature (Cao et al., 2022; Faraut et al., 2018; Kamiński et al., 2020; Kunz et al., 2021), indicating high recording quality. Isolation distances are in line with other human single-unit studies using Behnke-Fried electrodes. Together, these results demonstrate the robustness of our recording and spike-sorting quality. We thank the reviewer for raising this issue as we believe that addressing this point has helped us generate a stronger manuscript.

On a side note, in our experience Combinato (used with default parameters as done in this study) tends to overcluster the recorded unit data in automated mode, which is why manual merging based on waveform and other firing characteristics is typically necessary during manual curation. We have added this to the methods (p. 14; l. 39 - p. 15; l. 2): **“Since Combinato (used with default parameters in this study) tends to overcluster the recorded unit data in automated mode, we manually merged clusters based on their waveforms, cross correlograms, and other firing characteristics.”**

Fig. R3.2 (Extended Data Fig. 8): Spike-sorting and recording-quality metrics.

a, After automated spike sorting and manual verification, we identified 2,416 units, with an average of 2.19 ± 0.04 (mean \pm SEM, dotted vertical line) units per channel. Only channels with at least one recorded unit were included. Cumulative density functions (CDF) per brain region are shown as colored solid lines in the lower panels. **b**, Proportions of Inter-spike intervals shorter than 3 ms. Units exhibited an average proportion of (mean \pm SEM) $0.36 \pm 0.01\%$ of ISI intervals below 3 ms. More than 95% of all units showed less than 1.4% of ISIs below 3 ms (dashed vertical line). **c**, Distribution of mean firing rates (mean \pm SEM: 1.62 ± 0.05 Hz, dotted vertical line). **d**, Spike peak amplitude SNR (mean \pm SEM: 11 ± 0.1 , dotted vertical line). Peak SNR was calculated by dividing the peak amplitude by the standard deviation of the background activity, estimated based on the median absolute deviation (MAD) as $SD = MAD / 0.6745$ (Niediek et al., 2016). **e**, Mean spike peak amplitude distribution (mean \pm SEM: $44.8 \pm 0.5 \mu V$, dotted vertical line). **f**, Isolation distance (mean \pm SEM: 66 ± 12 , for the 1786 clusters for which this measure could be calculated).

Recording techniques

We thank the reviewer for raising this important point. Our manuscript now includes more details on microwire properties and implantation techniques. We made an effort to improve the presentation of several methodological aspects in the revised manuscript. Fig. R3.3a shows an image of a Behnke-Fried depth electrode with microwires protruding from the tip of the electrode. The detailed illustration in Fig. R3.3b depicts the electrode dimensions. We obtained scanning electron microscope (SEM) images before and after trimming of our electrodes and show them in Fig. R3.3c. Moreover, we now include further details on the neurosurgical procedures and electrode preparation in our manuscript. The following paragraph has been added to the Methods (p. 14; ll. 14-33): "Patients were implanted with Behnke-Fried depth electrodes (AdTech, Racine, WI) (Extended Data Fig. 1a & b). These hollow rodlike electrodes have a diameter of 1.25 mm with 8 cylindrical clinical macro electrodes (platinum-iridium). The innermost two macro contacts are spaced 3 mm apart, while the remaining contacts are equidistantly spaced. Through each electrode, a bundle of platinum-iridium microwires with a diameter of $40 \mu m$ was inserted. Each bundle contained 8 insulated high-impedance (typically 200-500 k Ω (Topalovic et al., 2023)) recording wires and one low-impedance reference wire without insulation. Electrodes were implanted using a rigid stereotactic frame (Leksell, Elekta, Sweden) with an orthogonal guide tube (Tay et al., 2023). Electrode target locations were determined by clinical

criteria and differed minimally within target regions across patients. This, along with the technical limitation of precisely localizing microwire positions, precluded us from targeting specific subregions, e.g., individual subnuclei of the amygdala or specific hippocampal subfields. Electrode placement was controlled by intraoperative CT scans co-registering the head-fixed frame to pre-operative MRI planning scans. After skin incision at the electrode entry point, a hole for an anchor bolt was drilled, and the anchor bolt was screwed into the skull using the guide tube. Microwire bundles were preloaded into the macroelectrodes and trimmed by a single cut with either a scalpel or surgical scissors on a back table in the operation room, such that they protruded from the tip of the clinical electrode by 3 to 5 mm. Extended Data Fig. 1c displays scanning electron microscope (SEM) images of uncut and cut microwires for comparison. After preparation, microwire bundles were replaced by a guiding rod for implantation. Following the insertion of the macro electrode into its target position, the guiding rod was retracted and the microwire bundle was carefully inserted avoiding kinking or bending (Tay et al., 2023).”

Fig. R3.3 (Extended Data Fig. 1): Characteristics of Behnke-Fried depth electrodes used for single-neuron recordings in the human PC and MTL.

a, Behnke-Fried depth electrode. Microwires inserted through the shaft of the hollow clinical macro electrode protrude from the tip of the electrode. The electrode features 8 cylindrical clinical platinum-iridium contacts. The two innermost contacts are 3 mm apart, while the remaining contacts are equidistantly spaced along the electrode. **b**, Illustration of the electrode geometry and dimensions. **c**, Scanning electron microscopy images of the tip of a microwire before (top) and after cutting (bottom).

To better introduce the reader to the recording method, we have now also included a SEM image of a cut microwire in the first main figure (Fig. R3.4, Fig. 1c) of our manuscript.

Recording single neurons in humans

Fig. R3.4 (Fig. 1c): Illustration of single-unit recordings in humans

Post-implantation CT scan, co-registered onto a pre-implantation MRI scan, visualizes bilaterally implanted Behnke-Fried electrodes (left). Schematic of Behnke-Fried electrode (upper right) and scanning electron microscopy image of the tip of a trimmed microwire (lower right).

We also added a detailed description of the microscopy protocol (titled “**Scanning electron microscopy of microwires**”) to our methods (p. 18; ll. 26-32): “For scanning electron microscopy, two microwires from a new bundle were utilized. One microwire was trimmed using a scalpel, while the other remained uncut. For imaging, wires were shortened to approximately 8 mm in length and mounted onto aluminum stubs using conductive carbon tape. The samples were then sputter-coated with 15 nm of gold using a Quorum 150R ES coating unit (Quorum Technologies, UK) and imaged using the Everhart-Thornley secondary electron detector in a Zeiss Sigma 300 (Zeiss, Germany) Field Emission Gun SEM operated at 2 kV.”

(2) The identification of odor responses in PCx and other areas is perfectly in line with various animal studies. While commendable to repeat those in humans they do not provide any new insight and I would recommend significantly reducing the description of those. The same holds true for the valence relationship of Amygdala SUA – which is very similar to what has been observed in animal models and in lower-resolution recordings in humans. All these aspects could and should be condensed to give more room to the SUA validation (see above) and the analysis of the highly exciting multimodal responses.

We agree that part of our work confirms that the results of decades of animal research, in fact, translate to humans. Moreover, we note that our research addresses critical aspects that are simply not accessible using animal models: reports of perceptual experiences such as valence ratings and odor identification (Fig. 4). In animal models, a “percept” can only be inferred from relatively coarse behavior (e.g., avoidance) and such inferences are limited to aspects linked to observable behavioral changes. Human subjects, on the other hand, can be questioned directly (and on a much finer scale). We have revised our manuscript to highlight the human-specific aspects of our study. The revised results now read (p. 7; ll. 2-5): “The central role of the amygdala in emotional processing is well established, and rodent studies have revealed valence coding in amygdala neurons. However, animals cannot directly report subjective preferences, and odor-valence coding remains elusive at the individual neuron level in humans.” and (p. 7; ll. 16-17): “Successful odor identification requires odor perception, recognition, and recall of the semantic odor label.”

We further note in the discussion (p. 11; ll. 35-38): "Neural representations of odors emerged first in the PC and amygdala and only approximately 500 ms later in EC. This delay in EC odor coding could support a connectivity scheme in which, unlike in rodents (Gretenkord et al., 2019; Pignatelli & Beyeler, 2019; Sosulski et al., 2011), human olfactory bulb mitral cells might not directly project to EC."

Another notable advancement of our study lies in the simultaneous recording from multiple brain areas along the central olfactory pathway. Thus, our data provide a comprehensive comparison of neuronal responses (Fig. 1), odor coding (Fig. 2), repetition suppression, and sparseness (Fig. 3) as well as valence ratings and odor identification (Fig. 4) across olfactory brain regions. These simultaneous recordings enhance our understanding of the human olfactory processing network, which to the best of our knowledge has not been achieved thus far in animal studies.

Our revised introduction now reads (p. 2; ll. 22-25): "In humans, it remains unknown if and how individual neurons respond to olfactory cues and encode odor identity. Thus, we investigated individual contributions of central olfactory areas to odor processing and their link to human behavior at the neuronal level."

We fully agree with the reviewer, however, that it is important to give more room to the SUA validation. Accordingly, we now added details on recording technique and spike sorting quality metrics (Fig. R3.2 - R.3.4) and included these aspects in the Methods and Extended Data Figures (Extended Data Figs. 1 & 8) as outlined above. We agree that these elaborations consolidate the rigor of our study. However, we believe that the findings presented as Figs. 1 - 4 are of broad interest and should thus not be presented in a more condensed form. That said, we acknowledge the reviewer's point that more room should be given to the analysis of the multimodal responses. Accordingly, we have now extended the analysis of multimodal representations and added new results to Fig. 4, ensuring they receive adequate attention in the context of our broader narrative (see our response to concern #3 below).

(3) Multimodal / mental imagery responses: This to me is the key point where human recordings truly add unique value to our understanding of the brain – but as is it is analyzed and presented only quite briefly and buried in Figure 5. The authors call the units they observe multimodal or concept units. Isn't it a possible interpretation that patients imagined smells when seeing a picture or reading a descriptor? Concept neurons have been described several times (Fried et al) and seeing them in different sensory modalities might not considered to be fundamentally novel. However, olfaction is special in that verbal descriptors, mental imagery is significantly more in doubt – I think the authors are missing an opportunity to make this point (especially discussions in philosophy and imagery e.g. RICHARD J. STEVENSON and TREVOR I. CASE Psychonomic Bulletin & Review 2005, 12 (2), 244-264 Olfactory imagery: A review or Young, B.D. (2019) Olfactory imagery: is exactly what it smells like. Philosophical Studies. <https://doi.org/10.1007/s11098-019-01371-4> - both argue for olfactory imagery and olfaction not being a special sense whereas some other studies e.g. Crowder, R. G., & Schab, F. R. (1995). Imagery for odors. In R. G. Crowder & F. R. Schab (Eds.), Memory for odors (pp. 93–107). Mahwah, NJ: Erlbaum. OR Herz, R. S. (2000). Verbal coding in olfactory versus nonolfactory cognition. Memory and Cognition, 28, 957–964. <https://doi.org/10.3758/BF03209343>. arguing against that point) My take is that the data presented here could go a long way towards answering this very fundamental question. Ideally, the authors would devise tasks where imagery is separable from direct

sensory inputs. Knowing how laborious and logistically challenging these recordings are I definitely do not want to exclude that such information can be extracted from existing data.

The reviewer asks the fascinating question of whether multimodal odor representations might reflect odor imagery. There is an ongoing debate about the existence of olfactory imagination in humans (Herz, 2000; Stevenson & Case, 2005; Young, 2020), distinguishing olfaction from other senses. Although our task did not include explicit instructions to imagine odors in absence of external stimuli, our study can still inform this debate by offering new insights into neuronal representations of semantic odor descriptors and their link to behavior. The multimodal neuronal responses of odors and related images observed in our study might represent correlates of olfactory imagination. As our study did not include a direct instruction to imagine an odor without external stimulation, we can only draw indirect conclusions. The question of whether olfactory imagery exists is inherently linked to the question of how odors are represented in the human brain and how these representations are reinstated. Specifically, our data demonstrate semantic neuronal coding principles in olfaction, which parallel findings in other modalities such as vision or auditory processing (Quiñero Quiroga et al., 2009). In this respect, olfaction does not fundamentally differ from other human senses. However, our task allowed us to go beyond pure multimodal representation of external sensory stimuli by directly questioning individuals' semantic descriptors of the odors. For each odor presentation in our identification task, we obtained two labels: (1) the chemical odor identity and (2) the chosen semantic odor descriptor (perceived odor identity). Assigning such semantic descriptors to a subjective sensory experience is a uniquely human ability, which enables us to compare neuronal representations to both the objective chemical identities and subjectively perceived and chosen semantic odor descriptor. In our revised manuscript, we ask if the neuronal activity across regions more closely reflects chemical odor identity or rather the chosen semantic odor descriptor. For this, trials in which the chosen semantic odor descriptor did not match the chemical odor identity are specifically interesting. Decoding revealed that neuronal activity in the PC more closely encodes chemical odor identities while hippocampal neurons more closely follow the subjectively perceived odor identities (selected odor descriptors). We have revised our manuscript to include these exciting findings (Fig. R3.5). We accordingly modified the abstract, introduction, results, discussion, and methods as outlined in detail below. We would like to thank the reviewer for raising this fascinating point which helped us to enhance our manuscript, and to provide additional input to the debate on olfactory imagery.

The abstract now reads (p. 1; ll. 26-27): **“While piriform neurons preferably encode chemical odor identity, hippocampal activity reflects subjective odor perception.”**

We added the following to our results (p. 7; ll. 27-31): **“We also analyzed whether neuronal odor representations reflected chemical odor identity (presented odors) rather than subjective perception (selected odor labels). Decoding revealed a dissociation between PC and hippocampus, with PC neurons coding preferably for chemical odor identity, and hippocampal neurons predicting perceived odor identity (Fig. 4h).”**

We added the following panel (Fig. R3.5) to Fig. 4 of our manuscript:

Fig. R3.5 (Fig. 4h): Piriform cortex encodes chemical odor features; hippocampus predicts subjective odor percept.

Difference in decoding accuracies based on chemical vs. perceived odor identity (selected odor labels). Firing of PC neurons decoded chemical odor identity more reliably, whereas hippocampal neurons predicted selected odor labels more accurately (PC: 75.8 % chemical vs. 22.1% perceived more accurate, $Z=19$, $P<10^{-10}$; Am: 45.6% chemical vs. 49.7% perceived, $Z=-0.95$, $P=0.34$; EC: 50% chemical vs. 45.2% perceived, $Z=1.7$, $P=0.083$; Hp: 26.7% chemical vs. 69% perceived, $Z=-16$, $P<10^{-10}$; PHC: 52.2% chemical vs. 43.5% perceived, $Z=3$, $P=0.0024$, Wilcoxon signed-rank across 1,000 subsampling runs). To improve visibility, the y-axis is truncated to display 99% of the data.

We also extended our discussion (p. 13; ll. 13-22):

“The lack of a specific odor-imagination task prevents us from delineating whether these multimodal representations are correlates of cross-modal integration or olfactory imagery (Bensafi et al., 2003). While there is an ongoing debate how olfaction differs from other human senses, particularly with regard to olfactory imagery and the role of verbal descriptors (Crowder & Schab, 1995; Herz, 2000; Stevenson & Case, 2005; Young, 2020), our findings suggest that conceptual neuronal coding schemes of olfactory information resemble those of other senses (Quiñero et al., 2009). Assigning semantic odor labels is a uniquely human ability. Here, we revealed that PC neurons preferably encode chemical odor identity, whereas hippocampal activity rather reflects subjectively perceived odors. This integrates well with our finding that hippocampal activity predicts behavioral odor identification, indicating that coherent internal and external odor representations facilitate semantic odor identification.”

Finally, we added a description of the additional decoding analysis in our methods (p. 17; ll. 24-32):

“To test whether neural activity predicted chemical odor identity better than perceived odor identity (i.e., sometimes falsely selected odor labels), we utilized a decoding analysis during the odor-identification task (4 trials per odor). An equal number of neurons was randomly subsampled from recordings in which each odor was chosen at least twice. In each anatomical target region, 100 neurons were randomly subsampled 1,000 times, and a decoder was trained using 2 cross-validation data splits and 10 resample runs. Decoders were trained based both on chemical odor identity and perceived odor identity (selected odor label) using the same neuronal populations. The differences between the two decoding accuracies were used to assess which labels were predicted more accurately by neuronal firing.”

Minor points:

(1) I am confused by the lack of correlation of Behavioral performance and Decoding accuracy anywhere but in Hippocampus – I'd expect from animal work to see significant correlations also e.g. in piriform. This could be a relevant finding – or due to e.g. different signal-to-noise from recordings in different brain areas? Generally, a discussion how to compare recording quality in different brain areas would be helpful – or tuning down the comparative analyses.

We thank the reviewer for raising this point and have added information on the recording quality across regions to our manuscript. Simultaneous recordings across multiple stages of the olfactory pathway are a unique advantage of our dataset, which have not previously been performed in animals. We are therefore convinced that it is important to include this aspect. We have added extensive information about the recording and implantation techniques to our manuscript (see major point #1 above). Importantly, we wish to emphasize that identical electrodes and implantation techniques were used across all recording sites and patients. We also analyzed data quality and spike sorting metrics for each recording site separately (Fig. R3.2). These additional analyses demonstrate a consistent recording quality across regions. Specifically, the mean signal-to-noise ratio ranged from 9.9 (PHC) to 11.8 (EC). Several influential human single unit studies have also employed simultaneous recordings across brain regions and compared results across sites, see e.g., (Aquino et al., 2023; Fu et al., 2019; Kamiński et al., 2017; Kunz et al., 2024; Minxha et al., 2020).

(2) The suppression is potentially interesting as well – however, I am not sure the conclusion that this is due to central effects is valid. Repeated exposure to odors in animal models results in suppression for a variety of reasons – notably a change in sampling behavior. Did the authors measure inhalation / sniffing in the patients? As the authors surely know, changes in sniff patterns have a profound impact on neural activity across the olfactory pathway in e.g. mice and rats and novel odors result in increased sniff rate. Thus, this could be one of several contributing factors. At the least, this needs to be discussed, better addressed by sniff recordings as done routinely in humans by Hummels, Sobel and others.

We agree with the reviewer that reduced neuronal activity to repeatedly presented odors can be caused by other factors than habituation, and changes in respiration could certainly have an impact. To minimize respiration variability within and across sessions and patients, we instructed patients (and, importantly, monitored their compliance) to take only one sniff per stimulus. In addition, we measured respiration in 13 out of 27 sessions. With this data, we can explore respiratory variability across trials.

First, we examined how well participants followed the instruction to inhale only once upon verbal cue. The results show a remarkable temporal synchrony of inhalation with odor presentation (Fig. R3.6). Respiration recordings across 13 sessions highlight that individual patients consistently followed the instruction to inhale only once upon odor presentation/command to inhale (Fig. R3.7c). To ensure that the activity of odor-modulated neurons is driven by odor identity, not respiration, we calculated a generalized linear model for each neuron taking into account respiration depth. The results show that the activity of odor-modulated neurons is explained by odor identity, but not by breathing (Fig. R3.7b). Importantly, breathing amplitudes were consistent (no significant difference) across odor repetitions as shown in Fig. R3.8, thereby corroborating that repetition suppression is not caused by reduced breathing amplitudes.

Fig. R3.6 (Fig. 1d): Averaged odor-locked respiration.

Average respiratory depth (mean \pm SEM) demonstrating how single inhalations align to odor delivery (N=13 sessions, Extended Data Fig. 5).

Fig. R3.7 (Extended Data Fig. 5): Odor identity, not respiration, drives odor-modulated neurons.

a, Respiration was measured with thoracic (upper, green) and abdominal (lower, lilac) inductive plethysmography belts. Respiration signals were amplified and recorded using the Neuralynx ATLAS system, ensuring reliable temporal synchronization with neural recordings. **b**, Performance (adjusted R^2) of linear regression models, predicting neuronal firing (z-scores) based on respiration, odor identity, or both. Respiration (inhalation depth) alone failed to predict firing of odor-modulated neurons above chance ($N=240$, $Z=-0.48$, $P=0.63$, Wilcoxon signed-rank across all odor-modulated neurons in recordings with respiratory monitoring). Odor identity predicted the firing of odor-modulated neurons significantly better than respiration (linear regression model for each odor-modulated unit to predict firing rates (z-scored): Respiration: $R^2=0.0040\pm 0.0013$; Odor identity: $R^2=0.19\pm 0.008$; Odor identity vs. respiration: $N=240$, $Z=13$, $P=1.2\cdot 10^{-40}$, Wilcoxon signed-rank). Adding respiratory information to odor identity did not significantly improved the model predictions of firing rates of odor-modulated neurons (odor identity & respiration: $R^2=0.19\pm 0.008$; odor identity & respiration vs. odor identity alone $N=240$, $Z=0.85$, $P=0.39$; odor identity & respiration vs. respiration: $N=240$, $Z=13$, $P=4.0\cdot 10^{-41}$, Wilcoxon signed-rank). Thus, odor-modulated neurons are primarily driven by odor-specific differences and not variations in respiration. **c**, Averaged odor-

locked respiration across recording sessions (mean \pm SEM, 13 sessions). The initiation of inhalation was closely synchronized with odor presentation (dashed line). **c**, Odor-locked respiratory signals for each individual recording session. Participants consistently inhaled once (single peak) during the first 2 seconds after odor onset (gray shaded area), the analysis time window used for identification of odor-modulated neurons.

Fig. R3.8 (Extended Data Fig. 7b-c): Repetition suppression is not caused by changes in odor sampling.

a, Average respiratory traces (mean \pm SEM) for each odor presentation (trials 1 to 8) across 13 recording sessions. **b**, Averaged inhalation depth for each odor presentation (1 to 8) and recording session (colored dots). Inhalation depth was consistent across odor repetitions (one-way ANOVA, $F(7,96)=0.3$, $P=0.95$).

We have modified the manuscript to include the additional respiratory data and conclusions. Experimental procedures (p. 15; ll. 13-19): “Odor pens were presented approximately 2 cm below the nose, centered between both nostrils. Patients were verbally instructed on each trial to inhale on command ('Please inhale NOW!'). To ensure consistent odor sampling across trials, participants were asked to inhale only once for each odor presentation and not sniff at their convenience. Odor pens were immediately removed after the first inhalation. This experimental protocol was devised to minimize odor-specific respiratory variability. The experimenter's (MSK) direct supervision ensured adherence to the instructions throughout the experiment.”

We added the following to the methods (p. 15; ll. 21-27): “In 13 of 27 recording sessions, respiration was measured using thoracic and abdominal plethysmography belts (Extended Data Fig. 5a; SleepSense, S.L.P. Scientific Laboratory Products, Elgin, IL). Data from both belts were averaged and analyzed using the *Breathmetrics* (Noto et al., 2018) (human respiratory belt default settings with sliding baseline correction. In the remaining 14 sessions, respiration belts could not be applied due to patient discomfort or noisy interference with the microwire recordings. Overall, participants complied accurately with the experimental protocol, inhaling once during odor exposure and well timed to odor delivery (Fig. 1d, Extended Data Fig. 5c).”

We further added the following to our results on repetition suppression (p. 6; ll. 16-17): “This effect was not caused by decreased inhalation (Extended Data Fig. 7b & c).”

(3) Several statements in the text tend to ignore animal literature – for example “the single neuron basis of olfactory valence coding is unexplored” might be true for human work but ample animal work has addressed exactly this (reviewed e.g. in Pignatelli Beyeler *Curr Opin Behav Sci.* 2019 Apr; 26: 97–106.). Moreover, Winston Gottfried, *Nolan J Neurosci* 2005 should be cited in the context of valence coding in general.

The authors should more carefully consider the animal literature and assess more directly where their findings with putative SU in humans adds, contradicts, or complements the literature.

This includes a more balanced perspective on the unimodal interpretation of piriform – my understanding of the rodent literature is that this is not the commonly held perspective anymore. To be fair, the authors do mention e.g. Poo et al and Schoorover et al – but the statement in the abstract that their work challenges “traditional views of a strictly unimodal (‘olfaction-only’) role” is overstated (the authors mention “human piriform cortex” but this is one example where after recent rodent work this perspective has also changed in the human olfactory neuroscience community). Again, this should not take away anything from the highly exciting findings of multimodal responses in piriform that might indicate cellular correlates of olfactory imagery but need to be phrased in a more balanced way.

We thank the reviewer for raising this point. We agree that recent rodent studies point towards a more multimodal role of the PC. We thus adjusted the wording in a more balanced way throughout the manuscript. We apologize that results from animal studies were not sufficiently included in the original manuscript. We have carefully revised our manuscript to better incorporate the relevant animal literature.

In the abstract, we now write (p. 1; ll. 27-29): “Critically, we uncover that piriform cortex neurons reliably encode odor-related images, supporting a multimodal role of the human piriform cortex.”

The section on valence coding now reads (p. 7; ll. 2-6): “The central role of the amygdala in emotional processing is well established (Winston et al., 2005; Lindquist et al., 2016; Pessoa & Adolphs, 2010), and rodent studies have revealed valence coding in amygdala neurons (Pignatelli & Beyeler, 2019). However, animals cannot directly report subjective preferences, and odor-valence coding remains elusive at the individual neuron level in humans. Consequently, we investigated whether amygdala neurons encode subjective odor preferences and valence.”

We also added a section on the multimodal role of PC to the discussion (p. 13; ll. 4-12): “Recent rodent studies have shown that neurons in the posterior PC precisely encode spatial information, suggesting a role in odor-place association (Poo et al., 2022). Further evidence for multimodal processing of odor-related information in the PC stems from rodents (Mandairon et al., 2014) and human Imaging studies (Djordjevic et al., 2008; González et al., 2006; Schulze et al., 2017). Here, we tested semantically coherent olfactory and visual stimuli to explore coding of PC neurons beyond olfactory perception. We discovered that PC neurons not only decode odors, but also odor-related image identities. The PC therefore not only processes olfactory stimuli, but also integrates top-down semantic information from higher cognitive areas. Most strikingly, odor-related images were decoded more accurately in PC than in MTL. Future research will need to explore whether PC neurons specifically encode odor-related images, or whether they also process images of odorless objects.”

(4) In my opinion, the authors overemphasize the importance of olfaction for human health and disease. The study as discussed above does have the potential to address fundamental questions in human neurophilosophy – so no need to overemphasize the health consequences of hyposmia.

We followed the reviewer’s suggestion and condensed our discussion on neurodegenerative diseases and olfaction. We would, however, prefer to not completely remove this point from the discussion since olfactory deficits play a role in neurodegenerative diseases and since the underlying mechanisms are still widely unexplored. The condensed paragraph in the discussion now reads (p. 12; ll. 33-39):

”Neurodegenerative diseases such as Parkinson's and Alzheimer's often first manifest with olfactory deficits, particularly concerning odor identification (Devanand et al., 2000; Doty, 2012; Ubeda-Bañon et al., 2020). Our results link odor representations of hippocampal neurons directly with behavioral odor-identification performance, indicating that hippocampal degeneration may contribute to odor-identification deficits. Impaired behavioral odor identification performance could be a direct result of local neurodegeneration or could instead result indirectly from degeneration of upstream circuits (e.g., olfactory bulb). Future research will have to explore causal contributions of odor-modulated neurons in odor identification.”

We are very grateful for the many important points raised by the reviewer. Although our study was not designed to settle the ongoing discussion on human olfactory imagination, it informs the debate by revealing multimodal neuronal coding schemes and uncovering a dissociation of odor representations between chemical and subjectively perceived odor identity in the PC and hippocampus, respectively.

References

- Aarts, E., Verhage, M., Veenvliet, J. V., Dolan, C. V., & van der Sluis, S. (2014). A solution to dependency: Using multilevel analysis to accommodate nested data. *Nature Neuroscience*, *17*(4), 491–496. <https://doi.org/10.1038/nn.3648>
- Allison, A. C. (1954). The secondary olfactory areas in the human brain. *Journal of Anatomy*, *88*(Pt 4), 481–488.2.
- Anderson, A. K., Christoff, K., Stappen, I., Panitz, D., Ghahremani, D. G., Glover, G., Gabrieli, J. D. E., & Sobel, N. (2003). Dissociated neural representations of intensity and valence in human olfaction. *Nature Neuroscience*, *6*(2), 196–202. <https://doi.org/10.1038/nn1001>
- Aquino, T. G., Cockburn, J., Mamelak, A. N., Rutishauser, U., & O’Doherty, J. P. (2023). Neurons in human pre-supplementary motor area encode key computations for value-based choice. *Nature Human Behaviour*, 1–16. <https://doi.org/10.1038/s41562-023-01548-2>
- Bao, X., Gjorgieva, E., Shanahan, L. K., Howard, J. D., Kahnt, T., & Gottfried, J. A. (2019). Grid-like Neural Representations Support Olfactory Navigation of a Two-Dimensional Odor Space. *Neuron*, *102*(5), 1066–1075.e5. <https://doi.org/10.1016/j.neuron.2019.03.034>
- Barr, D. J., Levy, R., Scheepers, C., & Tily, H. J. (2013). Random effects structure for confirmatory hypothesis testing: Keep it maximal. *Journal of Memory and Language*, *68*(3), 255–278. <https://doi.org/10.1016/j.jml.2012.11.001>
- Bensafi, M., Porter, J., Pouliot, S., Mainland, J., Johnson, B., Zelano, C., Young, N., Bremner, E., Aframian, D., Khan, R., & Sobel, N. (2003). Olfactomotor activity during imagery mimics that during perception. *Nature Neuroscience*, *6*(11), 1142–1144. <https://doi.org/10.1038/nn1145>
- Bhutani, S., Howard, J. D., Reynolds, R., Zee, P. C., Gottfried, J., & Kahnt, T. (2019). Olfactory connectivity mediates sleep-dependent food choices in humans. *eLife*, *8*, e49053. <https://doi.org/10.7554/eLife.49053>
- Bolding, K. A., & Franks, K. M. (2017). Complementary codes for odor identity and intensity in olfactory cortex. *eLife*, *6*, e22630. <https://doi.org/10.7554/eLife.22630>
- Bolding, K. A., Nagappan, S., Han, B.-X., Wang, F., & Franks, K. M. (2020). Recurrent circuitry is required to stabilize piriform cortex odor representations across brain states. *eLife*, *9*, e53125. <https://doi.org/10.7554/eLife.53125>
- Cao, R., Lin, C., Brandmeir, N. J., & Wang, S. (2022). A human single-neuron dataset for face perception. *Scientific Data*, *9*(1), Article 1. <https://doi.org/10.1038/s41597-022-01482-4>
- Chahre, F. J., & Rey, H. G. (2020). General Framework for Tracking Neural Activity Over Long-Term Extracellular Recordings. *bioRxiv*, 2020.02.11.944686. <https://doi.org/10.1101/2020.02.11.944686>
- Crowder, R. G., & Schab, F. R. (1995). Imagery for Odors. In *Memory for Odors*. Psychology Press.
- Devanand, D. p., Michaels-Marston, K. S., Liu, X., Pelton, G. H., Padilla, M., Marder, K., Bell, K., Stern, Y., & Mayeux, R. (2000). Olfactory Deficits in Patients With Mild Cognitive Impairment Predict Alzheimer’s Disease at Follow-Up. *American Journal of Psychiatry*, *157*(9), 1399–1405. <https://doi.org/10.1176/appi.ajp.157.9.1399>
- Djordjevic, J., Lundstrom, J. N., Clément, F., Boyle, J. A., Pouliot, S., & Jones-Gotman, M. (2008). A Rose by Any Other Name: Would it Smell as Sweet? *Journal of Neurophysiology*, *99*(1), 386–393. <https://doi.org/10.1152/jn.00896.2007>
- Doty, R. L. (2012). Olfactory dysfunction in Parkinson disease. *Nature Reviews Neurology*, *8*(6), 329–339. <https://doi.org/10.1038/nrneurol.2012.80>
- Echevarria-Cooper, S. L., Zhou, G., Zelano, C., Pestilli, F., Parrish, T. B., & Kahnt, T. (2022). Mapping the Microstructure and Striae of the Human Olfactory Tract with Diffusion MRI. *Journal of Neuroscience*, *42*(1), 58–68. <https://doi.org/10.1523/JNEUROSCI.1552-21.2021>
- Faraut, M. C. M., Carlson, A. A., Sullivan, S., Tudusciuc, O., Ross, I., Reed, C. M., Chung, J. M., Mamelak, A. N., & Rutishauser, U. (2018). Dataset of human medial temporal lobe single neuron activity during declarative memory encoding and recognition. *Scientific Data*, *5*, 180010. <https://doi.org/10.1038/sdata.2018.10>
- Fu, Z., Wu, D.-A. J., Ross, I., Chung, J. M., Mamelak, A. N., Adolphs, R., & Rutishauser, U. (2019). Single-Neuron Correlates of Error Monitoring and Post-Error Adjustments in Human Medial Frontal Cortex. *Neuron*, *101*(1), 165–177.e5. <https://doi.org/10.1016/j.neuron.2018.11.016>
- Gold, C., Henze, D. A., Koch, C., & Buzsáki, G. (2006). On the Origin of the Extracellular Action Potential Waveform: A Modeling Study. *Journal of Neurophysiology*, *95*(5), 3113–3128. <https://doi.org/10.1152/jn.00979.2005>
- González, J., Barros-Loscertales, A., Pulvermüller, F., Meseguer, V., Sanjuán, A., Belloch, V., & Ávila, C. (2006). Reading *cinnamon* activates olfactory brain regions. *NeuroImage*, *32*(2), 906–912. <https://doi.org/10.1016/j.neuroimage.2006.03.037>
- Gretenkord, S., Kostka, J. K., Hartung, H., Watznauer, K., Fleck, D., Minier-Toribio, A., Spehr, M., & Hanganu-Opatz, I. L. (2019). Coordinated electrical activity in the olfactory bulb gates the oscillatory entrainment of entorhinal networks in neonatal mice. *PLOS Biology*, *17*(1), e2006994. <https://doi.org/10.1371/journal.pbio.2006994>

- Harris, K. D., Quiroga, R. Q., Freeman, J., & Smith, S. L. (2016). Improving data quality in neuronal population recordings. *Nature Neuroscience*, *19*(9), Article 9. <https://doi.org/10.1038/nn.4365>
- Hayat, H., Marmelshtein, A., Krom, A. J., Sela, Y., Tankus, A., Strauss, I., Fahoum, F., Fried, I., & Nir, Y. (2022). Reduced neural feedback signaling despite robust neuron and gamma auditory responses during human sleep. *Nature Neuroscience*, *25*(7), Article 7. <https://doi.org/10.1038/s41593-022-01107-4>
- Herz, R. S. (2000). Verbal coding in olfactory versus nonolfactory cognition. *Memory & Cognition*, *28*(6), 957–964. <https://doi.org/10.3758/BF03209343>
- Iravani, B., Arshamian, A., Ohla, K., Wilson, D. A., & Lundström, J. N. (2020). Non-invasive recording from the human olfactory bulb. *Nature Communications*, *11*(1), Article 1. <https://doi.org/10.1038/s41467-020-14520-9>
- Iurilli, G., & Datta, S. R. (2017). Population Coding in an Innately Relevant Olfactory Area. *Neuron*, *93*(5), 1180–1197.e7. <https://doi.org/10.1016/j.neuron.2017.02.010>
- Jamali, M., Grannan, B. L., Fedorenko, E., Saxe, R., Báez-Mendoza, R., & Williams, Z. M. (2021). Single-neuronal predictions of others' beliefs in humans. *Nature*, *591*(7851), Article 7851. <https://doi.org/10.1038/s41586-021-03184-0>
- Jin, J., Zelano, C., Gottfried, J. A., & Mohanty, A. (2015). Human Amygdala Represents the Complete Spectrum of Subjective Valence. *Journal of Neuroscience*, *35*(45), 15145–15156. <https://doi.org/10.1523/JNEUROSCI.2450-15.2015>
- Kamiński, J., Brzezicka, A., Mamelak, A. N., & Rutishauser, U. (2020). Combined Phase-Rate Coding by Persistently Active Neurons as a Mechanism for Maintaining Multiple Items in Working Memory in Humans. *Neuron*, *106*(2), 256–264.e3. <https://doi.org/10.1016/j.neuron.2020.01.032>
- Kamiński, J., Sullivan, S., Chung, J. M., Ross, I. B., Mamelak, A. N., & Rutishauser, U. (2017). Persistently active neurons in human medial frontal and medial temporal lobe support working memory. *Nature Neuroscience*, *20*(4), 590–601. <https://doi.org/10.1038/nn.4509>
- Keller, A., Gerkin, R. C., Guan, Y., Dhurandhar, A., Turu, G., Szalai, B., Mainland, J. D., Ihara, Y., Yu, C. W., Wolfinger, R., Vens, C., schietgat, leander, De Grave, K., Norel, R., DREAM Olfaction Prediction Consortium, Stolovitzky, G., Cecchi, G. A., Vosshall, L. B., & meyer, pablo. (2017). Predicting human olfactory perception from chemical features of odor molecules. *Science*, *355*(6327), 820–826. <https://doi.org/10.1126/science.aal2014>
- Kolibius, L. D., Roux, F., Parish, G., Ter Wal, M., Van Der Plas, M., Chelvarajah, R., Sawlani, V., Rollings, D. T., Lang, J. D., Gollwitzer, S., Walther, K., Hopfengärtner, R., Kreiselmeier, G., Hamer, H., Staresina, B. P., Wimber, M., Bowman, H., & Hanslmayr, S. (2023). Hippocampal neurons code individual episodic memories in humans. *Nature Human Behaviour*, 1–12. <https://doi.org/10.1038/s41562-023-01706-6>
- Kunz, L., Brandt, A., Reinacher, P. C., Staresina, B. P., Reifensstein, E. T., Weidemann, C. T., Herweg, N. A., Patel, A., Tsitsiklis, M., Kempter, R., Kahana, M. J., Schulze-Bonhage, A., & Jacobs, J. (2021). A neural code for egocentric spatial maps in the human medial temporal lobe. *Neuron*, *109*(17), 2781–2796.e10. <https://doi.org/10.1016/j.neuron.2021.06.019>
- Kunz, L., Staresina, B. P., Reinacher, P. C., Brandt, A., Guth, T. A., Schulze-Bonhage, A., & Jacobs, J. (2024). Ripple-locked coactivity of stimulus-specific neurons and human associative memory. *Nature Neuroscience*, *27*(3), 587–599. <https://doi.org/10.1038/s41593-023-01550-x>
- Kutter, E. F., Bostroem, J., Elger, C. E., Mormann, F., & Nieder, A. (2018). Single Neurons in the Human Brain Encode Numbers. *Neuron*, *100*(3), 753–761.e4. <https://doi.org/10.1016/j.neuron.2018.08.036>
- Lindquist, K. A., Satpute, A. B., Wager, T. D., Weber, J., & Barrett, L. F. (2016). The Brain Basis of Positive and Negative Affect: Evidence from a Meta-Analysis of the Human Neuroimaging Literature. *Cerebral Cortex*, *26*(5), 1910–1922. <https://doi.org/10.1093/cercor/bhv001>
- Mandairon, N., Kermen, F., Charpentier, C., Sacquet, J., Linster, C., & Didier, A. (2014). Context-driven activation of odor representations in the absence of olfactory stimuli in the olfactory bulb and piriform cortex. *Frontiers in Behavioral Neuroscience*, *8*. <https://www.frontiersin.org/articles/10.3389/fnbeh.2014.00138>
- Mignot, C., Schunke, A., Sinding, C., & Hummel, T. (2022). Olfactory adaptation: Recordings from the human olfactory epithelium. *European Archives of Oto-Rhino-Laryngology*, *279*(7), 3503–3510. <https://doi.org/10.1007/s00405-021-07170-0>
- Minxha, J., Adolphs, R., Fusi, S., Mamelak, A. N., & Rutishauser, U. (2020). Flexible recruitment of memory-based choice representations by the human medial frontal cortex. *Science*, *368*(6498). <https://doi.org/10.1126/science.aba3313>
- Miura, K., Mainen, Z. F., & Uchida, N. (2012). Odor Representations in Olfactory Cortex: Distributed Rate Coding and Decorrelated Population Activity. *Neuron*, *74*(6), 1087–1098. <https://doi.org/10.1016/j.neuron.2012.04.021>
- Niediek, J., Boström, J., Elger, C. E., & Mormann, F. (2016). Reliable Analysis of Single-Unit Recordings from the Human Brain under Noisy Conditions: Tracking Neurons over Hours. *PLoS ONE*, *11*(12). <https://doi.org/10.1371/journal.pone.0166598>

- Noto, T., Zhou, G., Schuele, S., Templer, J., & Zelano, C. (2018). Automated analysis of breathing waveforms using BreathMetrics: A respiratory signal processing toolbox. *Chemical Senses*, *43*(8), 583–597. <https://doi.org/10.1093/chemse/bjy045>
- Oleszkiewicz, A., Schriever, V. A., Croy, I., Hähner, A., & Hummel, T. (2019). Updated Sniffin' Sticks normative data based on an extended sample of 9139 subjects. *European Archives of Oto-Rhino-Laryngology*, *276*(3), 719–728. <https://doi.org/10.1007/s00405-018-5248-1>
- Pessoa, L., & Adolphs, R. (2010). Emotion processing and the amygdala: From a “low road” to “many roads” of evaluating biological significance. *Nature Reviews Neuroscience*, *11*(11), Article 11. <https://doi.org/10.1038/nrn2920>
- Pignatelli, M., & Beyeler, A. (2019). Valence coding in amygdala circuits. *Current Opinion in Behavioral Sciences*, *26*, 97–106. <https://doi.org/10.1016/j.cobeha.2018.10.010>
- Poo, C., Agarwal, G., Bonacchi, N., & Mainen, Z. F. (2022). Spatial maps in piriform cortex during olfactory navigation. *Nature*, *601*(7894), Article 7894. <https://doi.org/10.1038/s41586-021-04242-3>
- Quian Quiroga, R., Kraskov, A., Koch, C., & Fried, I. (2009). Explicit encoding of multimodal percepts by single neurons in the human brain. *Curr. Biol.*, *19*, 1308–1313.
- Raithel, C. U., Miller, A. J., Epstein, R. A., Kahnt, T., & Gottfried, J. A. (2023). Recruitment of grid-like responses in human entorhinal and piriform cortices by odor landmark-based navigation. *Current Biology*, *33*(17), 3561-3570.e4. <https://doi.org/10.1016/j.cub.2023.06.087>
- Rey, H. G., Gori, B., Chaure, F. J., Collavini, S., Blenkman, A. O., Seoane, P., Seoane, E., Kochen, S., & Quian Quiroga, R. (2020). Single Neuron Coding of Identity in the Human Hippocampal Formation. *Current Biology*. <https://doi.org/10.1016/j.cub.2020.01.035>
- Roland, B., Deneux, T., Franks, K. M., Bathellier, B., & Fleischmann, A. (2017). Odor identity coding by distributed ensembles of neurons in the mouse olfactory cortex. *eLife*, *6*, e26337. <https://doi.org/10.7554/eLife.26337>
- Rutishauser, U., Ross, I. B., Mamelak, A. N., & Schuman, E. M. (2010). Human memory strength is predicted by theta-frequency phase-locking of single neurons. *Nature*, *464*(7290), 903–907. <https://doi.org/10.1038/nature08860>
- Rutishauser, U., Ye, S., Koroma, M., Tudusciuc, O., Ross, I. B., Chung, J. M., & Mamelak, A. N. (2015). Representation of retrieval confidence by single neurons in the human medial temporal lobe. *Nature Neuroscience*, *18*(7), Article 7. <https://doi.org/10.1038/nn.4041>
- Sagar, V., Shanahan, L. K., Zelano, C. M., Gottfried, J. A., & Kahnt, T. (2023). High-precision mapping reveals the structure of odor coding in the human brain. *Nature Neuroscience*, *26*(9), Article 9. <https://doi.org/10.1038/s41593-023-01414-4>
- Schonhaut, D. R., Ramayya, A. G., Solomon, E. A., Herweg, N. A., Fried, I., & Kahana, M. J. (2020). Single neurons throughout human memory regions phase-lock to hippocampal theta. *bioRxiv*, 2020.06.30.180174. <https://doi.org/10.1101/2020.06.30.180174>
- Schulze, P., Bestgen, A.-K., Lech, R. K., Kuchinke, L., & Suchan, B. (2017). Preprocessing of emotional visual information in the human piriform cortex. *Scientific Reports*, *7*(1), 9191. <https://doi.org/10.1038/s41598-017-09295-x>
- Sosulski, D. L., Bloom, M. L., Cutforth, T., Axel, R., & Datta, S. R. (2011). Distinct representations of olfactory information in different cortical centres. *Nature*, *472*(7342), 213–216. <https://doi.org/10.1038/nature09868>
- Stevenson, R. J., & Case, T. I. (2005). Olfactory imagery: A review. *Psychonomic Bulletin & Review*, *12*(2), 244–264. <https://doi.org/10.3758/BF03196369>
- Tay, A. S.-M. S., Caravan, B., & Mamelak, A. N. (2023). Which Are the Most Important Aspects of Microelectrode Implantation? In N. Axmacher (Ed.), *Intracranial EEG: A Guide for Cognitive Neuroscientists* (pp. 671–682). Springer International Publishing. https://doi.org/10.1007/978-3-031-20910-9_42
- Toet, A., Eijnsman, S., Liu, Y., Donker, S., Kaneko, D., Brouwer, A.-M., & van Erp, J. B. F. (2020). The Relation Between Valence and Arousal in Subjective Odor Experience. *Chemosensory Perception*, *13*(2), 141–151. <https://doi.org/10.1007/s12078-019-09275-7>
- Tong, A. P. S., Vaz, A. P., Wittig, J. H., Inati, S. K., & Zaghoul, K. A. (2021). Ripples reflect a spectrum of synchronous spiking activity in human anterior temporal lobe. *eLife*, *10*, e68401. <https://doi.org/10.7554/eLife.68401>
- Topalovic, U., Barclay, S., Ling, C., Alzuhair, A., Yu, W., Hokhikyan, V., Chandrakumar, H., Rozgic, D., Jiang, W., Basir-Kazeruni, S., Maoz, S. L., Inman, C. S., Stangl, M., Gill, J., Bari, A., Fallah, A., Eliashiv, D., Pouratian, N., Fried, I., ... Markovic, D. (2023). A wearable platform for closed-loop stimulation and recording of single-neuron and local field potential activity in freely moving humans. *Nature Neuroscience*, *26*(3), Article 3. <https://doi.org/10.1038/s41593-023-01260-4>
- Tukey, J. W. (1949). Comparing individual means in the analysis of variance. *Biometrics*, *5*(2), 99–114.
- Ubeda-Bañon, I., Saiz-Sanchez, D., Flores-Cuadrado, A., Rioja-Corroto, E., Gonzalez-Rodriguez, M., Villar-Conde, S., Astillero-Lopez, V., Cabello-de la Rosa, J. P., Gallardo-Alcañiz, M. J., Vaamonde-Gamo, J., Relea-Calatayud, F., Gonzalez-

- Lopez, L., Mohedano-Moriano, A., Rabano, A., & Martinez-Marcos, A. (2020). The human olfactory system in two proteinopathies: Alzheimer's and Parkinson's diseases. *Translational Neurodegeneration*, 9(1), 22. <https://doi.org/10.1186/s40035-020-00200-7>
- Umbach, G., Kantak, P., Jacobs, J., Kahana, M., Pfeiffer, B. E., Sperling, M., & Lega, B. (2020). Time cells in the human hippocampus and entorhinal cortex support episodic memory. *Proceedings of the National Academy of Sciences*, 117(45), 28463–28474. <https://doi.org/10.1073/pnas.2013250117>
- Viswam, V., Obien, M. E. J., Franke, F., Frey, U., & Hierlemann, A. (2019). Optimal Electrode Size for Multi-Scale Extracellular-Potential Recording From Neuronal Assemblies. *Frontiers in Neuroscience*, 13. <https://doi.org/10.3389/fnins.2019.00385>
- Winston, J. S., Gottfried, J. A., Kilner, J. M., & Dolan, R. J. (2005). Integrated Neural Representations of Odor Intensity and Affective Valence in Human Amygdala. *Journal of Neuroscience*, 25(39), 8903–8907. <https://doi.org/10.1523/JNEUROSCI.1569-05.2005>
- Yang, Q., Zhou, G., Noto, T., Templer, J. W., Schuele, S. U., Rosenow, J. M., Lane, G., & Zelano, C. (2022). Smell-induced gamma oscillations in human olfactory cortex are required for accurate perception of odor identity. *PLOS Biology*, 20(1), e3001509. <https://doi.org/10.1371/journal.pbio.3001509>
- Young, B. D. (2020). Olfactory imagery: Is exactly what it smells like. *Philosophical Studies*, 177(11), 3303–3327. <https://doi.org/10.1007/s11098-019-01371-4>
- Zheng, J., Schjetnan, A. G. P., Yebra, M., Gomes, B. A., Mosher, C. P., Kalia, S. K., Valiante, T. A., Mamelak, A. N., Kreiman, G., & Rutishauser, U. (2022). Neurons detect cognitive boundaries to structure episodic memories in humans. *Nature Neuroscience*, 25(3), Article 3. <https://doi.org/10.1038/s41593-022-01020-w>
- Zhou, G., Lane, G., Noto, T., Arabkheradmand, G., Gottfried, J. A., Schuele, S. U., Rosenow, J. M., Olofsson, J. K., Wilson, D. A., & Zelano, C. (2019). Human olfactory-auditory integration requires phase synchrony between sensory cortices. *Nature Communications*, 10(1), Article 1. <https://doi.org/10.1038/s41467-019-09091-3>

Reviewer Reports on the First Revision:

Referees' comments:

Referee #1 (Remarks to the Author):

The authors have done a terrific job carefully and satisfactorily addressing my concerns and, in my opinion, those of the other reviewers. This work is of remarkable significance and needs to be seen by a broad audience.

I want to comment on Reviewer 2's comments and "concerns" about odor delivery and respiration monitoring not being "state-of-the-art". As the authors make clear, these experiments were done under very challenging conditions in a dynamic and fast-changing environment in which the patient's health and compliance were more important than controlling the environment for scientific rigor. That they were able to get 17 people to do this task while electrodes were being stuck in their brains is fantastic. However, the authors may also be too polite to say that this is not a valid scientific concern and should not impact the decision to publish this manuscript. If the authors had not found anything and were reporting a negative finding (e.g., "Odor identity is not encoded in the piriform cortex") then it is valid to question whether the experimental methods were sufficiently robust to support their conclusion. However, in this case, the authors report many interesting and important findings, despite the technical obstacles and limitations. And so, it is reasonable to assume that more rigorous conditions would increase the robustness of their findings.

I have one small question: On p. 6, l. 18, they say, "For PHC odor-modulated neurons, no repetition suppression was observed." I thought there were no statistically significantly odor-modulated neurons in PHC (p. 4, l. 25). Please clarify.

Referee #2 (Remarks to the Author):

Overall, the authors have done an outstanding job addressing my initial comments and concerns. I have two lingering points.

1. I appreciate the description and analysis of respiratory measures. However, Extended Data Figure 5b now includes a circular analysis. Odor-modulated neurons were defined based on their activity being significantly modulated by odor identity. Comparing the results of regression models using odor identity and respiration to predict activity from odor-modulated neurons is highly biased (as can be seen by the fact that no data point in the middle and right bar in Extended Data Figure 5b is below $R^2 \sim 0.1$). NB, comparing odor identity models with and without respiration does not have the same flaw. I would suggest removing the left-most bar from Extended Data Figure 5b as well as the corresponding statistical analysis.

2. I further appreciate the new decoding and statistical analyses using data separated by session and subject. It is reassuring that the results hold statistically. However, there are staggering quantitative differences between decoding based data on pooled or separated across subjects and sessions (up to 4-fold increase in accuracy). This illustrates my initial concern that decoding on pooled data paints a biased picture. The fact that previous work has pooled across sessions and participants/animals does not make this practice more appropriate. As the field matures, appropriate methods should be used whenever possible. I therefore urge the authors to replace the results and figures in the main text with decoding analyses and statistical tests that are based on data segregated by session/subjects, whenever feasible. I believe this would increase the impact of this work beyond the scientific findings by standing out in the field of human neural recording studies as an example that uses proper statistical approaches.

Referee #3 (Remarks to the Author):

I want to congratulate the authors on an excellent manuscript. They comprehensively addressed all my points. I am particularly impressed by how thoroughly they are now describing their recording methodology, spike sorting, unit quality metrics. This alone would make this a landmark study for the emerging field of human single-unit recordings.

Moreover, as far as I can see they did address the other reviewer's points exhaustively as well. I agree with Reviewer 2 that ideally high-temporal control odor delivery devices and detailed nasal airflow measurements benefit the interpretation of neuronal recordings from olfactory areas. However, I do appreciate the author's discussion of the clinical constraints and in my opinion the unique opportunity to perform these well-designed experiments in patients and the high bar for reproducibility as outlined in the rebuttal response far outweigh any concerns with respect to further optimized odor delivery. In fact, with respect to the specific questions addressed by the authors I would struggle seeing a significant impact of using sniff sticks or the limited accuracy of using plethysmograph measurements as a proxy for airflow. Possibly result variability would go down further with improved sniff- or odor-onset-aligning but I struggle to see any results suffering from their odor delivery limitations. Fig R2.1/ Extended data Fig 5 is quite compelling in this respect.

Congratulations on a very important piece of work.

Author Rebuttals to First Revision:

Reply to the Referees' Comments

We thank the three reviewers for their positive and constructive evaluation of our manuscript.

To address the remaining Referees' comments, we submit a revised manuscript, as described below in a point-by-point reply (marked in **dark blue font**) to the reviews (in *italics*). Changes to the manuscript text are denoted in **red font**.

Comments to Referee 1:

The authors have done a terrific job carefully and satisfactorily addressing my concerns and, in my opinion, those of the other reviewers. This work is of remarkable significance and needs to be seen by a broad audience.

I want to comment on Reviewer 2's comments and "concerns" about odor delivery and respiration monitoring not being "state-of-the-art". As the authors make clear, these experiments were done under very challenging conditions in a dynamic and fast-changing environment in which the patient's health and compliance were more important than controlling the environment for scientific rigor. That they were able to get 17 people to do this task while electrodes were being stuck in their brains is fantastic. However, the authors may also be too polite to say that this is not a valid scientific concern and should not impact the decision to publish this manuscript. If the authors had not found anything and were reporting a negative finding (e.g., "Odor identity is not encoded in the piriform cortex") then it is valid to question whether the experimental methods were sufficiently robust to support their conclusion. However, in this case, the authors report many interesting and important findings, despite the technical obstacles and limitations. And so, it is reasonable to assume that more rigorous conditions would increase the robustness of their findings.

I have one small question: On p. 6, l. 18, they say, "For PHC odor-modulated neurons, no repetition suppression was observed." I thought there were no statistically significantly odor-modulated neurons in PHC (p. 4, l. 25). Please clarify.

We thank the reviewer for their positive assessment of our revised manuscript and for raising a valid point. Although some PHC neurons are identified as odor-modulated according to our criterion ($P_{ANOVA} < 0.05$, Fig. 1h), their number does not exceed chance level. We therefore agree that analyzing repetition suppression for this non-significant fraction of neurons does not lead to easily interpretable results. Accordingly, we have removed the respective sentence and the corresponding panels from Fig. 3b & Extended Data Fig. 7e. We retain the PHC in Extended Data Fig. 7d as this figure shows repetition suppression across all recorded neurons without prior selection of odor-modulated neurons.

Comments to Referee 2:

Overall, the authors have done an outstanding job addressing my initial comments and concerns. I have two lingering points.

1. I appreciate the description and analysis of respiratory measures. However, Extended Data Figure 5b now includes a circular analysis. Odor-modulated neurons were defined based on their activity being significantly modulated by odor identity. Comparing the results of regression models using odor identity and respiration to predict activity from odor-modulated neurons is highly biased (as can be seen by the fact that no data point in the middle and right bar in Extended Data Figure 5b is below $R^2 \sim 0.1$). NB, comparing odor identity models with and without respiration does not have the same flaw. I would suggest removing the left-most bar from Extended Data Figure 5b as well as the corresponding statistical analysis.

We thank the reviewer for raising this point. While the finding that variations in respiration alone do not explain the firing of odor-modulated neurons above chance is an interesting piece of information, we agree that the comparison between respiration alone and odor-identity is biased by the selection of odor-modulated neurons and might be misleading for the reader. Following the reviewer's suggestion, we have therefore removed the left-most bar in Extended Data Fig. 5b as well as the corresponding statistical comparisons.

2. I further appreciate the new decoding and statistical analyses using data separated by session and subject. It is reassuring that the results hold statistically. However, there are staggering quantitative differences between decoding based data on pooled or separated across subjects and sessions (up to 4-fold increase in accuracy). This illustrates my initial concern that decoding on pooled data paints a biased picture. The fact that previous work has pooled across sessions and participants/animals does not make this practice more appropriate. As the field matures, appropriate methods should be used whenever possible. I therefore urge the authors to replace the results and figures in the main text with decoding analyses and statistical tests that are based on data segregated by session/subjects, whenever feasible. I believe this would increase the impact of this work beyond the scientific findings by standing out in the field of human neural recording studies as an example that uses proper statistical approaches.

We thank the reviewer for their comment and agree that session and subject-wise decoding contributes to our results. However, the increased decoding accuracy in the population decoding is not the results of a biased analysis but a direct consequence of the increased number of neurons included ($n=200$ neurons). This is precisely what should be expected based on Fig. 2c, which shows how decoding accuracy increases with the number of randomly drawn neurons. While it is currently technically not feasible to simultaneously record this many neurons in a single human subject, we anticipate a similar increase in decoding accuracy with increasing neuron counts in individual participants. To assess this, we have replicated the decoding analysis of Fig. 2c with neurons sampled within and across recording sessions

(Extended Data Fig. 9g) and observe a consistent increase of decoding performance with increased neuron counts in both decoding scenarios.

Population decoding offers important additional insights. Pooling neurons within target regions across participants aids the identification of stereotypical neuronal activity patterns across the population.-An important contribution of our study is the comparisons of neuronal odor coding across brain regions involved in olfaction. Subsampling equal numbers of neurons within each brain regions guarantees a balanced comparison of decoding across regions, regardless of neuronal yield in individual participants To describe the rationale and added value of the population decoding analysis, we have included the following paragraph in the manuscript: "In the population decoding, equal numbers of neurons are randomly sampled across recording sessions, enabling a balanced comparison of performance between regions irrespective of individual variations in neuronal yield. Comparing decoding performance of randomly sampled neurons within and across recording sessions yielded consistent results (Extended Data Fig. 9g), indicating that population decoding extrapolates well to larger populations of neurons."

Extended Data Fig. 9g: Odor-decoding accuracy as a function of the number of neurons used for decoding, sampled across participants (red, mean \pm SEM) or within participants (blue, mean \pm SEM across sessions), as in Fig. 2c (100 times randomly subsampled with 8 cross-validation data splits and 10 resample runs). Chance level (6.25%) shown as dashed horizontal line. Note that with 8-32 microwires per anatomical target region, it is rarely possible to simultaneously record the activity of 30 or more neurons per participant.

Following the reviewer's suggestion we have now integrated the session-wise decoding (Extended Data Fig. 6a) in the main figures (Fig. 2e). Additionally, we have performed session-wise image-identity and cross-modal decoding, which paint a picture consistent with the population decoding. We have included these results in Extended Data Fig. 6d-f. Given that the population decoding allows accurate cross-regional comparison and generalizability, we suggest featuring both decoding analyses in the manuscript as both approaches provide important complementary information.

Extended Data Fig. 6 d-f: Session-wise image and cross-modal decoding.

d, Image-decoding accuracy (mean \pm SEM, black) per recording session and region (colored dots). Despite the limited and variable neuron count per session, image identity could be decoded significantly above chance (6.25%, dashed horizontal line) across sessions in PC, Am, EC, and Hp (PC: 7 out of $n=17$ sessions showed significant decoding compared to 1,000 image-label-permuted data, $P=9.7 \cdot 10^{-6}$; Am: 5 out of $n=20$, $P=0.0026$; EC: 5 out of $n=15$, $P=0.00061$; Hp: 10 out of $n=20$, $P=1.1 \cdot 10^{-8}$; PHC: 2 out of $n=17$, $P=0.21$; right-sided binomial test with $P_{chance}=0.05$, regions with ≥ 2 neurons in recordings with both olfactory and visual task). **e-f**, Cross-modal decoding per session trained on images and evaluated on odors (**e**), and *vice versa* (**f**), revealed significant cross-modal coding in PC and Am (Image-to-odor: PC: 4 out of $n=17$ sessions, $P=0.0088$; Am: 2 out of $n=20$, $P=0.26$; EC: 0 out of $n=15$, $P=1$; Hp: 2 out of $n=20$, $P=0.26$; PHC: 0 out of $n=17$, $P=1$; Odor-to-image: PC: 0 out of $n=17$, $P=1$; Am: 4 out of $n=20$, $P=0.016$; EC: 1 out of $n=15$, $P=0.54$; Hp: 2 out of $n=20$, $P=0.26$; PHC: 0 out of $n=17$, $P=1$; right-sided binomial test with $P_{chance}=0.05$, regions with ≥ 2 neurons in recordings with both olfactory and visual task as in (d)).

Comments to Referee 3:

I want to congratulate the authors on an excellent manuscript. They comprehensively addressed all my points. I am particularly impressed by how thoroughly they are now describing their recording methodology, spike sorting, unit quality metrics. This alone would make this a landmark study for the emerging field of human single-unit recordings.

Moreover, as far as I can see they did address the other reviewer's points exhaustively as well. I agree with Reviewer 2 that ideally high-temporal control odor delivery devices and detailed nasal airflow measurements benefit the interpretation of neuronal recordings from olfactory areas. However, I do appreciate the author's discussion of the clinical constraints and in my opinion the unique opportunity to perform these well-designed experiments in patients and the high bar for reproducibility as outlined in the rebuttal response far outweigh any concerns with respect to further optimized odor delivery. In fact, with respect to the specific questions addressed by the authors I would struggle seeing a significant impact of using sniff sticks or the limited accuracy of using plethysmograph measurements as a proxy for airflow. Possibly result variability would go down further with improved sniff- or odor-onset-aligning but I struggle to see any results suffering from their odor delivery limitations. Fig R2.1/ Extended data Fig 5 is quite compelling in this respect.

Congratulations on a very important piece of work.

We thank the reviewer for their important contributions and are flattered by their appreciation of our revised manuscript.